# Population dynamics of head-direction neurons during drift and reorientation

Zaki Ajabi[1,2,7 ✉], Alexandra T. Keinath[1], Xue-Xin Wei[3,4,5,6] & Mark P. Brandon[1,2 ✉]

The head direction (HD) system functions as the brain's internal compass[1,2], classically formalized as a one-dimensional ring attractor network[3,4]. In contrast to a globally consistent magnetic compass, the HD system does not have a universal reference frame. Instead, it anchors to local cues, maintaining a stable offset when cues rotate[5–8] and drifting in the absence of referents[5,8–10]. However, questions about the mechanisms that underlie anchoring and drift remain unresolved and are best addressed at the population level. For example, the extent to which the one-dimensional description of population activity holds under conditions of reorientation and drift is unclear. Here we performed population recordings of thalamic HD cells using calcium imaging during controlled rotations of a visual landmark. Across experiments, population activity varied along a second dimension, which we refer to as network gain, especially under circumstances of cue conflict and ambiguity. Activity along this dimension predicted realignment and drift dynamics, including the speed of network realignment. In the dark, network gain maintained a 'memory trace' of the previously displayed landmark. Further experiments demonstrated that the HD network returned to its baseline orientation after brief, but not longer, exposures to a rotated cue. This experience dependence suggests that memory of previous associations between HD neurons and allocentric cues is maintained and influences the internal HD representation. Building on these results, we show that continuous rotation of a visual landmark induced rotation of the HD representation that persisted in darkness, demonstrating experience-dependent recalibration of the HD system. Finally, we propose a computational model to formalize how the neural compass flexibly adapts to changing environmental cues to maintain a reliable representation of HD. These results challenge classical one-dimensional interpretations of the HD system and provide insights into the interactions between this system and the cues to which it anchors.

The HD system, commonly referred to as the neural compass, underlies a navigator's sense of direction[1,2,11–14]. In contrast to a traditional compass, the orientation of the HD system is anchored to local environmental cues[8,15,16]. Our understanding of the mechanisms that support the ability of the HD network to align with specific cues and maintain a consistent sense of direction remains limited. Recent research has shown that substantial variability in HD cell activity during sleep cannot be explained by a singular angular dimension[17,18]. This observation challenges the classical view of the internal HD representation as a unidimensional (that is, angular) construct and motivates further investigation of its complexity, from a population perspective, to understand whether extra dimensions are needed to fully capture how the system adapts to unstable conditions (that is, changing, missing and/or conflicting sensory information) in wakefulness. While updating the internal HD representation requires integration of information from multiple sensory modalities, manipulations of visual cues alone are sufficient to reorient this representation[6,7,19–21]. The visual input exerts a dominant influence on the HD network alignment, probably through a feedback correction that calibrates the integration of angular movements[22]. Computational models of the HD network suggest that plasticity mediates the integration of visual information within the network[23–25], confirmed recently in fruit flies[26,27] but not yet in mammals. Here we characterize the thalamic HD network response to visual manipulations, yielding new insights into the mechanisms that underlie anchoring and calibration of this representation. Furthermore, we provide a network model to assess the plausibility of synaptic plasticity as a mechanism to explain the observed variability in the system's response to externally controlled changes in visual information.

[1]Department of Psychiatry, Douglas Hospital Research Centre, McGill University, Verdun, Quebec, Canada. [2]Integrated Program in Neuroscience, McGill University, Montreal, Quebec, Canada. [3]Department of Neuroscience, University of Texas at Austin, Austin, TX, USA. [4]Department of Psychology, University of Texas at Austin, Austin, TX, USA. [5]Center for Perceptual Systems, University of Texas at Austin, Austin, TX, USA. [6]Center for Theoretical and Computational Neuroscience, University of Texas at Austin, Austin, TX, USA. [7]Present address: Department of Neurobiology, Harvard Medical School, Boston, MA, USA. ✉e-mail: zaki_ajabi@hms.harvard.edu; mark.brandon@mcgill.ca

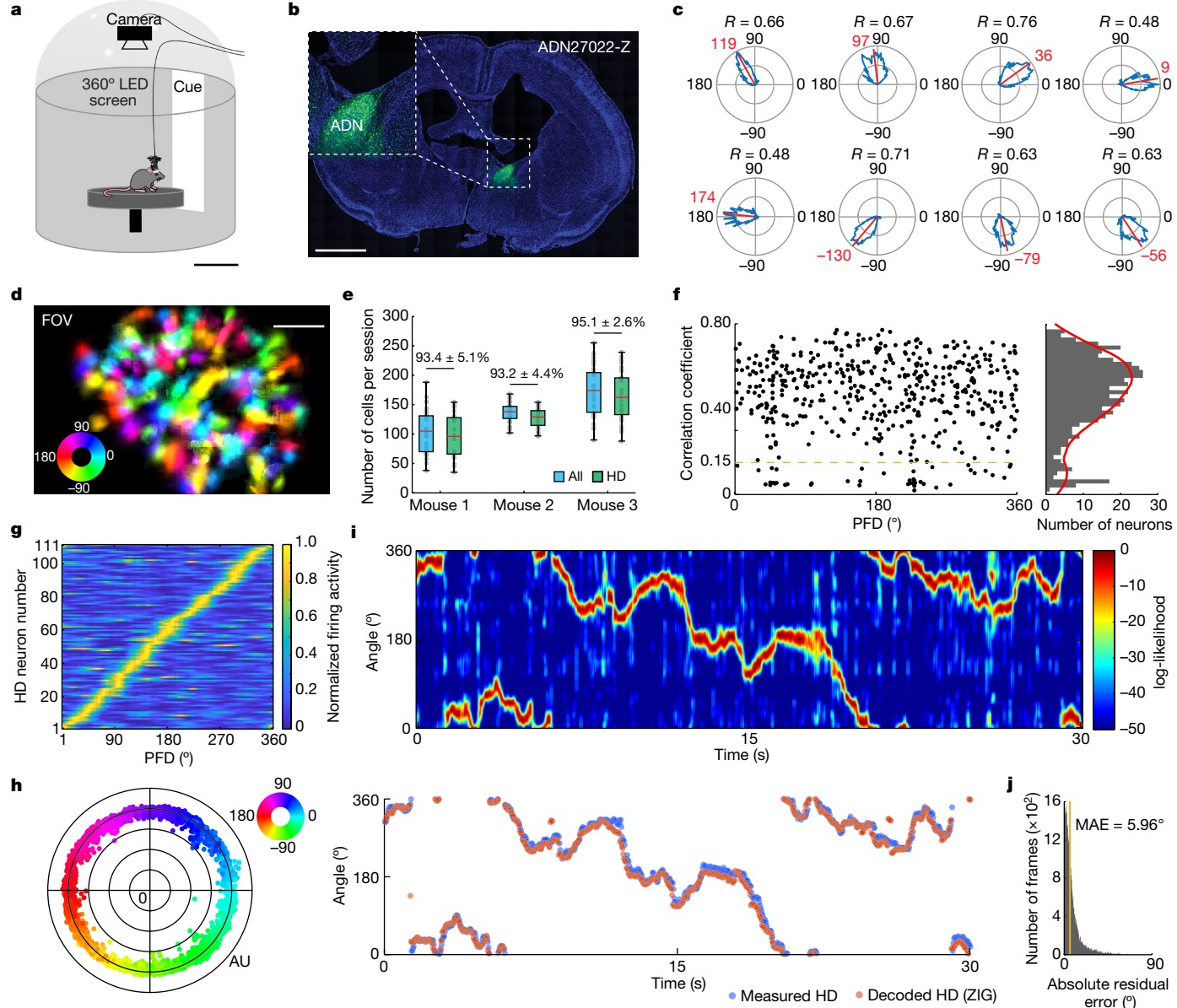

**Fig. 1 | Population recordings in the mouse ADN. a**, Schematic of the recording environment within a 360° LED screen. Scale bar, 20 cm. **b**, GCaMP6f expression in the ADN. In total, 12 mice were injected and implanted for this study, and only 3 (Extended Data Fig. 1a–c) provided enough simultaneously recorded HD cells for continued experimentation. Scale bar, 2 mm. **c**, Example tuning curves of ADN cells with high directional tuning in polar coordinates. The red lines and numbers show the mean resultant vectors and PFD, respectively. *R*, correlation coefficient. **d**, Field of view (FOV) of the ADN showing the PFD of each cell. Scale bar, 0.125 mm. **e**, The distribution of ADN cells recorded across mice (*n* = 3) and sessions (*n* = 99). The red line indicates the median (minimum, maximum, median, 25th percentile and 75th percentile, respectively, are as follows: mouse 1 (all): 38, 188, 105, 70 and 131; mouse 1 (HD): 35, 154, 96, 66 and 128; mouse 2 (all): 102, 168, 138, 126.5 and 147; mouse 2 (HD): 97, 154, 129, 114.75 and

139.75; mouse 3 (all): 90, 255, 174, 137 and 204.5; mouse 3 (HD): 88, 239, 162.5, 133 and 195.5). The values above the box plots indicate the percentage of HD cells (green) among all recorded ADN cells (blue) shown as mean ± s.d. **f**, The distribution of correlation coefficients of ADN cells. The dashed yellow line represents the HD neuron detection threshold (shuffled control: *P* < 0.05). Data from three 10 min baseline recording sessions (one per mouse). **g**, HD population coverage of the azimuthal plane from one session. **h**, Projection of high-dimensional neural data onto a 2D polar plane using a feedforward neural network during a baseline recording. **i**, HD decoding. Top, log-likelihood distribution across time. Bottom, measured HD (blue) and decoded HD (red) using maximum likelihood. **j**, The distribution of the absolute residual error across baseline recordings from the first experiment (*n* = 42 sessions).

## Calcium imaging of the HD network

We performed calcium imaging of the anterodorsal thalamic nucleus (ADN) in three mice using a head-mounted endoscope[28–30]. Our recordings (*n* = 102 20 min sessions, with baseline periods of 3, 5 or 10 min depending on the experiment) enabled us to monitor up to 255 ADN cells simultaneously as mice freely explored a small elevated circular platform inside a larger enclosed chamber (Fig. 1a–g and Extended Data

Fig. 1a–c). This chamber was composed of a 360° circular LED screen covered by an opaque dome (Fig. 1a). During baseline recordings at the start of each session, we displayed a polarizing vertical white stripe on the otherwise black LED screen. All subsequent testing involved the manipulation of this visual cue.

Calcium imaging data were motion-corrected and spiking activity was inferred from extracted fluorescent transients[31,32] (Extended Data Fig. 1d). Baseline recordings revealed HD cells with preferred firing

directions (PFDs) that tiled the full 360° horizontal plane (Fig. 1g and Extended Data Fig. 1c). Consistent with previous research, ADN neurons were tuned to specific HDs[1,33,34] albeit with higher proportions (Fig. 1c–g and Extended Data Figs. 1 and 2). HD tuning was stable in the presence of visual cues[34] (Fig. 1c and Extended Data Figs. 1d and 2), and exhibited anticipatory firing[35] (Extended Data Fig. 3a). In contrast to the HD system in the central complex of *Drosophila*[36], we did not observe topographic organization (Fig. 1d and Extended Data Fig. 1b).

To infer the internal HD representation (referred to here as internal HD), we trained a Bayesian decoder[37] to estimate the animal's HD on the basis of the baseline training data (Fig. 1i). This decoder accurately recovered the measured HD in stable experimental conditions (median absolute error (MAE) of test data = 5.96°; Fig. 1j).

To visualize the low-dimensional structure of the HD representation, we developed a method to project large ensemble recordings onto a two-dimensional (2D) polar state space (Methods). We trained a deep neural network on the measured head direction while allowing an untrained latent variable to capture variability in the neural data that cannot be explained by changes in the head direction alone. This inferred latent variable constitutes the radial component in the 2D polar state space (that is, secondary dimension). When applied to the baseline data, we obtained a ring-like structure (Fig. 1h), reminiscent of ring attractor models and previous analyses[3,17,38]. This further confirms that the internal HD representation is approximately unidimensional in stable conditions.

## Network gain covaries with reset dynamics

To investigate how HD network dynamics enable reorientation, we recorded the HD network during a cue-shift paradigm. After a baseline recording, the cue was removed for 2 min (darkness) and then reappeared at a 90° shifted position for 2 min. We repeated this sequence four times per recording session (Fig. 2a).

This manipulation resulted in predictable changes in the HD network's patterns of activity. To characterize these dynamics, we defined an 'offset' as the mismatch between the measured and decoded HD (see the 'Analysis of drift' section of the Methods). Tracking this offset, we observed a rotational response after cue reappearance that matched the cue shift. We refer to this phenomenon as a reset (Fig. 2a). Notably, resetting events were not homogeneous, as they occurred across a wide range of angles and speeds (Fig. 2b).

Cue shifts induced significant changes in the overall network activity. We observed modulation in the amplitude of the bump of activity (see the 'Reconstruction of the bump of activity' section of the Methods), which coincided with changes in the radius of the latent space (Fig. 2c,d). Intuitively, allowing the internal HD representation to occupy a 2D polar state space makes the distance between any two given angles $\theta_1$ and $\theta_2$ a function of the radial component, which led us to hypothesize that changes in radius not only reflected changes in the overall population activity but would also correlate with changes in the speed of reset (Extended Data Fig. 4). To quantify this, we computed the total population activity normalized to the baseline activity, a measure that we refer to as the network gain (Extended Data Fig. 5; see the 'Calculation of network gain' section of the Methods). State-space radius was highly correlated with network gain (Fig. 2d,e), indicating that gain can be used as an interpretable measure of the radial component. To better understand the relationship between resetting events and network gain, we first analysed the 90°-centred reset range (that is, [70:110]° range). We found that the speed of HD-network reorientation, or 'reset speed', was anticorrelated with network gain (Fig. 2g–i). Separation of the resetting events into two groups (Fig. 2g) revealed that fast resets were associated with a substantial reduction in network gain shortly after cue reappearance, whereas slow resets exhibited a smaller reduction in gain (Fig. 2h and Extended Data Fig. 6a). In all cases, resetting events took the form of a continuous rotation of the HD representation from

an initial orientation to the reset direction, passing by all intermediate angles without the appearance of secondary bumps of population activity (Extended Data Fig. 7a). This reset was slower than what has previously been reported[6], possibly due to differences in behaviour, habituation and/or the geometry of the testing set-up. Our modelling results show that an attractor network model that incorporated network gain replicated these dynamics with 71% accuracy in classifying fast versus slow resets (Fig. 2j,k and Extended Data Figs. 7b–d).

Behavioural differences as measured by the head angular velocity before and after cue onset could not explain the sharp decrease in gain amplitudes (Extended Data Fig. 8a). However, reduced head angular velocity immediately preceding cue events was predictive of fast resets, and vice versa (Extended Data Fig. 8b,c).

Resetting events also varied in the angular difference between their initial and stabilizing orientations. We grouped resets by the distance between pre-cue offset and the offset after stabilization (Extended Data Figs. 6b and 9a), and found that network gain was anticorrelated with reset range (Extended Data Figs. 6b and 9b,c). This relationship was independent of reset speed (Extended Data Fig. 9a), suggesting that network gain is independently modulated by the estimated error between the internal representation and the actual location of the visual cue.

We also detected a rapid spike in gain at the cue onset that was largest in short-range resets (Extended Data Fig. 9d). This may reflect visual inputs, but further investigation was limited by the temporal resolution of calcium imaging.

## HD neurons maintain a trace of the cue

The HD network drifts in the absence of visual cues[5,8–10]. We hypothesized that network gain would decrease once the visual cue is removed due to decreased sensory input. This would bring the HD system to a lower energy state and cause the internal representation to become prone to spontaneous shifts because of the decrease in signal-to-noise ratio of neural activity. During all darkness epochs (D1 to D4), we observed an increase in drift relative to baseline (Fig. 3a,b and Extended Data Fig. 10a), which coincided with an abrupt decrease in the network gain after cue removal (Fig. 3c). Notably, changes in the network gain were dependent on the internal HD during darkness. When the internal HD pointed towards the internal location of the visual cue (0°), the reduction in gain was minimal; deviations from this direction resulted in more pronounced gain decreases (Fig. 3d,e). This gain profile persisted across all darkness periods with the difference between peak and trough increasing from the first to last darkness epoch (Extended Data Fig. 10d,e). Animal behaviour did not significantly affect the gain landscape, except that gain amplitude increased with the absolute head angular velocity (Fig. 3e and Extended Data Fig. 11), consistent with previous observations[36,39]. This suggests that the HD network maintains a 'memory trace' of salient visual cues.

Drift patterns were not homogenous across the four darkness epochs (Extended Data Fig. 10a–c). During D1, drift fluctuated around the baseline orientation with no directional bias. By contrast, drift in D2, D3 and D4 exhibited directional biases dependent on the baseline orientation and previous cue location. During D2, drift diverged from its reset orientation towards its baseline orientation, counter to the rotation implied by the previous cue shift. During D3 and D4, drift rotated towards the baseline orientation but consistent with the direction implied by the previous cue shifts. These observations indicate that drift depends on previous visual experience. These predictable drift biases after exposure to the changing visual reference frame (D2 to D4) appeared to consistently bring the HD network closer to its original configuration (that is, the baseline state). Consecutive shifts of the visual cue in one direction further biased the drift in that direction (Extended Data Fig. 10b,c). These results suggest that both the stable allocentric and dynamic visual reference frames exert a persistent influence on the network orientation.

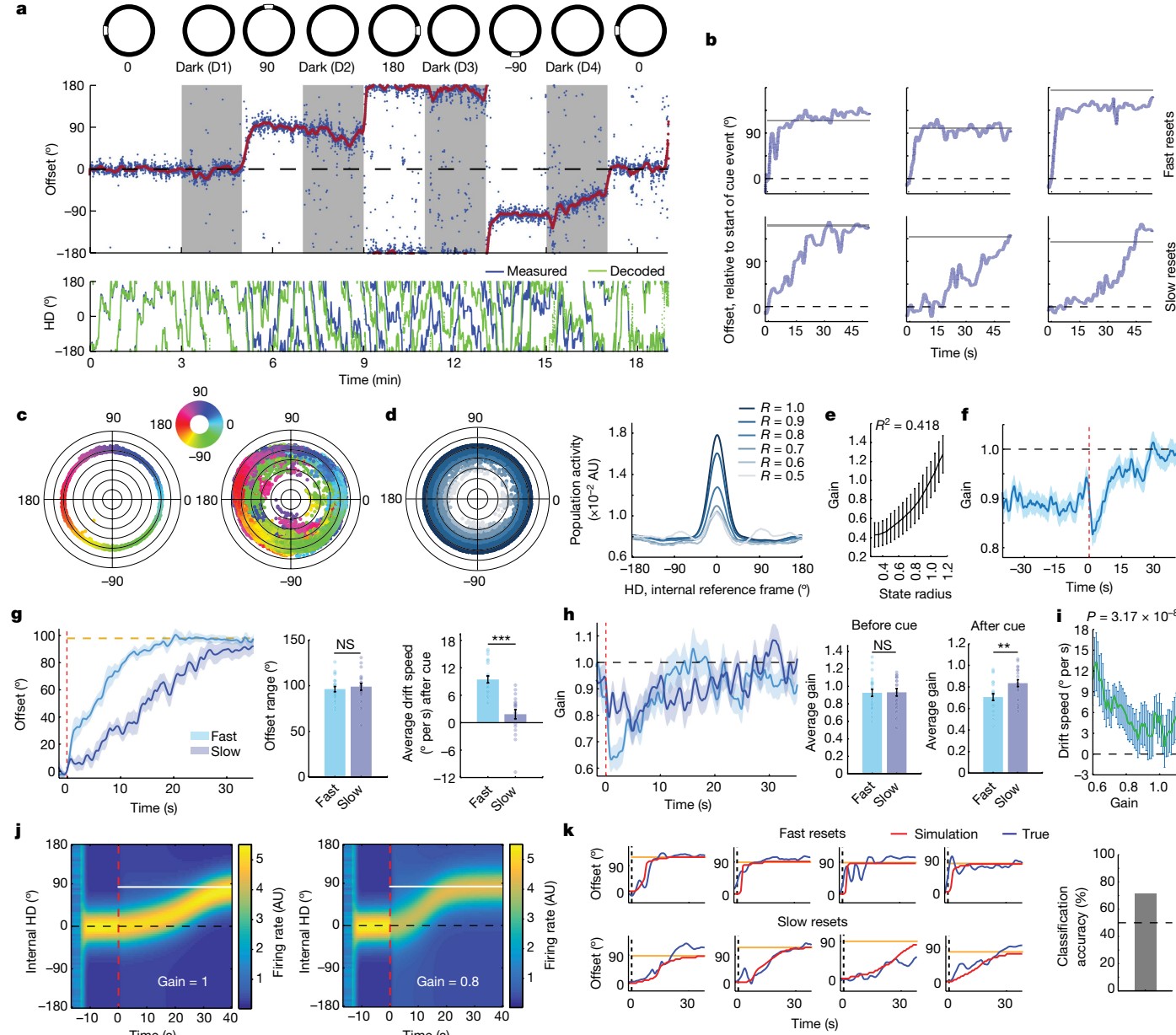

**Fig. 2 | Network gain covaries with resetting dynamics. a**, Experimental protocol (top). Middle, example session showing the offset obtained by subtracting decoded from measured HD (blue dots). Red, smoothed offset. Darkness periods are shaded in grey. Bottom, measured (blue) and decoded (green) HD. **b**, Example fast (top) and slow (bottom) resets. The horizontal solid line indicates the cue location. The offset is relative to its angle at the cue onset. **c**, Projection of population activity onto the polar plane for the baseline (left) and the entire session (right). **d**, The same as in **c**; however, points are shaded by their radius (left). Right, mean bump of activity in the internal reference frame across radius ranges. **e**, The relationship between network gain and state radius ($n = 42 \times L$ datapoints, where $L$ is the number of frames in a session). $R^2$ value of linear regression model fit. Data are mean ± s.d. **f**, Triggered average of network gain ($n = 168 = 4 \times 42$ cue events). The dashed red line indicates the cue display. **g**, Mean offsets for fast (light blue; $n = 22$ resets) and slow (dark blue; $n = 20$ resets) resets. Both groups have similar ranges (two-sided Wilcoxon rank-sum test, $P = 0.4131$, $Z = 0.82$), yet their speeds are different (two-sided Wilcoxon

rank-sum test, $P = 1.0982 \times 10^{-6}$, $Z = 4.87$; 150 frames (-5 s) after the cue). **h**, Network gains of fast and slow reset groups have similar amplitudes before cue display (two-sided Wilcoxon rank-sum test, $P = 0.6234$, $Z = 0.49$; 50 frames (-1.67 s) before the cue), yet are different after cue display (two-sided Wilcoxon rank-sum test, $P = 0.0085$, $Z = 2.63$; 150 frames (-5 s) after the cue). The same data as in **g**. **i**, The relationship between gain and reset speed within 150 frames (-5 s) after the cue ($n = 42 \times 150$ datapoints). The $P$ value was calculated using an $F$-test on a linear model fit. **j**, Simulation of the bump of activity showing gain control of reset speed. The gain remains constant after cue display (dashed red). The solid white lines show the relative cue location. **k**, Model-based prediction (red) and true reset (blue). The dashed black lines indicate the cue display. The solid yellow lines indicate the relative cue location. All clockwise sessions were reflected across the *x* axis and transformed into counter-clockwise ones. Time-dependent signals in **f**–**h** are shown as mean ± s.e.m. Bar graphs and error bars, except in **e**, show mean ± s.e.m. with individual datapoints.

## Drift patterns are experience-dependent

The presentation of a rotated visual cue for 2 min was sufficient to cause a representational shift and override the influence of non-visual cues (self-motion, olfactory and so on). Yet, we observed a tendency of the network to rotate back towards the initial configuration, that is, revert to baseline, during darkness. We hypothesized that this could implicate plastic processes through which a 'memory' of baseline state exerts influence on the internal HD representation. If true, drift dynamics might depend on the duration of exposure to the shifted-cue context.

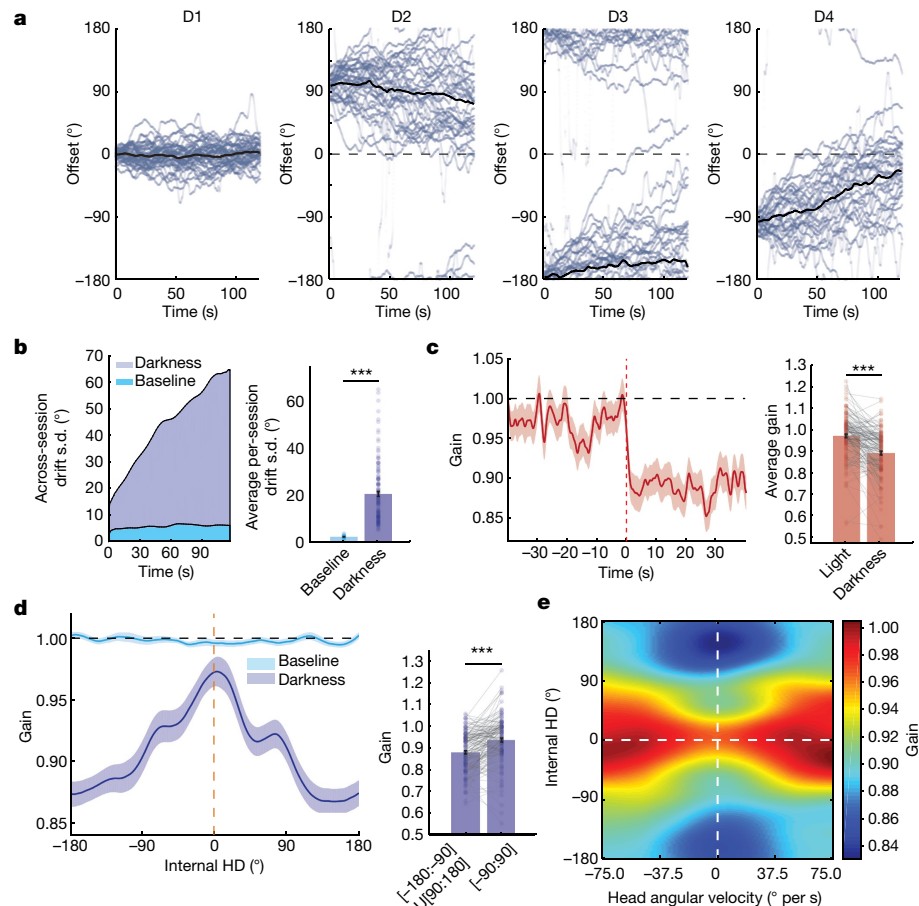

**Fig. 3 | The network gain maintains a trace of the visual cue in darkness.**
**a**, Offset traces (light blue) across darkness events D1 ($n = 42$), D2 ($n = 35$), D3 ($n = 33$) and D4 ($n = 35$). The black lines show the mean drifts. For D2, D3 and D4, only darkness epochs that follow a correct reset were considered. **b**, Drift variability increases with time during dark epochs compared with the baseline (baseline: $n = 42$ events; darkness: $n = 145$ events; two-sided Wilcoxon rank-sum test, $P = 7.5211 \times 10^{-23}$, $Z = 9.84$). **c**, The triggered average of network gain shows an abrupt drop at the transition between cue-on and cue-off epochs, marked by the dotted red line (two-sided Wilcoxon rank-sum test, $P = 9.0165 \times 10^{-12}$, $Z = 6.81$; comparison between mean values over 40 s before and 40 s after cue removal; the same data as in **a**). **d**, Network gain tuning curves at the baseline

(light blue) and during darkness (dark blue). The internal HD is relative to the baseline cue location (dashed yellow line). The gain remains flat at the baseline; however, it peaks around the internal cue location ([−90:90]°) and drops sharply away from it ([−180:−90]U[90:180]°) in darkness ($n = 145$ events; two-sided Wilcoxon rank-sum test, $P = 5.8683 \times 10^{-7}$, $Z = 5.00$). **e**, Network gain heat map. Note the increase in amplitude and width of the gain tuning curve at a larger head angular velocity. All clockwise sessions were reflected across the $x$ axis and transformed into counter-clockwise ones. Signals in **c** and **d** are shown as mean (solid line) ± s.e.m. (shaded area). Bar graphs in **b**−**d** show the mean ± s.e.m. with individual datapoints.

To test this, we limited the display of the rotated visual cue to 20 s (±90° from the baseline; Fig. 4a). These shortened cue events elicited resets followed by reversions towards baseline during darkness (Fig. 4a−c). However, in comparison to the 2 min experiment (specifically D2, which was similarly preceded by a ±90° rotated cue event), reversion was much stronger after the presentation of a 20 s visual cue (Fig. 4c). The attraction of the network to its baseline state was further demonstrated through vector field analysis (Fig. 4d; see the 'Vector field analysis' section of the Methods). These results indicate that the internal representation of the baseline allocentric reference frame is not entirely lost after a reset and can still influence the HD network in darkness, depending on the duration of experience within the competing reset reference frame context. Addition of Hebbian learning to our model shows that, indeed, HD neurons could form new associations with the unchanged allocentric cues, depending on the duration of exposure to the reset context. Given enough time, the synaptic strength of these new associations increased while old associations were depleted, resulting in a new steady state. In this scenario, our simulations of the internal HD representation showed limited baseline attraction. By contrast, synaptic weights did not change significantly after shorter (20 s) cue

exposures, and baseline associations remained dominant, resulting in strong reversions (Extended Data Fig. 12).

Next, we examined potential competition between baseline and reset reference frames by comparing network gain patterns after long and short shifted-cue exposures. While the gain landscape during D2 exhibited a single peak at 0° in the internal HD (corresponding to the shifted cue orientation after reset), the gain landscape during darkness after the 20 s cue events exhibited additional peaks at ±90° (Fig. 4e,f and Extended Data Fig. 13). Notably, these peaks matched the alternating ±90° cue structure of the experimental design, suggesting the coexistence of associations between HD neurons and allocentric cues from multiple visual contexts. These differences provide additional evidence of time-dependent effects of visual experience.

## Cue rotation causes persistent drift bias

In the 2 min cue-shift experiment, visual information provided a dominant polarizing cue to reset the HD system. In some cases, resets were slow (>30 s), indicating that non-visual cues competed with visual information to stabilize the network. The 20 s cue-shift experiment

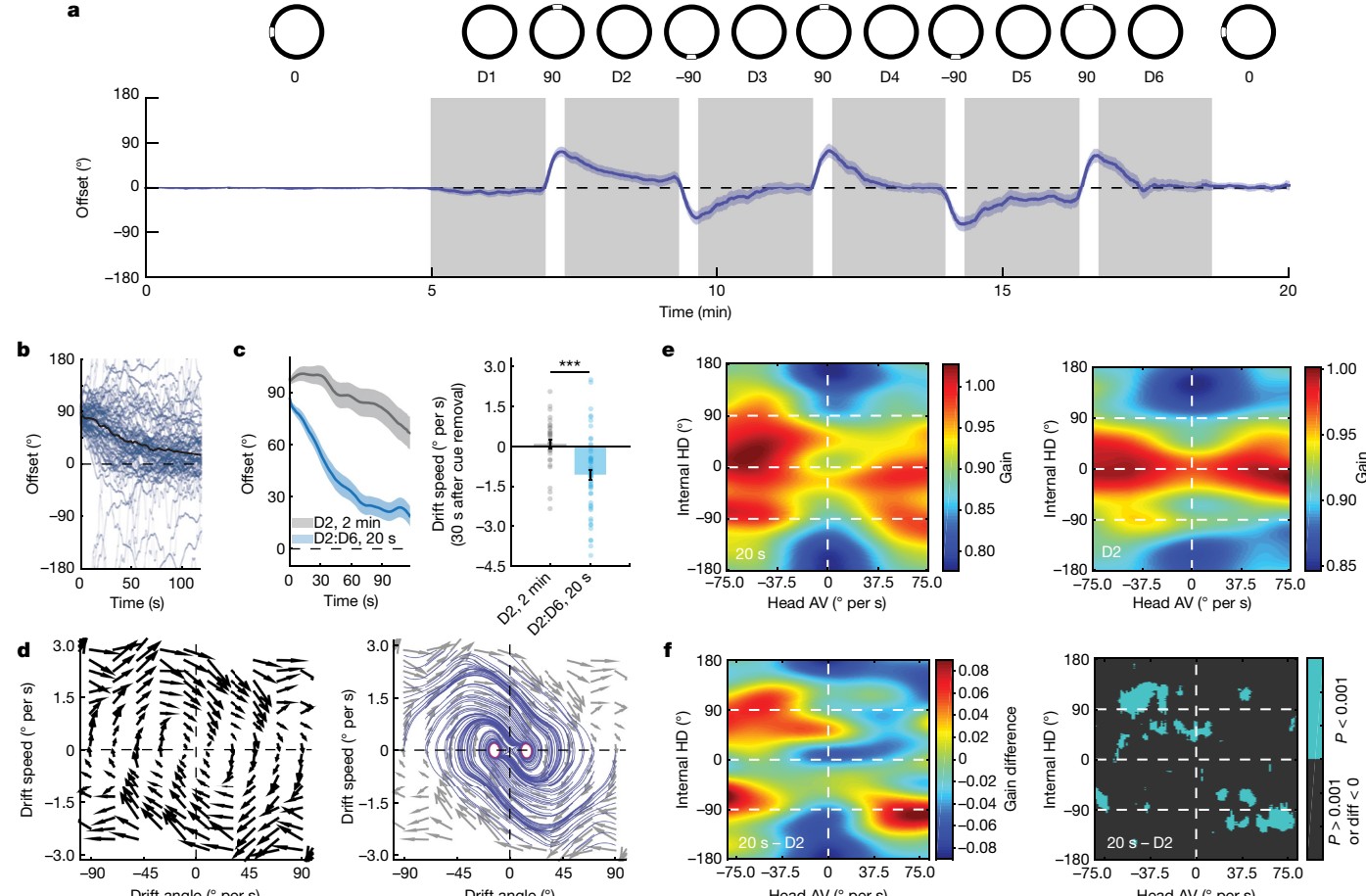

**Fig. 4 | Attraction of internal representation to the baseline reference frame is time dependent. a**, Experimental protocol (top row). Bottom row, average across-session offset (relative to baseline) during the 20 s cue-shift experiment. $n = 18$ sessions. Darkness periods are highlighted in grey. **b**, Individual offset traces after resets (light blue; $n = 58$ events) across darkness periods D2 to D6. The black line shows the mean offset. Offset traces after a −90° reset were reflected across the 0° axis. **c**, Mean offset during darkness in the 20 s cue-exposure experiment ($n = 58$ events; dark blue) and in D2 of the 2 min cue-exposure experiment ($n = 34$ events; grey) (left). Right, mean drift speed over the first 30 s shows a strong reversion to the baseline after a 20 s cue exposure (two-sided Wilcoxon rank-sum test, $P = 1.4264 \times 10^{-5}$, $Z = 4.34$). Data are mean ± s.e.m. with individual datapoints. **d**, Drift vector field (left). The

arrows indicate the direction of mean drift speed and mean drift acceleration (from $n = 58$ events). The arrow length was scaled down for illustration. Right, simulated streamlines. The stable regime is highlighted in red. **e**, Network gain heat maps. Left, 20 s cue-exposure experiment. Data represent instances of reversion to the baseline ($n = 43$ events). Right, D2 of the 2 min cue-exposure experiment ($n = 34$ events). **f**, The gain difference between heat maps in **e** showing the appearance of new bumps at the locations of cue-shifts (±90°) (left). Right, $P$-value matrix for the data on the left (two-sided Wilcoxon rank-sum test; pixels where $P > 0.001$ and/or gain (20 s) < gain (D2) are marked as not a number (NaN)). Time-dependent signals in **a** and **c** are shown as mean (solid line) ± s.e.m. (shaded area).

provided further evidence that the baseline reference frame maintains a persistent influence on the HD network. To better understand the dynamics of this competition, we tested whether visual information could drive resetting when in continuous conflict with non-visual information, including self-motion cues. We recorded HD cell populations during presentation of a slow rotating visual cue (1.5 or 3.0° per s) for 7 min (Fig. 5a,b). In all cases, and for both speeds, the HD network was continuously updated by the visual cue (Fig. 5a–c), highlighting the dominant effect that the visual input has over all other inputs in controlling the HD system.

Notably, the HD network continued to rotate in the same direction and at a similar speed when the rotating cue was turned off (Fig. 5a,b,d and Extended Data Fig. 14a; see the 'Analysis of cue-rotation sessions' section of the Methods). This persistent bias was replicated in our network model by adding a recalibration circuit to asymmetrically change the strength of vestibular input through visual feedback (Extended Data Fig. 14b–e and Supplementary Information). We also observed an attraction to the baseline internal representation, similar to our prior experiments. The system started to stabilize once the internal

HD representation approached the baseline state (Fig. 5e and Extended Data Fig. 14a). This phenomenon could also be reproduced in our model in which, after 7 min of cue rotation, no strong new associations between HD neurons and allocentric cues could emerge to form a new steady state. Instead, baseline associations remained dominant, albeit with a significant weight decay in the synaptic matrix (Extended Data Fig. 14d). Together, these results indicate that experience with dynamic reference frames can bias the HD network and implicate asymmetric recalibration of vestibular input integration within the HD network as a potential source of this bias.

## Discussion

We combined large population recordings of ADN neurons with visual cue manipulations to characterize the population dynamics of the mammalian HD system. Controlled manipulations of a visual cue induced global fluctuations in network activity captured by a measure that we termed network gain. Network gain represents a functional dimension in the internal HD representation, in addition to its

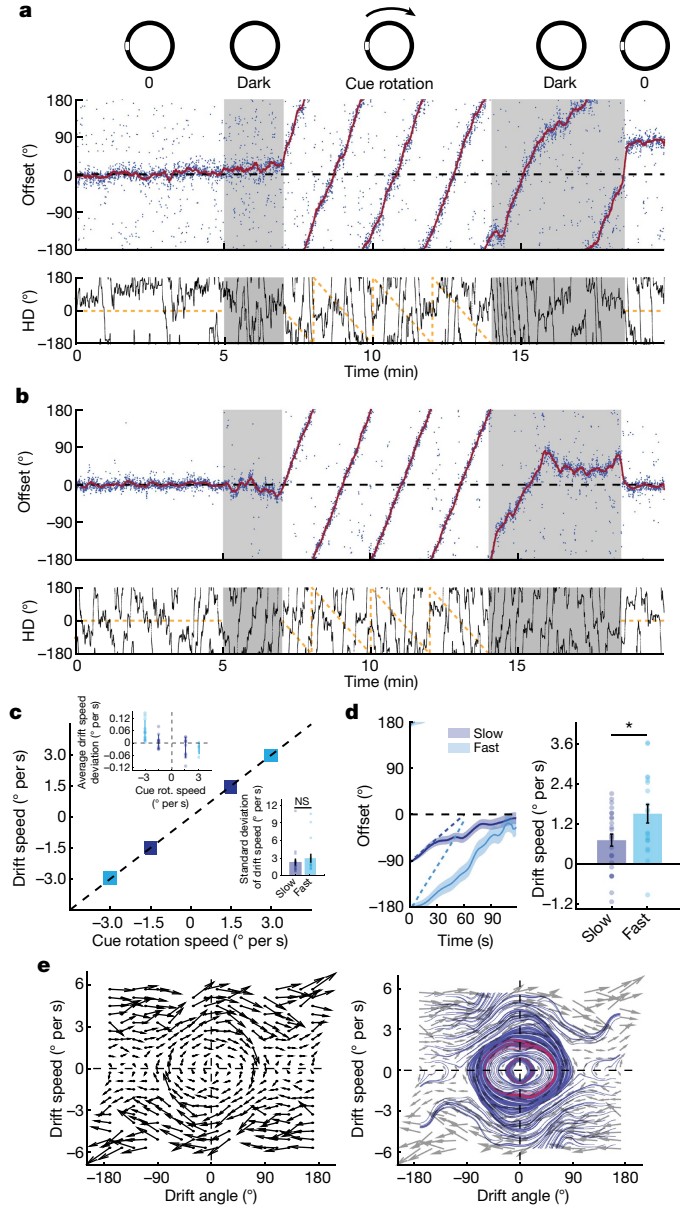

**Fig. 5 | Optic flow calibrates integration of change in HD. a**, Experimental protocol (top). Middle, example offset (blue dots) during fast cue rotation (3° per s) showing persistent drift bias after cue removal. The solid red lines show low-pass filtered offset. Bottom, cue location (dashed yellow) and measured HD (solid black) relative to baseline cue location. Darkness periods are shown in grey. **b**, Example fast-cue-rotation session showing stabilization of the internal representation with an overshoot past the baseline orientation (top). Bottom, the same as in **a**. **c**, Mean drift speed during cue rotation (rot.) for fast (light blue) and slow (dark blue) sessions. Data are across-session mean ± s.d. Top left inset: subtraction of the cue-rotation speed from the average drift speed for individual sessions shows the average deviation of drift-speed with reference to the cue-rotation speed per session. Bottom right inset: comparison of drift-speed s.d. values between fast and slow sessions (two-sided Wilcoxon rank-sum test, $P = 0.5262$, $Z = 0.63$). **d**, The mean offset for fast (light blue; $n = 19$ events) and slow (dark blue; $n = 25$ events) sessions (left). The dotted lines correspond to the natural progression of offset if drift speed matched the speed of cue rotation. Data are mean (solid line) ± s.e.m. (shaded area). Right, drift-speed comparison between fast and slow sessions in the first minute after cue removal (Wilcoxon rank-sum test, $P = 0.0393$, $Z = 2.06$). All clockwise sessions were reflected across the x axis and transformed into counter-clockwise ones. **e**, Drift vector field (left). The arrows indicate the direction of mean drift speed and mean drift acceleration ($n = 60$ sessions). Arrow length was scaled down for illustration purposes. Right, simulated streamlines. The stable regime is highlighted in red. In **a**–**d**, fast sessions in which the offset angle at the beginning of the second darkness was within [−180:−145]U[145:180]° were considered, whereas slow sessions in which the offset's initial position in the second darkness was within [−125:−55]° were included. In **e**, all sessions were considered regardless of the offset angle at the start of the second darkness. For **c** and **d**, data are mean ± s.e.m. with individual datapoints.

Network gain reduction during realignment of the HD network suggests that a feedback signal downstream of ADN provides global inhibition to the network. Similar ideas have been proposed in the central complex of fruit flies[41,42]. Modulation of global neural activity might allow the HD system to operate at different energy levels with varying degrees of stability, reflecting uncertainty in the HD representation. We hypothesize that the animal's engagement in exploratory behaviour together with increased familiarity with the experimental environment and its geometric specificities could sustain a high-gain/high-certainty regime of operation and cause resistance to HD network reorientations imposed by visual cue shifts.

Recent research in fruit flies demonstrated plasticity between visual input and the compass neurons[26,27]. The current study complements these studies and provides evidence for experience-dependent influence of visual landmarks in mammals. Indeed, the mammalian brain appears to maintain a memory of the associations between HD neurons and visual landmarks, in the form of preferential firing, long after landmarks disappear. We propose that memory traces of salient cues in ADN cells help to stabilize the HD system during navigation, even in the absence of reliable environmental anchors by maintaining high activity levels (that is, high signal-to-noise ratio) around internal cue locations. Whether these network gain bumps have an active role in guiding navigation behaviour remains an open question. Moreover, baseline-state attraction during darkness is further evidence of long-term effects in the HD system. As the strength of this attraction depends on the duration of exposure to the shifted-cue context, we speculate that the underlying mechanisms leading to such behaviour involve synaptic plasticity. Through network modelling, we demonstrated that, by adding Hebbian learning, the observed drift patterns could be replicated. Our model proposes that neurons akin to the ring cells of fruit flies could mediate experience-dependent drift dynamics. Whether such neurons exist in the mammalian brain is yet to be determined.

The fact that a continuously rotating visual scene caused persistent biases in the HD representation, as demonstrated in our cue-rotation

classically appreciated angular dimension. Reorientations, in response to visual cue shifts, were associated with a transient decrease in network gain, the magnitude of which was correlated with the speed of the HD network's realignment with the rotated visual reference frame (that is, reset). By extending a standard model of the HD system[40] to incorporate network gain, we were able to predict the speed of the reset response. These results suggest that modulation of network gain provides the HD system with a mechanism to rapidly reorient. Network gain also reflected the past experience of the system—a polarizing visual landmark induced persistent distortions in the network gain profile, forming a memory trace in darkness. These network gain patterns were dependent on the duration of the previous shifted-cue exposure, suggestive of plastic processes in the HD network. Evidence for plasticity was further strengthened by experience-dependent drift behaviours in darkness periods. Incorporating Hebbian plasticity into our network model replicated these observations. Finally, the HD system anchored to a continuously rotating visual cue and continued to rotate after the cue was removed. A model of asymmetric vestibular input recalibration reproduced these results, suggesting that the integration of vestibular information within the HD network is also experience dependent.

experiment, suggests that the integration of vestibular inputs undergoes an experience-dependent calibration. Our findings complement a similar finding in place cells[22] and support a model of hierarchical transfer of information from HD neurons to downstream cells of the navigation system (that is, place cells, grid cells and so on) to maintain consistent and flexible cognitive maps[1,41,43,44].

Ultimately, our findings provide insights into the mechanisms that govern realignment and stabilization of the HD network, and how long-term effects of previous experience affect its dynamics. Importantly, these findings highlight the complexity of the internal HD representation and motivate studying this cognitive system in a multidimensional framework. Here we show evidence for the functional importance of the global fluctuations in network activity (that is, gain) as a critical, yet previously underappreciated, dimension. Future studies examining the origins of such fluctuations will be critical to unveil the complete picture of the intrinsic structure of this circuit.

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

## Methods

### Mice

Twelve male wild-type mice (C57Bl/6, Charles River) were used for this study, three of which provided enough simultaneously recorded HD cells for continued experimentation. Mice were housed individually at 22 °C under a 12 h–12 h light–dark cycle and 40% humidity with food and water ad libitum. All of the experiments were performed in accordance with McGill University and Douglas Hospital Research Centre Animal Use and Care Committee (protocol 2015-7725) and in accordance with Canadian Institutes of Health Research guidelines.

### Surgeries

During all surgeries, mice were anesthetized by inhalation of a combination of oxygen and 5% isoflurane before being transferred to the stereotaxic frame (David Kopf Instruments), where anaesthesia was maintained by inhalation of oxygen and 0.5–2.5% isoflurane for the duration of the surgery. Body temperature was maintained with a heating pad and eyes were hydrated with gel (Optixcare). Carprofen (10 ml kg$^{-1}$) and saline (0.5 ml) were administered subcutaneously, respectively, at the beginning and end of each surgery. Preparation for recordings involved three surgeries per mouse. First, at the age of 7–8 weeks, each mouse was injected with 600 nl of the non-diluted viral vector AAV9. syn.GCaMP6f.WPRE.eYFP, sourced from University of Pennsylvania Vector Core. All injections were administered through glass pipettes connected to the Nanoject II (Drummond Scientific) injector at a flow rate of 23 nl s$^{-1}$. One week after injection, a 0.5-mm-diameter gradient refractive index (GRIN) relay lens (Go!Foton) was implanted above the ADN (AP, −1.05; ML, 0.8; DV, −3). No aspiration was required. In addition to the GRIN lens, three stainless steel screws were threaded into the skull to stabilize the implant. Dental cement (C&B Metabond) was applied to secure the GRIN lens and anchor screws to the skull. A silicone adhesive (Kwik-Sil, World Precision Instruments) was applied to protect the top surface of the GRIN lens until the next surgery. Then, 2 weeks after lens implantation, an aluminium baseplate was affixed by dental cement (C&B Metabond) to the skull of the mouse, which would later secure the miniaturized fluorescent endoscope (miniscope) in place during recording. The miniscope/baseplate was mounted to a stereotaxic arm for lowering above the implanted GRIN lens until the field of view contained visible cell segments, and dental cement was applied to affix the baseplate to the skull. A polyoxymethylene cap was affixed to the baseplate when the mice were not being recorded to protect the baseplate and lens. After surgery, animals were continuously monitored until they recovered. For the initial 3 days after surgery, the mice were provided with a soft diet supplemented with Carprofen for pain management (MediGel CPF). Screening and habituation to recording in the experimental environment began 2–3 days after the baseplate surgery. The first 3–4 weeks of recordings were used to confirm the quality and reliability of the calcium data while the animal was exploring the environment with different screen displays.

### Data acquisition

In vivo calcium videos were recorded with a miniscope (v3; https://miniscope.org) containing a monochrome CMOS imaging sensor (MT9V032C12STM, ON Semiconductor) connected to a custom data acquisition (DAQ) box (https://miniscope.org) with a lightweight, flexible coaxial cable. The cable was attached to a noiseless pulley system with a counterbalance (placed outside the recording environment) to prevent interference with the recorded animal's movements and to alleviate the weight of the miniscope. The DAQ was connected to a PC with a USB 3.0 SuperSpeed cable and controlled using Miniscope custom acquisition software (https://miniscope.org). The outgoing excitation LED was set to 3–6%, depending on the mouse, to maximize the signal quality with the minimum possible excitation light to mitigate the risk of photobleaching. The gain was adjusted to match the dynamic range

of the recorded video to the fluctuations of the calcium signal for each recording to avoid saturation. Behavioural video data were recorded using a webcam mounted above the environment. The DAQ simultaneously acquired behavioural and cellular imaging streams at 30 Hz as uncompressed AVI files and all recorded frames were timestamped for post hoc alignment. Two controllable LEDs (green and red) were added and used for tracking such that, whenever the miniscope was attached to the baseplate, the green LED pointed to the right side of the mouse's head and the red LED pointed to the left side. All other light sources from the miniscope were covered. All recordings took place inside a 360° LED screen (height: 1 m, diameter: 90 cm; Shenzhen Apexls Optoelectronic), at the centre of which we placed a wall-less circular platform (diameter, 20 cm) raised 50 cm above the ground. Mouse bedding was evenly spread over the platform before each recording session. In all recordings, mice were free to move on top of the raised platform. A half spherical dome was used to cover the environment and prevent external light from entering, while it also held the behavioural camera. The experimental environment was designed to maximize circular symmetry, in the absence of any screen display. During habituation, mice were recorded while exposed to a single vertical stripe or no visual display (darkness). These recordings were also used to confirm the quality of tracking the head direction and the cue location, in different conditions. In all experiments in this study, the visual cue refers to a single white vertical stripe (width, 15 cm; height, 1 m).

### Data preprocessing

Calcium imaging data were preprocessed before analyses through a pipeline of open-source MATLAB (MathWorks; v.R2015a) functions to correct for motion artifacts[45], segment cells and extract transients[32,46], and infer the likelihood of spiking events by deconvolution of the transient trace through a first-order autoregressive model[31]. We wrote a MATLAB (MathWorks, v.2015a) program to perform offline tracking of the LEDs and determine, at each frame, the animal's head direction. Another custom-written program was used to estimate the location of the visual cue. Both scripts were incorporated into the preprocessing pipeline.

### Data analysis

In this study, neural activity refers to the deconvolved calcium traces as described previously[31] unless specified. The resulting time series (per neuron, per session) correspond to the inferred likelihood of spiking events. A moving average filter of width 3 frames (~100 ms) is then applied on each time series. We refer to the obtained signal as firing activity.

### Identification of HD cells

For every identified cell segment (ROI), we construct an HD tuning curve by measuring the occupancy-normalized firing activity within each angle bin (1° per bin) of the horizontal plane ($x$ axis). The tuning curve is circularly smoothed with a moving average filter of width 50°. This enables us to have a better estimate of the angle bin that corresponds to the maximum firing activity of a given neuron's tuning curve, which we will refer to as the PFD. We next construct a stimulus signal for that specific PFD by convolving the measured HD signal (from the behavioural camera) with a narrow Gaussian kernel (mean = PFD, s.d. = 17°) such that for every neuron $i$:

$$\text{stim}_i(t) = e^{-\frac{(\text{angdiff}(\text{PFD}_i, \theta_{\text{HD}}(t)))^2}{2\sigma^2}}$$

Where, $\theta_{\text{HD}}$ is the measured HD time series, $\sigma$ is the s.d. of the Gaussian kernel and, angdiff $(a, b)$ is a MATLAB function that gives the subtraction of $a$ from $b$, wrapped on the [−π,π] interval. We correlate the stimulus signal with a normalized version of the firing activity to obtain the Pearson correlation coefficient $r$ of each neuron. To determine the threshold value of $r$ above which a cell can be identified as an HD neuron, we used data from ten baseline recordings (3 min) per animal,

randomly selected from the reset experiment. We start with a relatively high value $r_{\text{thresh}}$ and select all neurons such that $r > r_{\text{thresh}}$. For each neuron, we produce 1,000 shuffles of the firing activity using the MATLAB circshift function (to preserve the temporal correlation of the firing activity signal) at random shifts. We next correlate each shuffled version with the stimulus signal of the corresponding neuron to obtain a distribution of correlation coefficients (3 separate distributions, 1 per mouse). We define $r_m^{95\text{th}}$ as the value that corresponds to the 95th percentiles of the distribution, for mouse $m$. If $r_{\text{thresh}} > r_m^{95\text{th}}$, we keep iterating the same procedure while decreasing $r_{\text{thresh}}$ by 0.01 until convergence (that is, $r_{\text{thresh}} \simeq r_m^{95\text{th}}$), which constitutes the correlation coefficient threshold to identify HD neurons for mouse $m$ (see Extended Data Fig. 1d,e for an illustration of the results).

## HD decoding from neural data

We trained a recently developed Bayesian decoder[37] to infer the HD direction from the deconvolved calcium responses of the imaged neural population. Noise independence across neurons was assumed. Conceptually, this decoder is similar to the Bayesian decoding method for spike trains as commonly used in the literature[47], except that we used zero-inflated-gamma distribution to model the stochasticity of the deconvolved calcium responses, instead of Poisson distribution. Our previous results showed that the zero-inflated-gamma model could better capture the noise of the calcium signal and provide better decoding results compared with the Poisson noise model and a few other alternatives. Details of this procedure can be found in section 4 of ref. [37]. Here we smoothed the log-likelihood matrix (rows, angle bins; columns, frames) by iteratively summing the likelihoods over 5 frames (~166.7 ms) centred around the corresponding timestep of each iteration, for each angle bin. Note that, owing to the predominance of HD-tuned neurons among detected cell segments and to avoid selection biases, the neural activity from all ADN cells was used as an input to the decoding algorithm.

## Analysis of drift

We define the offset as the angular difference between the measured head direction ($\theta_{\text{measured}}$) and the decoded head direction ($\theta_{\text{decoded}}$):

$$\text{Offset}(t) = \text{angdiff}(\theta_{\text{decoded}}, \theta_{\text{measured}})$$

In all analyses involving drift estimation, both measured and decoded HDs were smoothed with a moving average filter of width 20 frames (~667 ms). For the analysis of drift during darkness (except for heat maps), further smoothing was applied to extract the low-frequency component of the signal whereby a moving average filter of width 300 frames (~10 s) was used. In all cases, a simple linear regression was performed on the unwrapped offset signal over a sliding window of 20 frames (~667 ms) to estimate the drift speed at the centre of the regression window (that is, slope of the fitted line).

## Separation of fast and slow resets

Classification of resets within the [70:110]° range was done using the $k$-means clustering function in MATLAB. We used data from the first 1,450 frames after cue display. The algorithm separates between two clusters by generating 50 replicates with different initial cluster centroid positions for each replicate and then calculating the sums of point-to-centroid distances for each cluster using the 'city block' distance metric.

## Reconstruction of the bump of activity

At any given time, we can reconstruct the bump of activity from the firing activity of each neuron and their respective tuning curves using a normalized weighted sum of tuning curves[48]:

$$A(\theta, t) = \frac{\sum_i f_i(\theta) r_i(t)}{\sum_i f_i(\theta)},$$

where, $A$ is a 360-by-$T$ matrix (each row is a 1° bin of the horizontal plane and each column is a frame within range $T$ of the analysis), $f_i$ is the tuning curve of neuron $i$ and $r_i$ is the firing activity of neuron $i$.

## Calculation of network gain

We assume that, at any given time, the thalamic HD network is subject to a global gain modulation of the firing activity, applied homogeneously on all ADN neurons such that:

$$r_{i,t} = \alpha_t f_i(\theta_t) + \varepsilon, \quad \varepsilon \sim \mathcal{N}(0, \sigma^2),$$

where $r_{i,t}$ is the instantaneous firing activity of ADN neuron $i$; $\alpha_t$ is the instantaneous gain factor; $f_i$ is the tuning curve of ADN neuron $i$ (calculated on the basis of the response measured in the baseline condition); $\theta_t$ is the decoded head direction from neural activity at time $t$; and $\varepsilon$ is the additive Gaussian noise.

Our goal is to estimate the value of $\alpha_t$ at any given time $t$ using maximum-likelihood estimation approach.

Given the decoded head direction at time $t$, $\theta_t$ as well as the tuning curves $f_i$ for all ADN neurons, we obtain the likelihood of observing $r_{i,t}$ with parameter $\alpha_t$:

$$P(r_{i,t}|f_i(\theta_t); \alpha_t) = \mathcal{N}(\alpha_t f_i(\theta_t), \sigma^2).$$

We define the vectors:

$$R_t = \begin{bmatrix} r_{1,t} \\ r_{2,t} \\ \vdots \\ r_{N,t} \end{bmatrix}, \quad F(\theta_t) = \begin{bmatrix} f_1(\theta_t) \\ f_2(\theta_t) \\ \vdots \\ f_N(\theta_t) \end{bmatrix},$$

where $N$ is the number of ADN neurons in the network.

Assuming independent activity between said neurons, we can calculate the likelihood of observing $R_t$:

$$P(R_t|F(\theta_t); \alpha_t) = \prod_i P(r_{i,t}|f_i(\theta_t); \alpha_t)$$
$$\propto \prod_i \exp\left(\frac{(r_{i,t} - \alpha_t f_i(\theta_t))^2}{-2\sigma^2}\right).$$

We apply the logarithm on both sides:

$$\log(P(R_t|F(\theta_t); \alpha_t)) \propto -\frac{1}{2} \sum_i \frac{(r_{i,t} - \alpha_t f_i(\theta_t))^2}{\sigma^2}.$$

Our goal is to determine the parameter $\hat{\alpha}_t$ that maximizes the log-likelihood such that:

$$\hat{\alpha}_t = \underset{\alpha_t}{\text{argmax}}\ \log((P(R_t|F(\theta_t); \alpha_t)) = \underset{\alpha_t}{\text{argmin}} \sum_i (r_{i,t} - \alpha_t f_i(\theta_t))^2.$$

To do so, we take the derivative of the objective function with reference to $\alpha_t$ and set it to zero:

$$\frac{d}{d\alpha_t} \sum_i (r_{i,t} - \alpha_t f_i(\theta_t))^2 = 0.$$

Thus:

$$\hat{\alpha}_t = \frac{\sum_i r_{i,t} f_i(\theta_t)}{\sum_i f_i(\theta_t)^2}.$$

Similar to the offset signal, the obtained gain is smoothed with a moving-average filter of width 20 frames (~667 ms), unless otherwise

specified. With the exception of Extended Data Fig. 5b–d, the gain was always normalized to the average baseline activity. In Extended Data Fig. 5b–d, the tuning curves represent the average firing rates as a function of the internal HD for the entire recording session.

### Gain heat-map analysis

Gain heat maps are 2D matrices in which each pixel $p(x, y)$ is a 2D bin of width 1.5° per s corresponding to the measured angular head velocity and height 1° corresponding to the decoded HD. Pixel $p(x, y)$ represents the mean network gain—across mice and across sessions—within a 2D average window of width $[x − 3:x + 3]°$ per s and height $[y − 15:y + 15]°$. A 2D Gaussian filter of s.d. = 15 (15° × 22.5° per s) is then applied. The network gain, the decoded HD and the measured HD were all smoothed with a moving-average filter of width 20 frames (~667 ms) while the measured head angular velocity was approximated by a simple linear regression with a regression window of similar width. To evaluate the significance of the difference between gain heat maps (Fig. 4f), we performed a Wilcoxon rank-sum test to compare, at each pixel, the gain distributions within the 2D window of width $[x − 3:x + 3]°$ per s and height $[y − 15:y + 15]°$ between darkness epochs of the 20 s experiment and D2 of the 2 min experiment. As we are only interested in the significance of the positive values (indicating the appearance of new bumps), negative values as well as $P > 0.001$ were marked as NaN.

### Drift-speed heat-map analysis

Drift-speed heat maps were generated according to the same approach as for the gain heat maps. However, drift speed was approximated by a simple linear regression with a regression window of width 20 frames (~667 ms). The $P$-value matrix for drift speed difference between the 20 s experiment and D2 of the 2 min experiment (Extended Data Fig. 13e) was calculated as described above. However, only $P$ values $> 0.001$ were marked as NaN.

### Vector field analysis

The purpose of this analysis is to illustrate baseline attractiveness. We define the state space ($y$ axis, drift-speed (° per s); $x$ axis, drift-angle (°)). We construct a vector field matrix by dividing the $x$ axis into 18 bins of width 20° each within the range $[−180:180]°$, and the $y$ axis into 20 bins of width 0.03° per s each, within the range $[−3:3]°$ per s. At each bin $(x,y)$, we calculate the mean drift speed and mean drift acceleration across mice and across sessions. The two latter quantities represent the velocity components $(u,v)$ that determine the length and direction of the velocity vector. We assume the vector field has a central symmetry with reference to the baseline point (0,0) owing to the symmetry in the experimental design. We therefore generate an image of the original vector field that is its reflection across the origin. The two versions are then averaged to produce the final 2D vector field. Streamlines are generated using the streamline function in MATLAB. For Figs. 4d and 5e, streamlines were simulated over 1,000 timesteps.

### Analysis of cue-rotation sessions

At the beginning of each continuous cue-rotation epoch, the visual cue was displayed at the same location as in the baseline. After cue removal and depending on its previous rotation speed, the cue would have either reached ±180° or ±90° (cue orientation in clockwise-cue-rotation sessions was reflected across the $x$ axis so that the cue ends at ±180° (fast cue-rotation) or −90° (slow cue-rotation)). The offset is therefore expected to start within a close range of these two directions during the second darkness epoch. However, in some cases, drifts during the first darkness epoch were large enough so that the initial anchoring to the rotating cue occurred considerably far from baseline. This caused the drift signal during the second darkness to start further away from the expected location. In Fig. 5d, we limited our analysis to drifts starting within $[−180:−145]U[145:180]°$ for fast cue rotation and $[−125:−55]°$ for slow cue rotation, to study the effects across sessions with similar stability during baseline (total $n = 44$ out of 60).

### Dimensionality reduction

It is generally believed that the main function of the HD system is to provide an estimate of the HD at any given time. As most studies of this network, including ours, are conducted while recording the neural activity in animals placed on horizontal planes, it is fair to assume that most of the variability in the activity of HD neural population can be captured by a single variable representing the angle faced by the animal, at an instant $t$, with reference to a given allocentric reference frame. Indeed, previous studies have shown that, in stable conditions, different dimensionality reduction methods[17,38] would produce a circular manifold that can be fairly approximated in a unidimensional polar state-space with a fixed radius. Nevertheless, a previous study[17] observed that the structure becomes more complex during slow-wave sleep. Our guiding hypothesis is that the intrinsic geometric structure of the neural activity in the HD network lies in a multidimensional state space and that latent variables other than the angular component are needed to explain the variability in spiking data, during non-stable conditions such as resets and drift situations. Here we propose the simplest augmentation to the latent structure by adding a radial component that we expect to indicate instantaneous changes in global firing activity of the HD network. Although we believe that the true intrinsic dimensionality of the HD neural data is higher than two, the current paper mainly focuses on the necessity of at least a second dimension of the HD system during instability.

To test our hypothesis, we developed a deep feedforward neural network that maps the high-dimensional input (neural) data onto the 2D polar space (angular dimension $\theta$ and radial dimension $R$). The network is trained on circular data from the measured head direction. The radial component $R$ is a latent variable that can take any non-negative value. Our previous analyses (not included here) have shown that, although methods such as principal component analysis and Isomap can uncover looped latent structures, these unsupervised learning algorithms tend to produce distorted circles, in the presence of noise, when applied on baseline data (that is, stable condition), which makes the definition of a radius less straightforward and motivates our use of a supervised learning method.

We used a feedforward neural network with three parallel branches. Two of these branches have three fully connected hidden layers (referred to as first and second or $B_1$ and $B_2$, respectively), while the third branch has two fully connected hidden layers (referred to as middle or $B_m$) (Extended Data Fig. 4d). The input layer receives a $N × 1$ vector of neural activity from $N$ ADN neurons at time $t$ (both calcium traces as well as firing activity from deconvolved spikes can be fed to the model). The output layer is composed of two units that are the results of multiplying the output $g_t$ of the middle branch with the output $z_{1,t}$ of the first branch, on one hand, and the output $z_{2,t}$ of the second branch, on the other hand, as illustrated in the diagram of Extended Data Fig. 4d.

We trained our model on baseline data. The objective is to find the set of weights $W$ that minimize the distance between the network output $\begin{pmatrix} g_t z_{1,t} \\ g_t z_{2,t} \end{pmatrix}$ and the vector $\begin{pmatrix} \cos(\theta_t) \\ \sin(\theta_t) \end{pmatrix}$, where $\theta_t$ is the measured head direction of the animal at instant $t$. We define the loss function as the mean squared error:

$$\text{MSE} = \frac{1}{T} \sum_{t=1}^{T} \left\| \begin{pmatrix} \cos(\theta_t) \\ \sin(\theta_t) \end{pmatrix} - \begin{pmatrix} g_t z_{1,t} \\ g_t z_{2,t} \end{pmatrix} \right\|_2^2,$$

where, $T$ is the duration of the training epoch and $\|.\|_2$ is the $L_2$ norm. If the algorithm converges, we obtain the following approximations:

$$
\begin{cases}
z_{1,t} \approx \dfrac{\cos(\theta_t)}{g_t} \\[2ex]
z_{2,t} \approx \dfrac{\sin(\theta_t)}{g_t}
\end{cases}.
$$

Let $\hat{R}_t = \frac{1}{g_t}$, then we can rewrite the output of each branch:

$$
\begin{cases}
B_1 : z_{1,t} \approx \hat{R}_t \cos(\theta_t) \\[1ex]
B_2 : z_{2,t} \approx \hat{R}_t \sin(\theta_t) \\[1ex]
B_{\mathrm{m}} : g_t = \dfrac{1}{\hat{R}_t}
\end{cases}.
$$

In effect, this would allow branches $B_1$ and $B_2$ to learn a mapping from the input (neural) space to the Cartesian transformation of the polar coordinates of a given state $s_t$, at any time $t$ (respectively, $B_1$ projects the input onto the $x$ axis and, $B_2$ projects the input onto the $y$ axis). From these two branches, we can extract the decoded angle $\hat{\theta}_t = \arctan\left(\frac{z_{2,t}}{z_{1,t}}\right)$. While branch $B_{\mathrm{m}}$ would learn a mapping from the input space to the inverse of the approximate radius $\hat{R}_t$ of said state, in polar space. If we assume that $\hat{R}_t$ is a certain reflection of global neural activity, as per our hypothesis, then we expect small fluctuations of population activity in the training data (baseline) to be sufficient to allow the network to extrapolate $\hat{R}_t$ on test data with larger fluctuations.

## Statistics and reproducibility

All statistical tests are noted where the corresponding results are reported throughout the main text and Supplementary Information. All tests were uncorrected two-tailed tests unless otherwise noted. Outliers were identified as data points that fall outside the mean $\pm$ (3 s.d.) range.

## Reporting summary

Further information on research design is available in the Nature Portfolio Reporting Summary linked to this article.

## Data availability

The complete dataset for all experiments is available at Figshare (https://doi.org/10.6084/m9.figshare.21792689). The dataset should not be used for republication without prior consent from the authors.

## Code availability

All source codes used in the current study are available on request to the corresponding authors.

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

**Acknowledgements** We thank S. Kim, H. C. Yong and A. Nieto-Posadas for the technical assistance and help gathering the histology data; L. Paninski, M. Hasselmo, S. Badrinarayanan, J. Ying, M. Yaghoubi and A. Peyrache for comments on previous versions of this paper; D. Aharoni for advice and assistance with the UCLA miniscope; and the members of the Brandon laboratory for discussions, in particular, M. Yaghoubi and R. R. Rozeske. This work was supported by funding from the Canadian Institutes of Health Research (project grants 367017 and 377074), the Natural Sciences and Engineering and Research Council of Canada (Discovery 74105) and the Canada Research Chairs Program to M.P.B.

**Author contributions** Z.A., A.T.K. and M.P.B. contributed to the experimental design. Z.A. performed all surgeries, recordings, data analysis and modelling. X.-X.W., A.T.K. and M.P.B. provided guidance for data analysis. Z.A. wrote the initial draft. All of the authors contributed to editing and revising the paper.

**Competing interests** The authors declare no competing interests.

**Additional information**
**Correspondence and requests for materials** should be addressed to Zaki Ajabi or Mark P. Brandon.

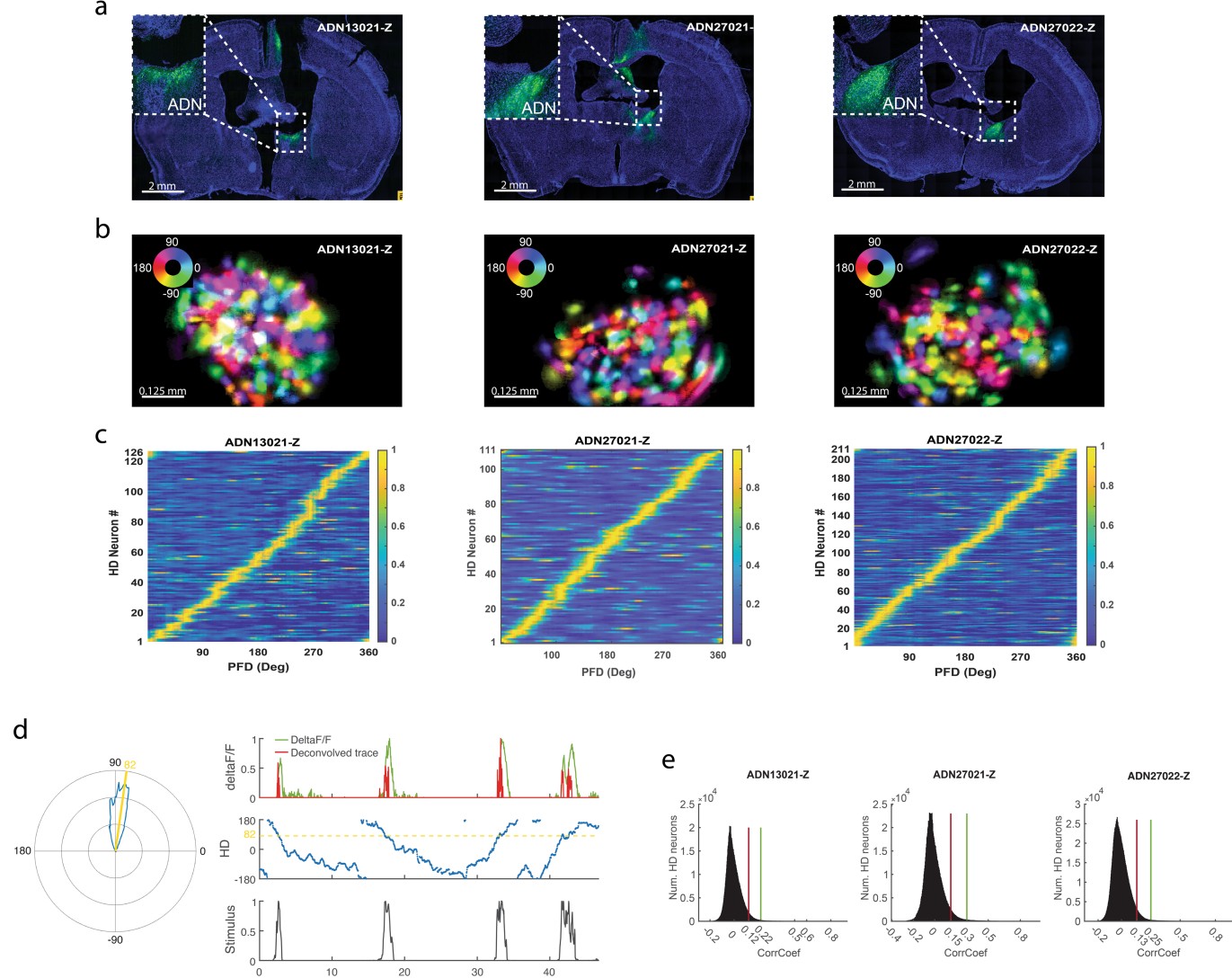

**Extended Data Fig. 1 | Calcium imaging in the anterodorsal thalamic nucleus (ADN) and identification of HD neurons.** a. Histology data showing coronal brain sections from each mouse with GCaMP6f expression, in ADN (anterior part). Mouse ID written in the top right and scale-bars shown in the bottom left of each panel. In total, 12 mice were injected and implanted for this study, only 3 (shown here) provided enough simultaneously recorded head-direction cells for continued experimentation. b. Directional maps of ADN in each mouse. HD cells are coloured by their preferred firing direction (PFD). Colour-wheel shows angle-colour assignments. Mouse ID written on the top right and scale-bars shown in the bottom left of each panel. c. Examples of HD cells' coverage of the azimuthal plane, in each mouse. Rows in each matrix represent tuning curve heatmaps of individual HD cells. The amplitudes of individual tuning curves are normalized. Mouse ID written above each panel. d. Left: An example polar tuning curve for a HD neuron. Yellow line: direction of maximum firing activity (that is, PFD). Firing activity is occupancy normalized. Right: Top-row: Example calcium signal deltaF/F (green) from one HD neuron and deconvolved trace (red). Both traces were normalized. Middle-row: Measured HD. Bottom-row: Extracted stimulus signal of the HD neuron's PFD. Peaks indicate instances of the animal facing the particular PFD. The deconvolved signal is cross-correlated with the stimulus signal in order to obtain the Pearson's correlation coefficient which reflects the cell's degree of HD tuning (r = 0.85 in the case of the current example). e. Distributions of correlation coefficients after 1000 circular-shift shuffles of the firing activity signals (smoothed deconvolved traces) of all HD neurons, in each mouse. Red and green vertical lines indicate 95th and 99th percentiles, respectively. Data includes 10 baseline recordings of 3 min each, for every mouse. Of all recorded cells, ~94% met the 95th percentile selection criterion while ~83% met the 99th percentile selection criterion.

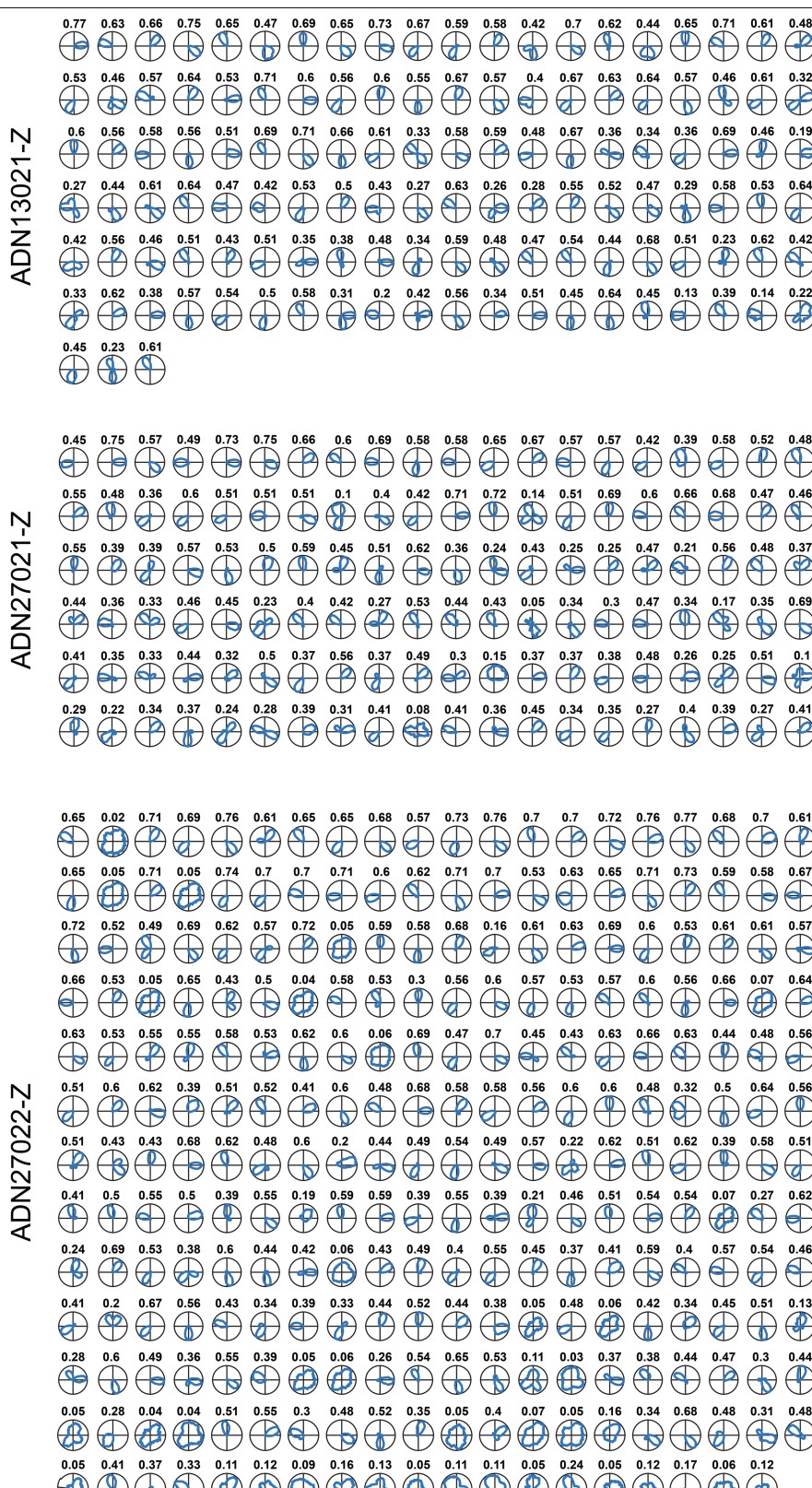

**Extended Data Fig. 2 | Polar tuning curves of ADN neurons from a 10-minute baseline recording for each mouse (total number of neurons = 502).**
The directional tuning of each ADN neuron is shown by the correlation coefficients above each tuning curve. Mouse ID written on the left side.

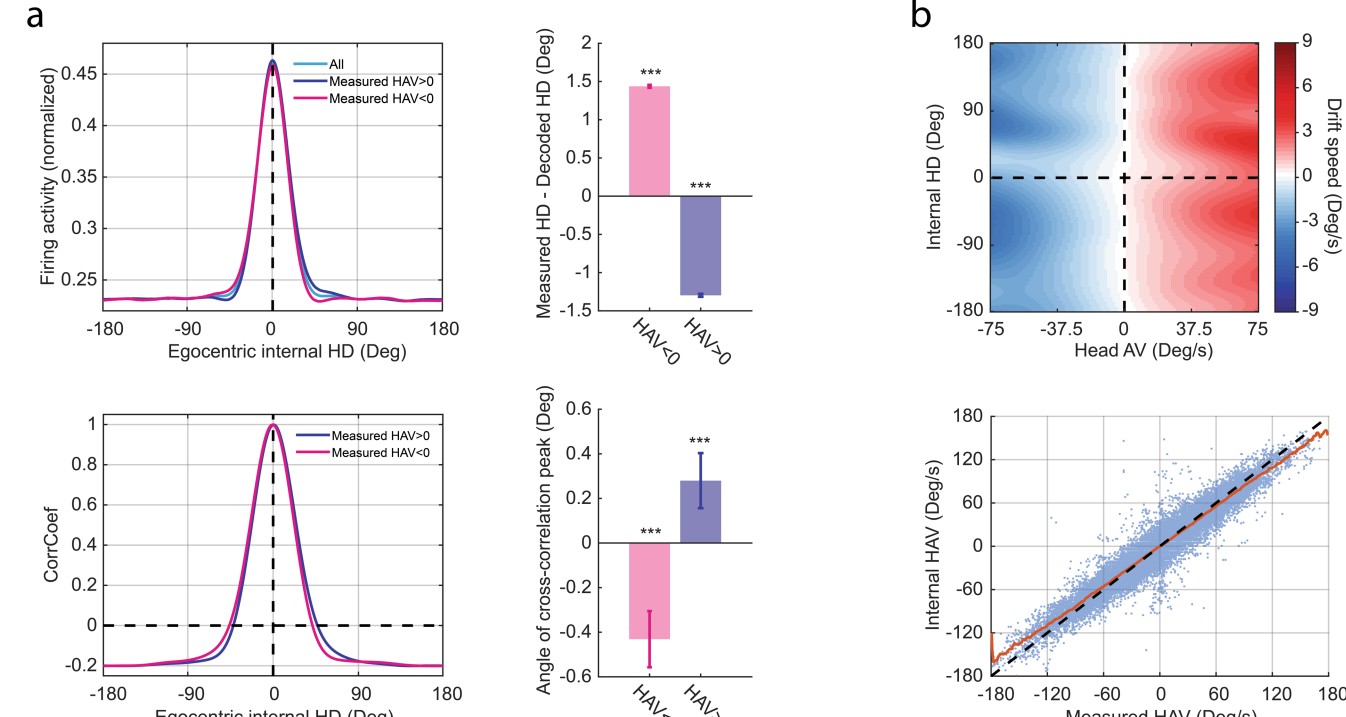

**Extended Data Fig. 3 | Anticipatory behaviour and drift-speed pattern during baseline.** a. Top row: Mean bump of activity divided between positive (blue) and negative (pink) head angular velocities (HAV). Bar graph: Mean difference between measured and decoded HD (n = 42x5000 data points from baseline recordings, between both groups; Two-sided Wilcoxon signed rank test: HAV < 0: p = 0, Z = 83.71; HAV > 0: p = 0, Z = −76.81). Bottom row: Mean cross-correlation of the mean bump of activity, per epoch, with the mean bump of activity for positive (blue) and negative (pink) HAVs. Bar graph: Mean peak angle of cross-correlation (n = 42x5000 data points from baseline recordings, between both groups; Two-sided Wilcoxon signed rank test: HAV < 0: p = 0, Z = −115.24; HAV > 0: p = 0, Z = 113.13). Both analyses show a significant amount of anticipation of future heading by the HD network. b. Top: Drift-speed heatmap showing an increased latency in updating the internal representation as the HAV becomes larger. Bottom: same pattern as the above, seen here in Internal HAV-versus-Measured HAV space. Notice the deviations of the mean signal (orange) from the diagonal, at high measured HAVs. Bar graphs indicate mean ± SEM.

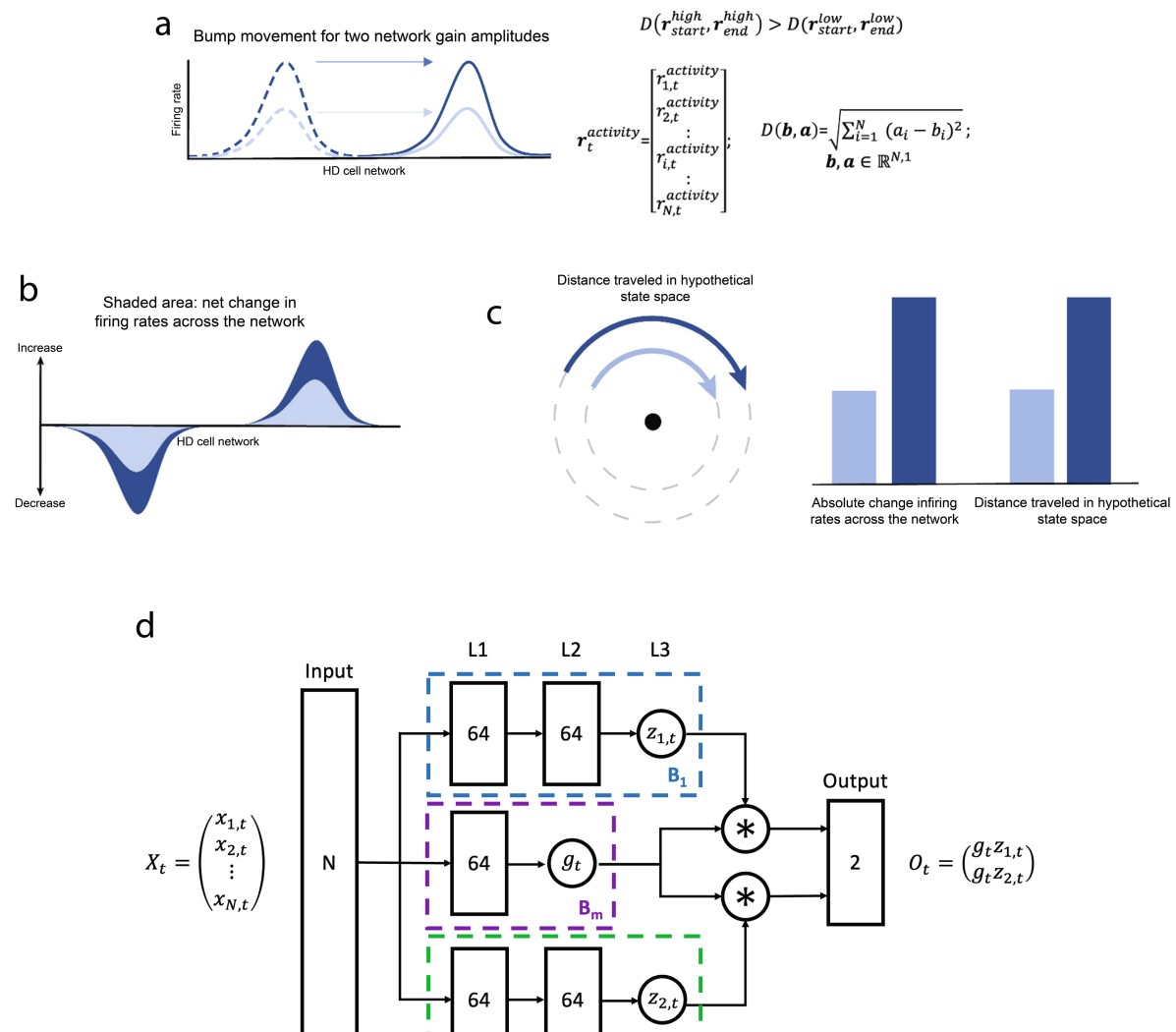

**Extended Data Fig. 4 | Hypothesized relationship between the population-activity and movements in the low dimensional polar state-space. a, b.** The amount of the change of neural activity during bump movement depends on the gain of the network. $x$ axis represents the neuronal space (assuming uniform distribution of HD cells by PFD). Mathematically, the distance between representations of internal HD from start to end of a rotation, in the Euclidean sense, is smaller at lower network gain. $D$: Euclidean distance, $r_t^{activity}$: Nx1 vector of firing rates from N HD neurons, at time $t$ for 'high' or 'low' activity levels.
**c.** The concept of decreasing distance between internal HD representations, at lower network gain, is naturally captured in the 2D polar plane if we assume that radius reflects the level of network activity. The distance travelled in the hypothetical state-space of the HD network is greater when the radius is larger as well as when the net gain is higher, which could be quantified by the total

change of firing rate across the network. Thus, we hypothesize that radius is correlated with overall population activity (that is, network gain) and that decreasing distance facilitates rotations across the HD network. Assuming that the internal HD representation lives in a 2D polar state-space where each state is defined by phase and radius, state transitions would be fastest at the lower end of the radial component because of the decreasing distance between states representing different angles, near the centre of the baseline ring. Bar-graphs are only indicative and not to scale. **d.** Diagram of the artificial neural network used to project high-dimensional neural activity onto 2D polar space. Numbers inside each box correspond to the unit count. All activation functions are 'relu' except for nodes $z_{1,t}$ and $z_{2,t}$ where the activation function is 'tanh'. In all layers, we apply $L_2$ regularization with regularization factor 0.001. Input data is normalized.

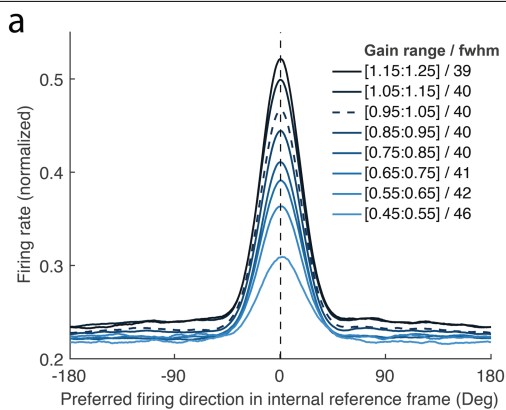

**Extended Data Fig. 5 | Relationship between network gain and population activity.** a. Reconstructed bump of activity (averaged over n = 42 sessions of the first experiment) for varying network gain ranges. Gain modulation not only affects the activity packet but also baseline activity. The decreasing baseline amplitude at low network gain indicates that the modulation is not driven by increased activity outside the main activity packet. Notice that the width of the activity packet remains within a narrow range. 'fwhm': the full width at half maximum in °. b. Method used to determine the variance explained by gain. Using the internal HD and neural activity from all recorded neurons per session as inputs ($S_{neuron\,i}$; 5 examples shown for illustration purposes), we can extract the tuning curve of each neuron (average firing activity as a function of internal HD, $f(\theta_t)$) as well as the gain signal ($g_t$), while assuming that pairwise coherence between HD cells is preserved. Two reconstructions of the neural activity are then produced from tuning curves and internal HD: In the first case (Dark-blue) neural activity is multiplied by gain ($R^g_{neuron\,i}$) while in the second case (Light-blue), gain is not taken into account ($R_{neuron\,i}$). The sum of variance across neurons is calculated for each group of neural activity (including ground-truth ($S_{neuron\,i}$)). c. Comparison of variance explained in percentage between the neural activity reconstruction with and without gain (sum of variance in each group is divided by the sum of variance in the ground-truth group) (n = 42 sessions; Two-sided Wilcoxon rank-sum test: p = 0.0245, Z = 2.2499). Error bars show mean ± SEM. d. Increase in variance explained when gain is applied to reconstructed neural activity relative to the case where gain is not applied (that is, ratio between % variance explained with and without gain, minus 1) (n = 42 sessions; Mean = 13.71%, s.d. = 5.14%). Error-bars show mean ± s.d. Dots represent individual datapoints.

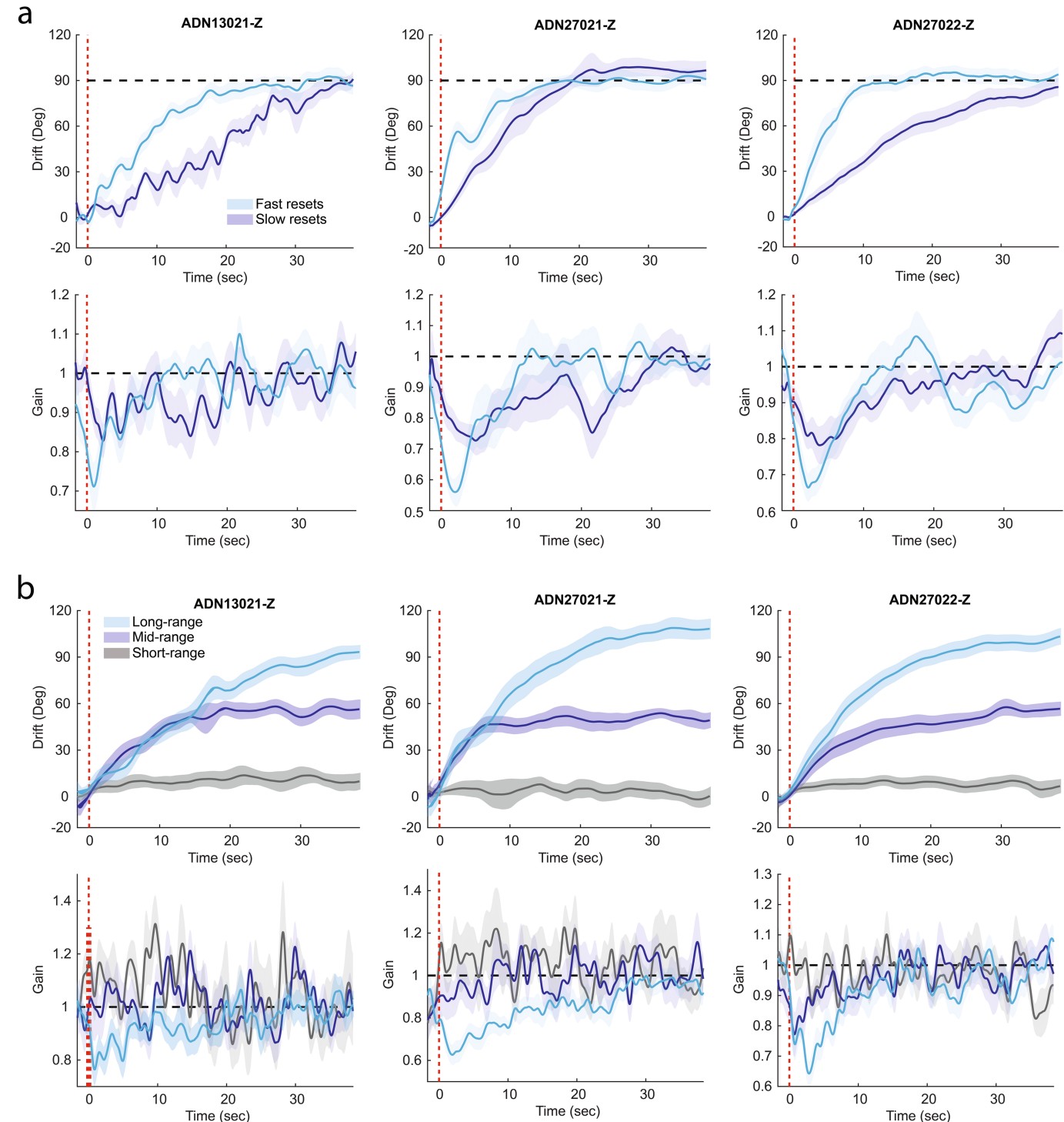

**Extended Data Fig. 6 | Reset behaviour and gain modulation across mice.**
**a**. Top row: Resets separated according to their speeds between fast (light blue) and slow (dark blue) groups. Mouse ID written above each panel. Bottom row: Corresponding gain signals for fast and slow resets. **b**. Top row: Resets separated according to their range between long- (light blue), mid- (dark blue) and short- (grey) groups. Mouse ID written above each panel. Bottom row: Corresponding gain signals for long-, mid- and short-range resets. Data are mean ± s.e.m.

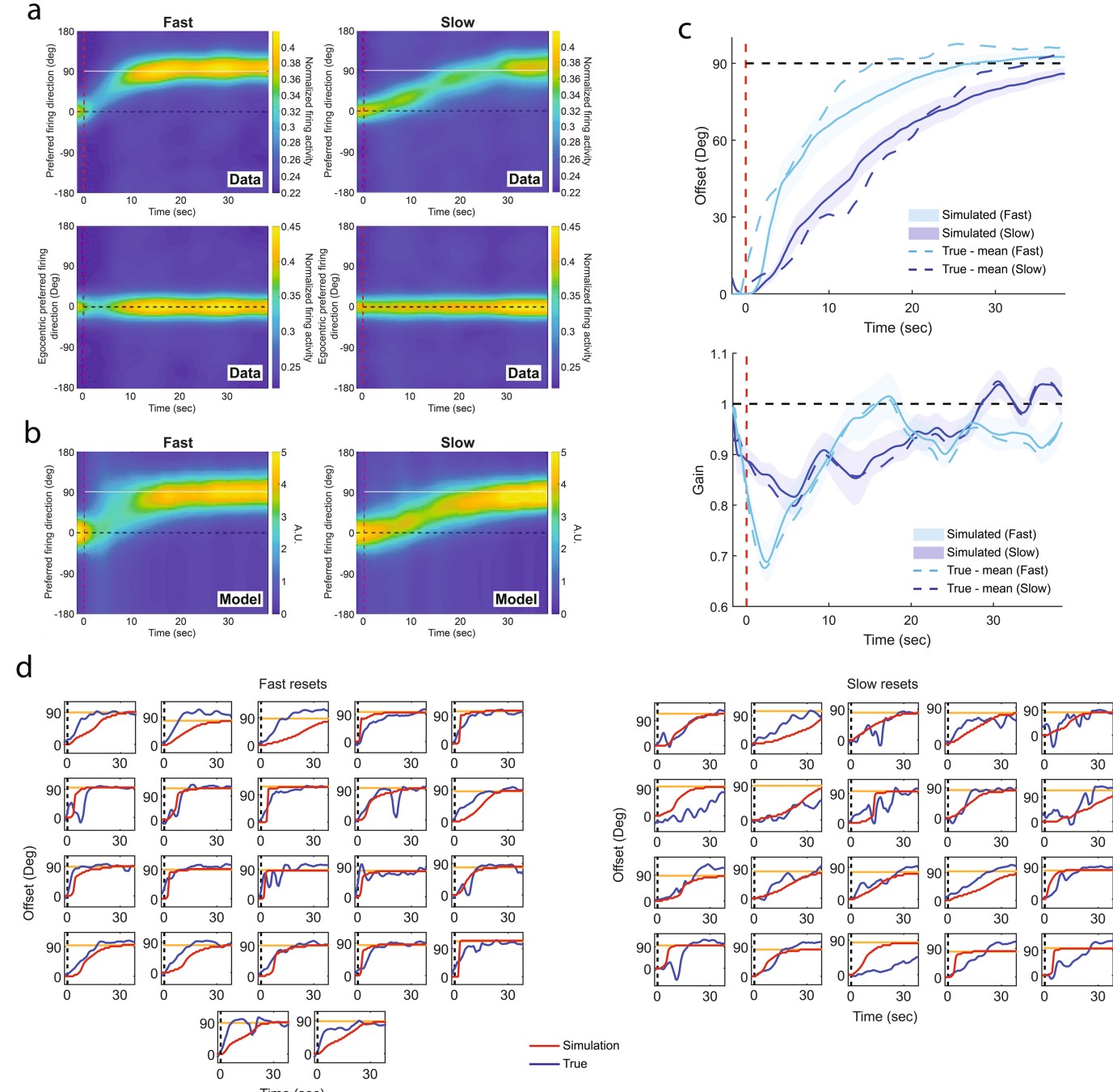

**Extended Data Fig. 7 | Agreement between true and model-predicted resets.**
a. Averaged heatmaps of the reconstructed bump of activity during fast (left column) and slow (right column) resets (same data as in Fig. 2g,h). Data is presented in the egocentric reference frame, without drift adjustment (top row) and with drift adjustment (bottom row) showing, in both cases, no additional bumps outside the main activity packet. Dashed red line indicates cue-onset, while white horizontal line at 90° is for reference. Firing activity is normalized. b. Simulation output of the gain-modulated attractor model taking input data (that is, gain) as in a. c. Top: Mean simulated reset signals for fast (light blue) and slow (dark blue) groups. Bottom: Mean simulated gain signals for the same groups. Data are mean ± s.e.m. Dashed signals represent means of ground-truth data. d. Individual examples of simulation predictions (red lines) for fast and slow reset groups, plotted against actual resets (blue lines). Yellow lines indicate cue location. Amplitudes are relative to angles at cue-onset (dashed black line).

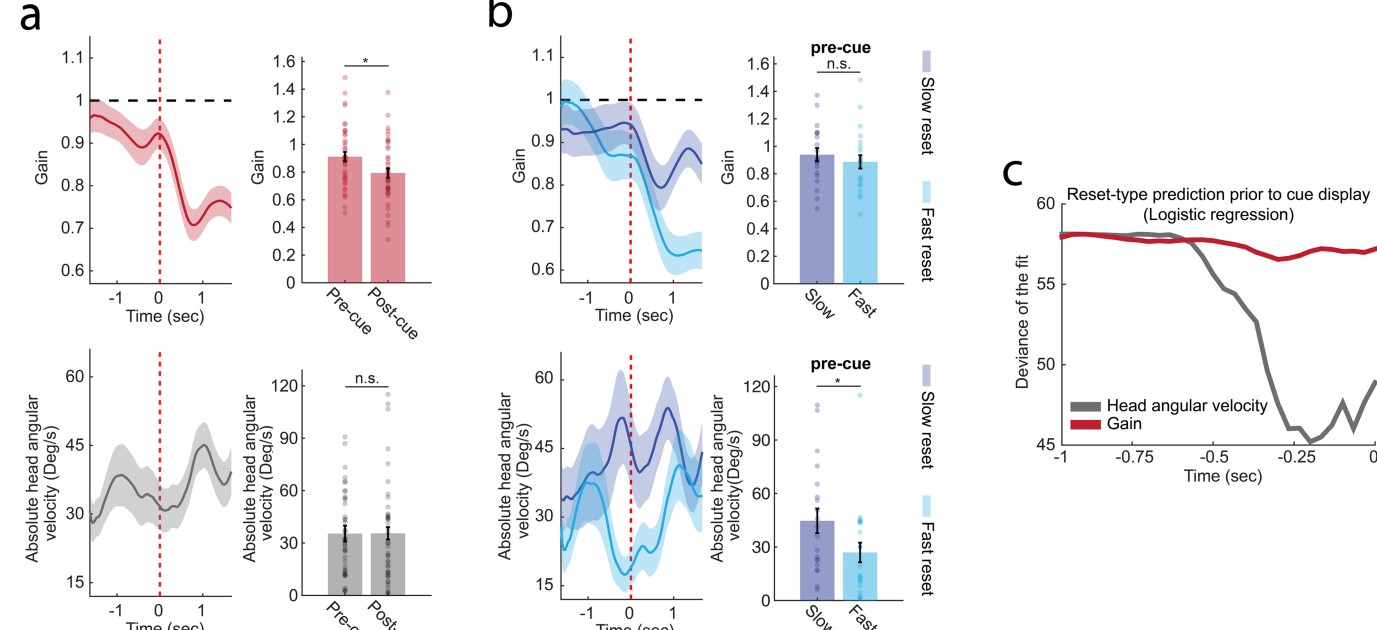

**Extended Data Fig. 8 | Animal behaviour, prior to cue display, is predictive of reset speed.** a. Triggered average of gain shows a sharp decrease after cue display (Two-sided Wilcoxon rank-sum test: average gain 1-second pre-cue versus average gain 1-second post-cue: p = 0.0228, Z = 2.28) (top). However, overall absolute head angular velocity (aHAV) does not seem to differ before and after cue display (Two-sided Wilcoxon rank-sum test: average aHAV 1-second pre-cue versus average aHAV 1-second post-cue: p = 0.6259, Z = 0.49) (bottom). Same reset events as in Fig. 2g,h (n = 42 events). b. Separation of signals in a. between fast (Light blue; n = 22 events) and slow (Dark blue; n = 20 events) resets shows similar gain amplitudes over a 1-second interval prior to cue display (Two-sided Wilcoxon rank-sum test: p = 0.3580, Z = 0.92) (top). However, aHAV is lower for fast resets compared with slow resets, over the same period (Two-sided Wilcoxon rank-sum test: p = 0.0294, Z = 2.18) (Bottom). c. Head angular velocity becomes more predictive of reset type closer to the moment of cue-display when compared with prediction performance based on gain amplitudes within the same time interval. Deviance of the fit is used as defined in Matlab's mnrfit function for logistic regression. Data shown is same as in Fig. 2g,h. Time dependent signals, in a and b, are shown as mean ± s.e.m. and bar-graphs show mean ± s.e.m. with individual datapoints.

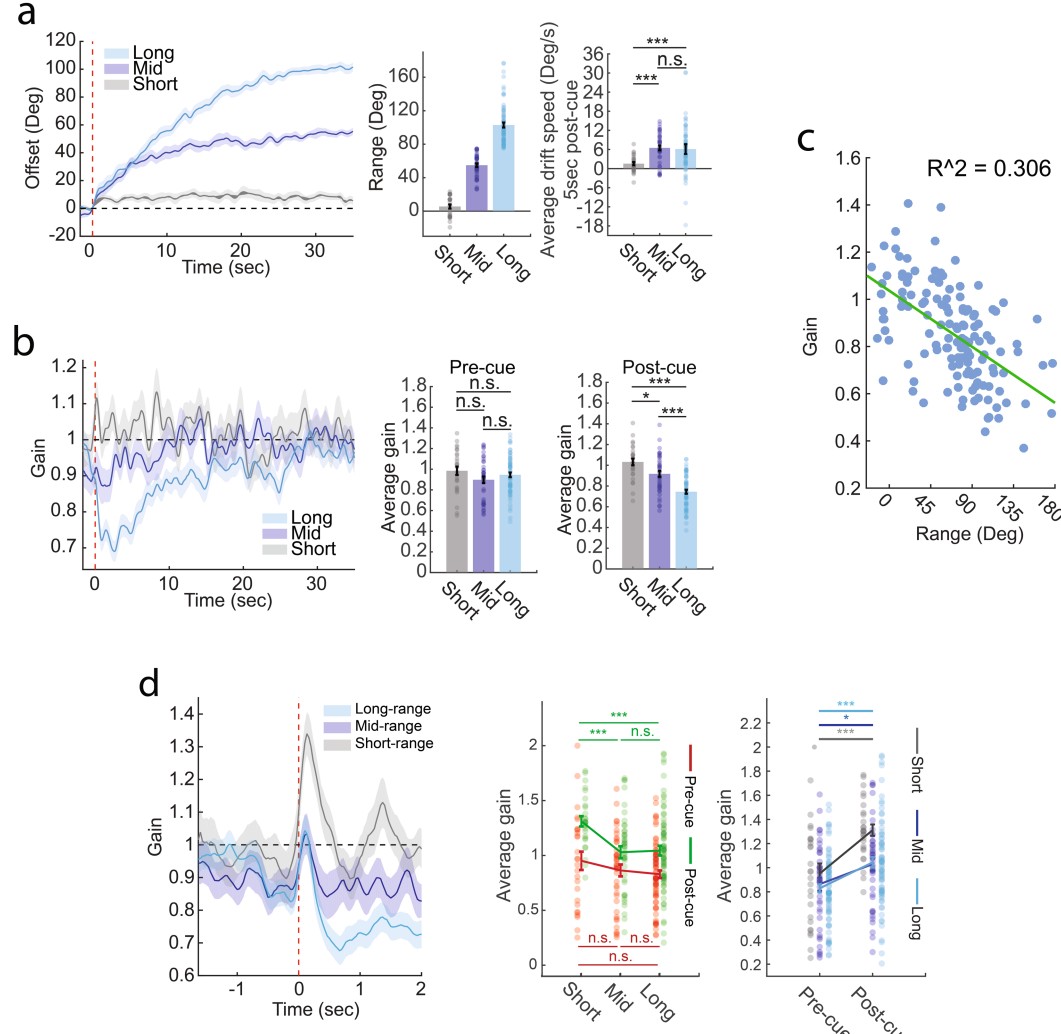

**Extended Data Fig. 9 | Relationship between reset range and gain modulation.** a. Mean drifts for short- (grey; n = 27 events), mid- (dark blue; n = 40 events) and long- (light blue; n = 67 events) range reset-groups showing non-significant difference in drift-speeds between mid- and long-range groups (Two-sided Wilcoxon rank-sum test: Short-Mid: p = 4.19e-5, Z = 4.10; Short-Long: p = 7.73e-5, Z = 3.95; Mid-Long: p = 0.62, Z = 0.50; 150 frames (-5 s) post-cue). b. Network gains for the short-, mid- and long- ranges have similar amplitudes prior to cue-display (Two-sided Wilcoxon rank-sum test: Short-Mid: p = 0.1174, Z = 1.57; Short-Long: p = 0.32, Z = 1.00; Mid-Long: p = 0.2984, Z = 1.04; 50 frames (-1.67 s) pre-cue), yet they exhibit gradual decrease after cue-display (Two-sided Wilcoxon rank-sum test: Short-Mid: p = 0.0129, Z = 2.49; Short-Long: p = 2.6876e-9, Z = 5.95; Mid-Long: p = 1.2130e-5, Z = 4.38; 150 frames (-5 s) post-cue). c. Relationship between average gain and reset range. Each dot represents a correct reset event (n = 134 events). The $R^2$ value corresponds to a

linear regression model fit (green line). All clockwise sessions have been reflected across the x-axis and transformed into counter-clockwise ones. d. Rapid gain spikes can be seen shortly after cue-display, in the three reset-range groups (Same data as in b, with higher temporal resolution). All reset ranges start at similar amplitudes at the end of the darkness period (Two-sided Wilcoxon rank-sum test: short-mid: p = 0.3940, Z = 0.85; short-long: p = 0.2090, Z = 1.26; mid-long: p = 0.4686, Z = 0.72). Following cue-display, each group exhibits a brief gain increase (5 frames (-150 ms) pre-cue vs 5 frames (-150 ms) post-cue: Two-sided Wilcoxon rank-sum test: short: p = 6.9690e-4, Z = 3.39; mid: p = 0.0369, Z = 2.09; long: p = 2.6898e-4, Z = 3.64). These gain spikes are largest for the short-range group (Two-sided Wilcoxon rank-sum test: short-mid: p = 4.4888e-4, Z = 3.51; short-long: p = 1.8600e-4, Z = 3.74; mid-long: p = 0.9326, Z = 0.08). Time-dependent signals are shown as data are mean ± s.e.m. and bar-graphs show mean ± s.e.m. with individual datapoints.

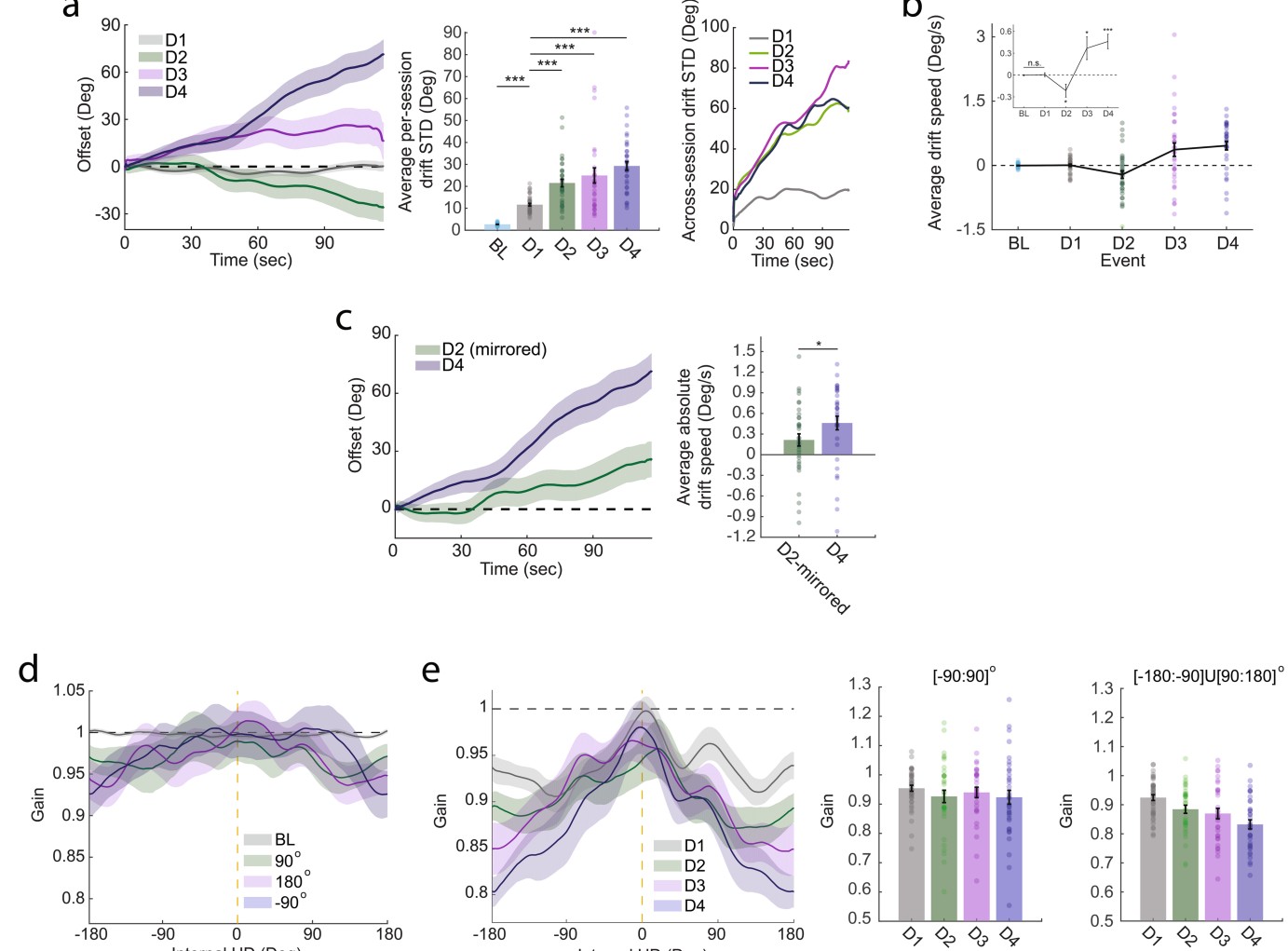

**Extended Data Fig. 10 | Distinct drift and gain patterns across darkness periods.** a. Drift variability increases significantly following a reset (D2, D3 and D4) in comparison with D1 (Mean drift s.d. compared across darkness epochs: Two-sided Wilcoxon rank-sum test: BL-D1: p = 3.1214e-15, Z = 7.89; D1-D2: p = 1.1477e-6, Z = 4.86; D1-D3: p = 8.3761e-5, Z = 3.93; D1-D4: p = 5.6600e-11, Z = 6.55). Drift s.d. also increases with time after a reset (D2, D3 and D4) while it remains constant following baseline (D1). (Number of epochs: D1: n = 42; D2: n = 35; D3: n = 32; D4: n = 35). b. Mean drift-speed in each darkness epoch shows systematic biases that depend on prior cue-event. (Two-sided Wilcoxon rank-sum test: BL-D1: p = 0.1250, Z = 1.53; Two-sided Wilcoxon signed rank test: D2: p = 0.0168, Z = −2.39; D3: p = 0.0313, Z = 2.15; D4: p = 2.9929e-4, Z = 3.62). (Number of epochs: D1: n = 42; D2: n = 35; D3: n = 33; D4: n = 34). c. Comparison between drifts in D2 and D4 of the 90°-cue-shift experiment. Although the two

events are experimentally symmetric to each other with reference to baseline, drifts in D4 appear to have larger biases (in absolute value terms) than D2. Left: Mean drift signals, in D2 (green) and D4 (dark-blue). Drifts in D2 have been mirrored across the 0°-line for comparison purposes. Right: Comparison between average drift speeds, in D2-mirrored (green; n = 35 epochs) and D4 (dark-blue; n = 34 epochs) (Two-sided Wilcoxon rank-sum test: p = 0.0184, Z = 2.36). d. Average gain tuning curves across light conditions. e. Average gain tuning curves across darkness conditions show a gradual decrease of the network gain away from the internal cue location (dashed yellow line) from D1 to D4 (Number of epochs: D1: n = 42; D2: n = 35; D3: n = 33; D4: n = 35). Time-dependent signals and gain tuning curves are shown as mean ± s.e.m. bar-graphs show mean ± s.e.m. with individual datapoints.

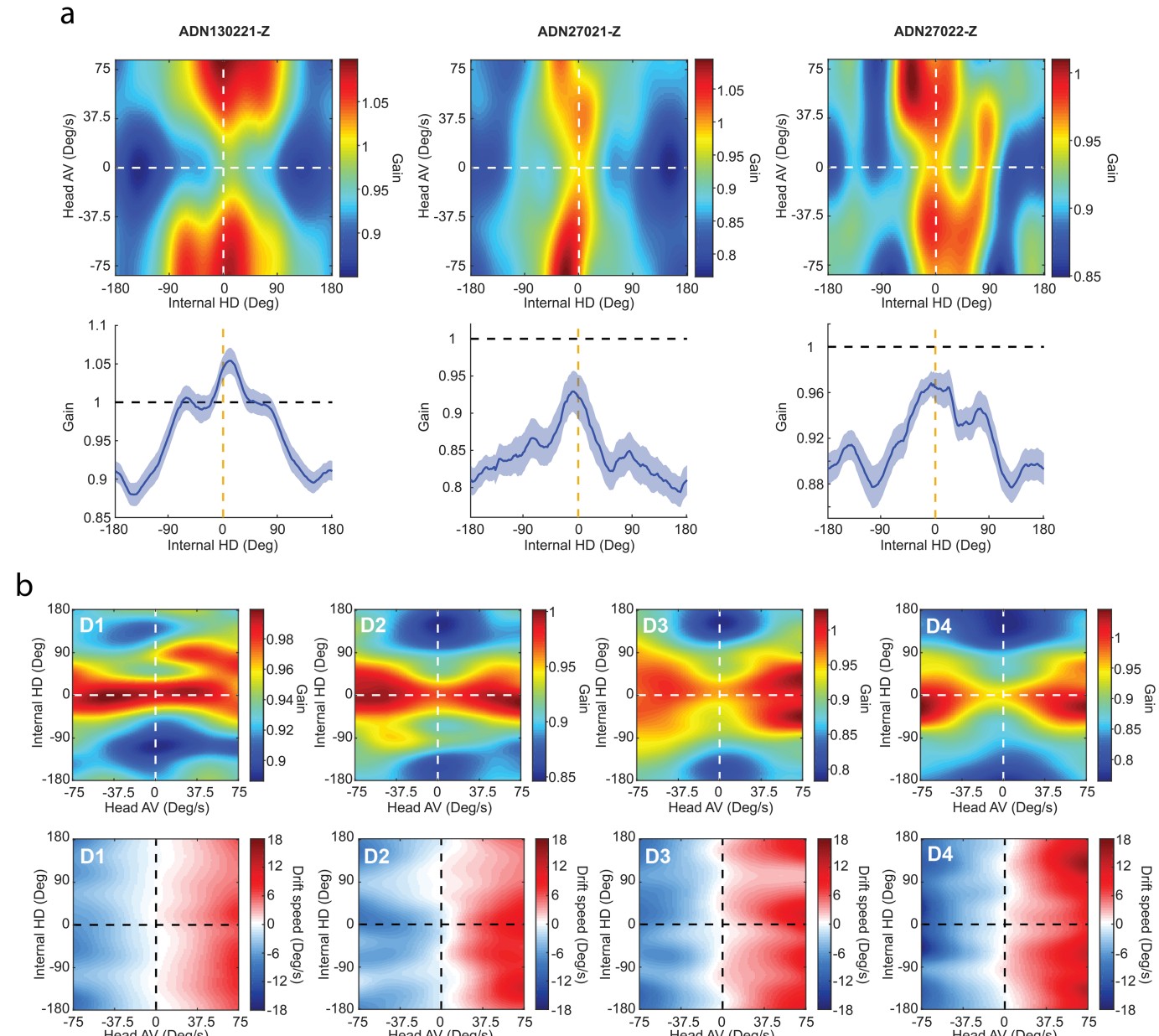

**Extended Data Fig. 11 | Network gain patterns across mice and darkness epochs.** a. Network gain during darkness shown as heatmaps (top row) and tuning curves (bottom row), per mouse. In both cases, data is averaged across sessions and darkness epochs (D1 to D4) of the 90°-cue-shift experiment. Values for the tuning curves are shown as mean ± s.e.m. Mouse ID written above each panel. b. Top row: Network gain heatmaps showing same data as in a, split (from left to right, respectively) across the different darkness epochs D1 to D4 of the 90°-cue-shift experiment. Bottom row: Drift speed heatmaps showing a consistent pattern, yet with varying amplitudes, across darkness epochs D1 to D4. No obvious effect of the gain landscape can be seen in these patterns and gain fluctuations did not correlate with any measurable distortion to the drift-speed landscape within the Head AV-vs-Internal HD state-space which maintained similar patterns to baseline (Extended Data Fig. 3b). This observation draws a clear distinction from the rapid representational shifts seen during resets and may point to a completely different mechanism linking network gain and drifts in dark conditions.

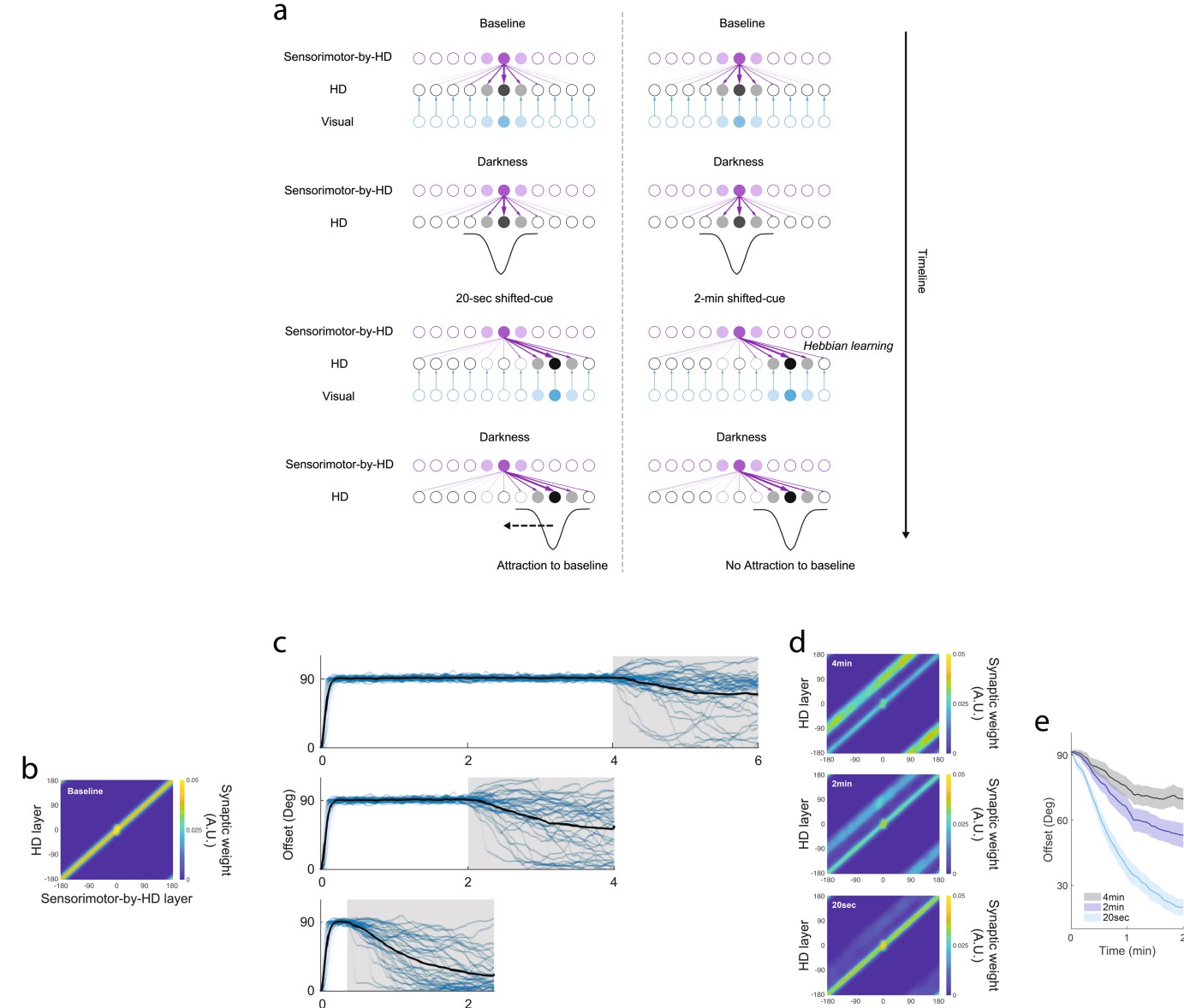

**Extended Data Fig. 12 | Model of the hypothesized role of plasticity in baseline attraction during darkness.** a. To explain drift dynamics that we observed in darkness, we propose a model that incorporates a 'sensorimotor-by-HD' layer that represents a cortical consensus about the directional sensory experience. Each neuron in this layer synapses onto all HD neurons via plastic synapses. Depending on the duration of exposure to the shifted cue context, the network either has enough time (that is, 2 min case) to form new associations between neurons of the HD and sensorimotor-by-HD layers which results in the emergence of a new steady state and no reversion to baseline, or not enough time (that is, 20 s case) and so, baseline associations between the two layers are maintained which causes the internal HD representation to revert to baseline

state. b, c, d, and e. Model simulations. b. Synaptic weight matrix linking the HD layer to the Sensorimotor-by-HD layer, during baseline. c. Simulations of representational drifts in 20 s (bottom row), 2 min (middle row) and 4 min (top row) exposures to the reset context. Behaviour for individual examples (that is, head angular velocity) is shared across scenarios and is taken from actual recordings. d. Synaptic weight matrices in darkness for the three scenarios showing the strengthening of new associations between HD and Sensorimotor-by-HD layers while baseline connections become weaker with increased duration of exposure to the reset context. e. Mean drifts (solid lines) in darkness across scenarios shaded areas indicate SEM.

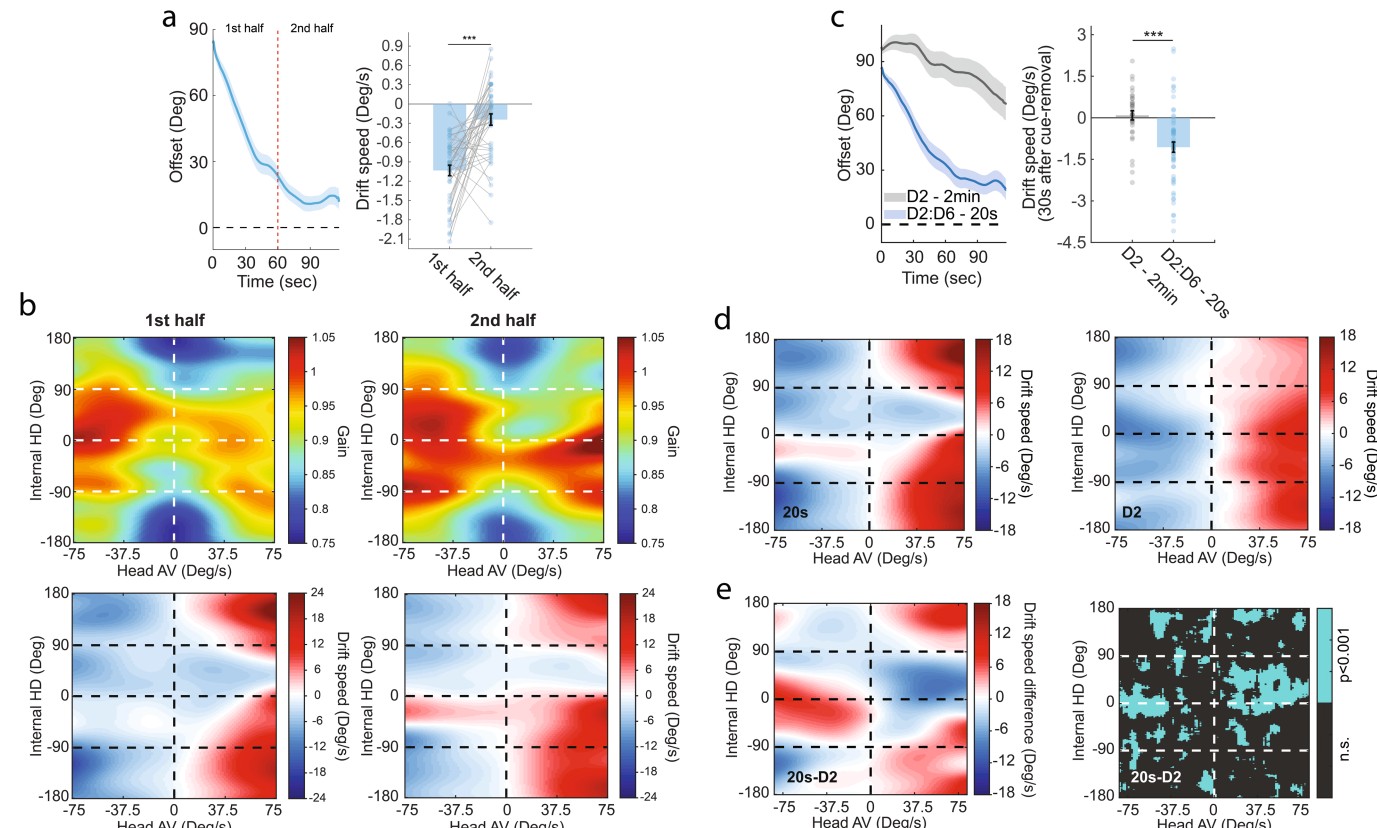

**Extended Data Fig. 13 | Drift and gain patterns during reversion.** a. Mean drift signal during reversion (n = 43 epochs). Dashed yellow line divides the darkness period in two halves with contrasting states of the HD network: drifting (1st half) and stabilizing (2nd half). Data shown as mean ± s.e.m. Bar graph: Comparison of mean drift-speeds between the first and second halves of the darkness period (Two-sided Wilcoxon rank-sum test: p = 3,5802e-8, Z = 5.51). Data are mean ± s.e.m. with individual datapoints. b. Top row: Heatmaps of network gain during the first (left) and second (right) halves. Bottom row: Heatmaps of drift-speed during the first (left) and second (right) halves showing state-dependent distortions of the drift-speed pattern. No obvious relationship between drift speed and network gain landscapes could be determined, unlike what we observed during reset events, indicating that the relationship between gain and network state updating depends on the

particular external input and/or current regime of the network. c, d, and e. Comparison of drift patterns between darkness epochs of the 20 s cue-exposure experiment and D2 of the 2 min cue-exposure experiment. c. Same as main Fig. 4c. d. Drift-speed heatmaps. Left: 20 s cue-exposure experiment (n = 43 epochs). Right: D2 of the 2 min cue-exposure experiment (n = 35 epochs). e. Left: Drift-speed difference (same data as in d) showing a significant distortion of the pattern seen in the first experiment around the internal location of the cue. Right: p-value matrix for data in left (Wilcoxon rank-sum test; pixels where p > 0.001 were marked as NaN). In addition to the network gain, the drift pattern also shows systematic differences as a function of angular velocity and internal head direction between D2 and the darkness following 20 s visual-cue display.

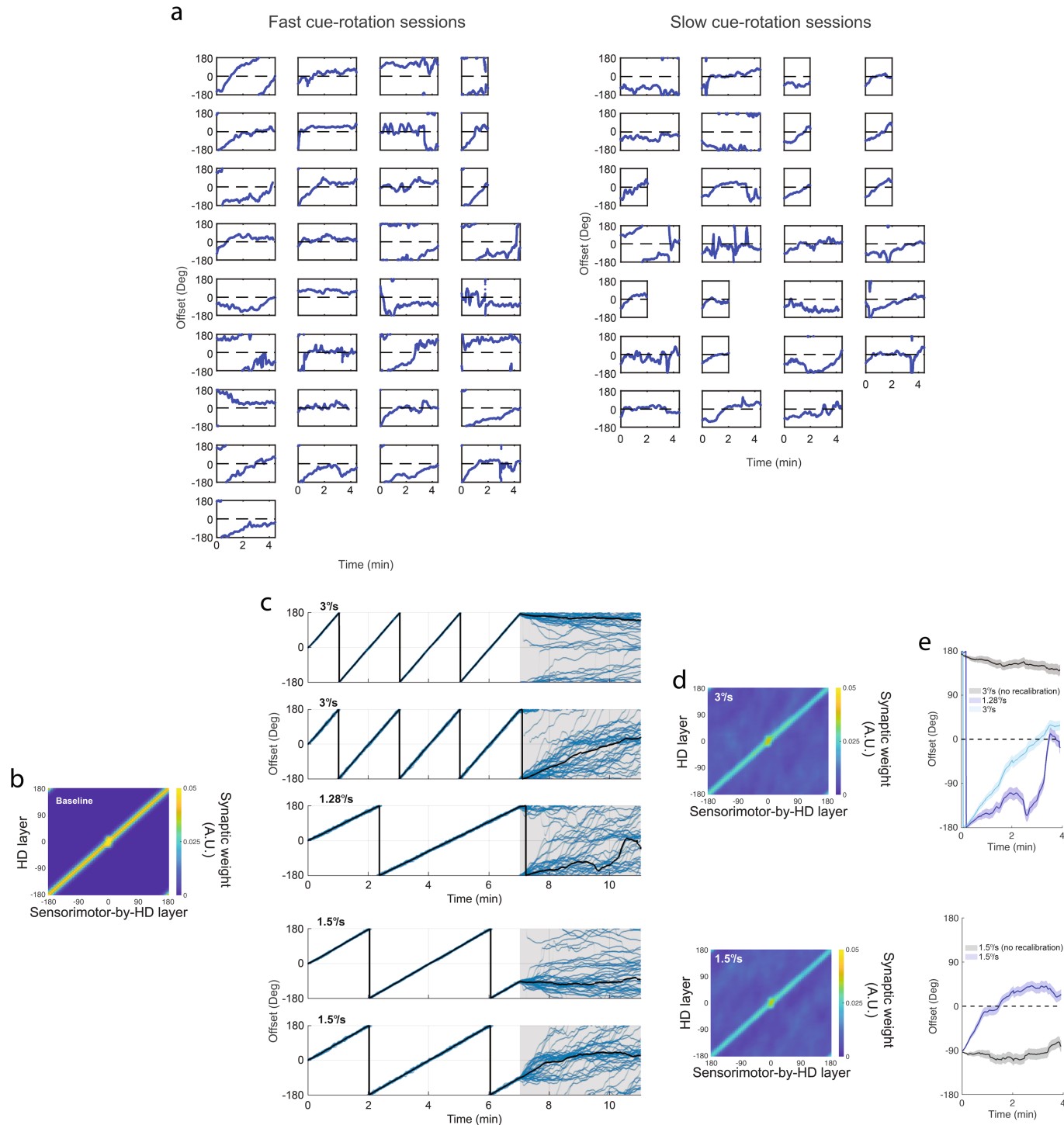

**Extended Data Fig. 14 | Persistent drift biases in darkness after cue rotation, between actual data and model-based simulations.** a. Recorded examples of drift biases for continuous fast (left group) and slow (right group) cue-rotation. b, c, d and e. Model-simulation of vestibular input recalibration by visual experience. b. Synaptic weight matrix linking the HD layer to the Sensorimotor-by-HD layer (see model in Extended Data Fig. 12), during baseline. c. Simulations of offset during cue-rotation and in subsequent darkness for the fast (3°/s) and slow (1.5°/s; 1.28°/s) cases. Behaviour for individual examples (i.e. head angular velocity) is shared across scenarios and is taken from actual recordings. Sessions without vestibular input recalibration (that is, vestibular

angular velocity neurons do not receive input from the bias cells – see model details in Supplementary Information) for both 3°/s and 1.5°/s cases were used as test examples. The 1.28°/s cue-rotation sessions were used to show the effect of cue rotation-speed on drift biases regardless of offset proximity to baseline condition. d. Synaptic weight matrices at the beginning of the 2nd darkness phase for fast (3°/s) and slow (1.5°/s) scenarios showing that baseline associations remain dominant even after 7 min of cue rotation which explains the stabilization around the 0°-offset line. e. Mean drifts (solid lines) in darkness across scenarios. Shaded areas indicate s.e.m.

Zaki Ajabi

# Reporting Summary

## Statistics

For all statistical analyses, confirm that the following items are present in the figure legend, table legend, main text, or Methods section.

| n/a | Confirmed | |
|---|---|---|
| ☐ | ☒ | The exact sample size (*n*) for each experimental group/condition, given as a discrete number and unit of measurement |
| ☐ | ☒ | A statement on whether measurements were taken from distinct samples or whether the same sample was measured repeatedly |
| ☐ | ☒ | The statistical test(s) used AND whether they are one- or two-sided *Only common tests should be described solely by name; describe more complex techniques in the Methods section.* |
| ☐ | ☒ | A description of all covariates tested |
| ☒ | ☐ | A description of any assumptions or corrections, such as tests of normality and adjustment for multiple comparisons |
| ☐ | ☒ | A full description of the statistical parameters including central tendency (e.g. means) or other basic estimates (e.g. regression coefficient) AND variation (e.g. standard deviation) or associated estimates of uncertainty (e.g. confidence intervals) |
| ☐ | ☒ | For null hypothesis testing, the test statistic (e.g. *F*, *t*, *r*) with confidence intervals, effect sizes, degrees of freedom and *P* value noted *Give P values as exact values whenever suitable.* |
| ☒ | ☐ | For Bayesian analysis, information on the choice of priors and Markov chain Monte Carlo settings |
| ☒ | ☐ | For hierarchical and complex designs, identification of the appropriate level for tests and full reporting of outcomes |
| ☐ | ☒ | Estimates of effect sizes (e.g. Cohen's *d*, Pearson's *r*), indicating how they were calculated |

*Our web collection on statistics for biologists contains articles on many of the points above.*

## Software and code

Policy information about availability of computer code

| Data collection | Data collection was done using open source software from miniscope.org (2018 version) allowing simultaneous data collection from an implanted miniscope and a behavioural camera |
|---|---|
| Data analysis | Data preprocessing was done using MATLAB (MathWorks, v. R2015a). Most of the data analysis was done using MATLAB (MathWorks, v. R2018b) a few exceptions include Python scripts run on PyCharm (v. 2018.3 (Edu)) with their outputs transferred to MATLAB. Open source code was used for cell segmentation (Miniscope software package (from miniscope.org (2018 version)); MATLAB (MathWorks, v. R2015a)), motion correction (Miniscope software package (Miniscope software package (from miniscope.org (2018 version)); MATLAB (MathWorks, v. R2015a)), spike inference (Miniscope software package (from miniscope.org (2018 version)); MATLAB (MathWorks, v. R2015a)) and head direction decoding (Python (PyCharm v. 2018.3 (Edu))). The remaining analysis was done using custom code written in MATLAB (MathWorks, v. R2015a)) except for the dimensionality reduction which was done using Python (PyCharm v. 2018.3 (Edu)) script that transfers the output to MATLAB. |

For manuscripts utilizing custom algorithms or software that are central to the research but not yet described in published literature, software must be made available to editors and reviewers. We strongly encourage code deposition in a community repository (e.g. GitHub). See the Nature Portfolio guidelines for submitting code & software for further information.

## Data

Policy information about availability of data

All manuscripts must include a data availability statement. This statement should provide the following information, where applicable:

- Accession codes, unique identifiers, or web links for publicly available datasets
- A description of any restrictions on data availability
- For clinical datasets or third party data, please ensure that the statement adheres to our policy

> All data used in this manuscript will be made publicly available via: DIO:10.6084/m9.figshare.21792689

# Field-specific reporting

Please select the one below that is the best fit for your research. If you are not sure, read the appropriate sections before making your selection.

☒ Life sciences ☐ Behavioural & social sciences ☐ Ecological, evolutionary & environmental sciences

For a reference copy of the document with all sections, see nature.com/documents/nr-reporting-summary-flat.pdf

# Life sciences study design

All studies must disclose on these points even when the disclosure is negative.

| | |
|---|---|
| Sample size | The number of subjects in this study was limited by the difficulty of accessing the mouse anterodorsal thalamic nucleus using miniaturized microscopes combined with thin (0.5mm diameter) relay lenses. The success rate of the surgeries being very low, our objective was to record from at least three mice per experiment.<br>On the other hand, for each mouse we get an average of over a hundred neurons simultaneously recorded per session which constitutes a sample size with an order of magnitude higher than the largest neural population ever reported in the studies of the head direction system, at the time of submission of this manuscript. |
| Data exclusions | Outliers were identified as data points outside the range of mean +/- 3*(standard deviation). If found, these outliers were excluded from the analysis.<br>Data automatically included based on specific selection criteria was manually checked to ensure inclusion was not biased by measurement noise. |
| Replication | Findings in this manuscript have been replicated in all animals and data per animal is shown in supplementary figures. |
| Randomization | This is not relevant to our study. We have a single experimental group. |
| Blinding | This is not relevant to our study. We have a single experimental group. |

# Reporting for specific materials, systems and methods

We require information from authors about some types of materials, experimental systems and methods used in many studies. Here, indicate whether each material, system or method listed is relevant to your study. If you are not sure if a list item applies to your research, read the appropriate section before selecting a response.

## Materials & experimental systems

| n/a | Involved in the study |
|---|---|
| ☒ | Antibodies |
| ☒ | Eukaryotic cell lines |
| ☒ | Palaeontology and archaeology |
| ☐ | ☒ Animals and other organisms |
| ☒ | Human research participants |
| ☒ | Clinical data |
| ☒ | Dual use research of concern |

## Methods

| n/a | Involved in the study |
|---|---|
| ☒ | ChIP-seq |
| ☒ | Flow cytometry |
| ☒ | MRI-based neuroimaging |

## Animals and other organisms

Policy information about studies involving animals; ARRIVE guidelines recommended for reporting animal research

| | |
|---|---|
| Laboratory animals | All animals used in this study are male wild-type mice (C57Bl/6, Charles River) aged between 6 and 8 weeks at the time of the first surgery. |

| Wild animals | The study did not involve wild animals. |
|---|---|
| Field-collected samples | The study did not involve field collected samples. |
| Ethics oversight | All experiments were carried out in accordance with McGill University and Douglas Hospital Research Centre Animal Use and Care Committee (protocol #2015-7725) and in accordance with Canadian Institutes of Health Research guidelines. |

Note that full information on the approval of the study protocol must also be provided in the manuscript.

