## [Peer Review File · Nature]

Manuscript Title: Population dynamics of head-direction neurons during drift and reorientation

Reviewer Comments & Author Rebuttals

Reviewer Reports on the Initial Version:

Reviewers' comments:

Referee #1:

Remarks to the Author:

This paper reports an investigation of anterior thalamic head direction neurons, made using microendoscopy in mice exploring a dynamically cue-controlled arena. The main finding is that in addition to the orientation of network activity relative to the outside world (the head direction signal, which was known) there is a second experience-modulated variable, network gain, that is related to the first in various interesting ways: most surprising of which is its coupling to the location of an absent visual cue, during periods of darkness.

I think these findings are interesting, and the experiment is well conducted. I have some a few comments on how the findings are presented and interpreted, as follows.

Abstract: "brief presentations of a rotated landmark revealed an attraction of the network back to its initial orientation, suggesting a time-dependent mechanism underlying the formation of these network gain memory traces" – the meaning of this wasn't clear, without yet knowing the comparison between the long and short cue exposures. Perhaps "network showed attraction back to baseline after brief but not longer exposures to a rotated cue" or something similar. Also, the conclusion from the abstract about "new mechanistic insights" is weak and uninformative – what mechanistic insights exactly? A similar comment pertains to "novel insights" on p3.

P3 It would be helpful to state the number of animals and sessions. How long were the recording sessions?

I have issues with the term "firing rate" for the calcium signal; firing rate pertains to spike data and is measured in Hz, whereas what is being measured here is spiking probability. Network gain is thus the ensemble average of this probability. I note that gain might decrease if new neurons became weakly active, which might actually reflect an increase in activation (more neurons crossing firing threshold, but pulling the average down) so it would be good to have it confirmed that this was not the case.

A question about gain: did this contra-vary with individual cells' tuning curve width? That is, did the attractor bump get smaller and broader vs. taller and narrower?

P5 The dimension reduction procedure needs some additional explanation, as does Figure 1h. At first read-through it seemed to me that the authors were getting out what they put in: they constrained the first dimension to be circular and then found a ring-like structure... this seemed trivial. Also, Figure 1h seemed to be using three graphical variables (angle, radial distance and color) to describe two data variables, with color being redundant. Eventually I realised that there are in fact three data variables: one is head direction and one is the internal directional representation of the network (the third is the network gain), and it's only in the baseline condition that head direction and internal orientation are the same. These become decoupled with cue shift, however, and this is why, in Figure 2c, color no longer maps exactly to angle. It might be useful to explain the situation in these more concrete terms for readers not used to

thinking in terms of state space variables. Additionally, Figure 2c and 1h could perhaps be shown side by side so that the reason for this way of presenting the data (which is useful) is understood.

"state transitions would be fastest at the lower end of the radial component because of the decreasing distance between states representing different angles" - this doesn't make sense, because the smaller distance just arises from the arbitrary geometry of polar plots. If the variables were represented in, say, a Cartesian coordinate system then the distances would be the same. Indeed I'm not sure there is a direct physical or geometric analogy or explanation for the (interesting) observation that the network is easier to reorient when it is in a low-gain state: it likely arises from, as the authors say, the lower levels of the lateral inhibition that stabilizes the attractor bump. However, this is a post hoc explanation, not a prediction.

Fig. 2 legend: "however state points are color coded ..." perhaps "shaded", as they are all blue.

"Reconstructed mean bump of activity in egocentric reference frame across radius ranges." - it needs to be explained what "egocentric" means in this context, and indeed I don't think it is the right word since it isn't strictly speaking egocentric. Perhaps "Internal network reference frame"? (I assume that's what was meant).

The fast/slow resetting - was this a bimodal phenomenon or did reset rates span the whole range? I gather the latter and wonder if a correlation analysis would be better than binarizing the data.

P7 "Behavioral differences before and after cue onset were not significant" - what behaviors specifically? They only report AHV so it would be better to specify.

P8 "we observed an increase in the variability of drift" - it was ambiguous whether this meant variability of drift *rate* or of total drift, and it is implied in the next sentence that the meaning was not in fact variability but amount - this could be clarified (see later comment about the figure).

P9 "One interpretation of these results is that the HD network keeps a 'memory trace' of the visual input even after it is removed which might be used as an internal reference to guide behaviour" - this is an intriguing idea but the notion that it has functional significance in driving behavior seems a little implausible, since the natural world has many cues in every direction, which would thus activate the whole network uniformly. It seems more likely to arise from an asymmetry in synaptic connections in this specific, unusual environment having only one single focal cue. That is, perhaps those cells that were active when the animal faced the cue (when it was present) acquired stronger focal directional inputs from other cues as well, via Hebbian association, and thus continue to fire more strongly when the animal faces in their preferred firing direction even without the visual cue. One could call this a memory trace of sorts, but not one with a likely functional significance. Nevertheless it is an intriguing observation, revealing of selective plasticity in the visual/HD network, and merits further investigation to find out what does lie behind it.

P10 "each epoch exhibited a different stereotyped pattern of drifts" - it should be mentioned that there is a temporal confound here, as the rotations were always conducted in the same direction and always by 90 degrees. This might account for at least some of the difference between the 90 deg CW and 90 deg CCW drift rates.

P11/12 "These results indicate that the internal representation of the baseline allocentric reference frame is not entirely lost after a reset and can still influence the HD network, in darkness, depending on the duration of experience within the competing reset reference frame

context, which implicates plastic processes at play at some stage in this network." Why does this necessarily imply plasticity? (Other than in the trivial sense that the system originally had to learn about the cue orientations with respect to the network). Could it not just be cue combination, with the static background cues (olfactory etc) and the dynamic visual cue exerting competing effects on the network?

Discussion: there were some areas that could be clarified, as follows.

The summary paragraph of the main findings omitted the interesting drop in network gain with reorientation, although this is discussed later.

"informatively linked to future network dynamics" – I wasn't clear what this meant

I'm not sure I agree with "network gain landscape can maintain a memory trace of a stable reference frame" as I don't feel that this was demonstrated. Insofar as there were traces in the gain landscape, these were of the missing visual cue, but this is not necessarily a reference frame. The stable reference frame defining the baseline state of the system is surely preserved by the persisting static environmental cues, not memory – because where would the memory be held, and how, given the movements of the animal? Are the authors proposing the superposition of two simultaneous reference frames in the network itself, both continuously updated by path integration? This seems unlikely to me. If that's what they *are* proposing it needs better explanation and some defense. Surely the simplest explanation is that the network has inputs from two sets of cues, static background cues that remain fixed in the room reference frame and a dynamic visual cue – when these are dissociated the attractor is excited at two places. The increased gain when the rat faces the removed cue is likely due, as mentioned earlier, to cue-potentiated enhancement of the other co-existent inputs at that orientation.

"This suggests that the visual flow can be used by the HD system to recalibrate the integration of angular velocity information (i.e. vestibular input) in order to anchor the internal HD representation to a dynamic visual reference frame". This was an interesting observation, consistent with other recent reports in the spatial literature of dynamic neuron recalibration, but I slightly take issue with "in order to anchor..." as this is making interpretations that go beyond the data. Indeed, I think it's very unlikely that this would be the adaptive function of this process because when, in nature, would an animal need to anchor its internal compass to a rotating world? It seems more likely that the adaptive reason for this process is the reverse: it is to use stable landmarks to calibrate the vestibular system to compensate for fluctuations in its gain, perhaps due to factors like changes in endolymph density. In any case, whatever the speculation, I suggest it be reserved for the discussion.

"realign" should be "realigning"

"strong influence of the internal representation of the baseline reference frame" – as above, I don't think we can conclude that it's internal; surely it must be external. That is, due to olfactory and other static environment cues, linked to the network, that help drive it when the visual cue starts to get near their range of influence and thus pull the network back towards that baseline. As mentioned above, if the authors want to argue that it is not due to persistent environmental cues they would need to come up with some explanation for how two reference frames could be maintained and constantly updated in the one network, and why they would sometimes interact (when the frame orientated by the visual cue gets near the frame defined by the baseline state.)

P16 typo – "grid ells"

P25 Where the variables are explained, N should be listed here as well. Also, perhaps combine

HD network simulation and Attractor network model as there doesn't seem a good reason to split these.

P33 I am slightly uneasy this arbitrary division of the data into fast and slow, and the comparison of a factors (AHV) that a priori wouldn't have been predicted to co-vary like this. It feels a little like data mining, although I appreciate that many findings in this field were not predicted a priori. One thing that could help is an internal replication: i.e., to show that the effect is present separately in all three mice. Incidentally, here and elsewhere, the y axes should start at zero so as not to visually inflate effects.

P34 The drift speed analysis doesn't seem right to me. The authors have picked an arbitrary time point, 5s, at which to compare the deg/s drift rate but it is evident from the first graph that the rate is an exponential function of time, so the best comparison would have been the exponential time constant. What was the take-home message from this analysis? In part c I didn't quite understand "Each dot represents a correct reset" since shouldn't the resets have been at 90 deg (or multiples)?

P36 Figure part a – Drift variability: I'm not sure why attention is on the variability (size of error bars) rather than the amount of drift. The error scales with the amount so this seems slightly trivial. The more important observations here are (i) that the drift increases across epochs (maybe due to decreased cue control with mismatch learning?) and (ii) that the direction is towards baseline in D2 and D4.

P38 I suggest to rotate the heat plots 90 deg so that the x axes align with the line plots underneath.

P39 "2m-cue-exposure" should be "2min-cue-exposure" and "(same data as in e)" should (I think) be "(same data as in b)"

P44 The description of the inhibition-based attractor should mention and reference Song and Wang (<https://pubmed.ncbi.nlm.nih.gov/15673682/>) who were the first (I believe) to show that a ring attractor could work this way.

Signed: Kate Jeffery

Referee #2:

Remarks to the Author:

This paper explores the dynamics of the head-direction (HD) representation in neurons imaged from the anterodorsal thalamus of freely behaving mice, following various manipulations of a prominent visual cue in the environment. The first manipulation consists of alternating epochs of light and darkness, where in different light periods the cue is presented at different locations on a screen that surrounds the animal. It is well known that the HD representation is "anchored" to the cue, i.e. if the cue is moved from one side of the room to another, the HD representation rotates accordingly – in other words, the mouse HD signal is referenced to the location of the cue. The first question asked by the authors is how the dynamics of this anchoring process unfold. They identify the population activity gain – the strength of the HD representation essentially – as an important factor contributing to these dynamics. When the gain is low (high), the HD signals anchors quickly (slowly). This phenomenon can be accounted for by a semi-standard neural network model of the HD signal, where the gain is realized by an external input. In my understanding, when the gain input is low, the external visual inputs to the network dominate the recurrent inputs, resulting in faster anchoring. The second question is what

happens during the dark epochs. Analysis of the neural dynamics reveals that the population gain is higher in correspondence of the previous location of the cue. This striking observation could potentially underlie the very slow anchoring dynamics observed during dark epochs. In other words, despite the absence of visual cues, the HD representation could maintain a memory of a past cue location in the form of a location-dependent gain modulation. In a second set of cue manipulations, the authors zoom into the memory phenomenon revealed by the first experiment, where the HD representation appears to slowly anchor to a visual cue that is no longer present. To better characterize this phenomenon, the authors perform a new experiment where the light periods (except the first) are of shorter duration. In this case, the anchoring to new cue locations is only transient, and in the absence of cues the representation anchors back to the location of the cue during the first light period. A dynamical systems-based analysis of the population activity is consistent with this relaxation phenomenon. Lastly, the authors wonder whether a slowly rotating cue during the light period would not only continuously anchor the HD representation, but also similarly affect its dynamics during the subsequent dark period. Strikingly, this was indeed the case.

The HD system is of fundamental importance for spatial navigation, and the analysis of large-scale imaging of thalamic activity in this contribution reveals novel, intriguing neural correlates that might underlie, for instance, memory-guided goal-directed behavior. In my opinion, the topic is of potential interest to a broad audience.

However, this reviewer found the ms difficult to understand. This is likely due, in no particular order, to the (i) large amount of different analysis techniques and modeling employed in the paper, which might not always lead towards a crisp main message (ii) the enumeration of several phenomena, as somewhat implied by the generic Title, without the guidance that could provide the “mechanistic insights” promised by the authors in the Abstract (iii) potentially lacking or confusing description/use of core concepts, methods, terminology, and interpretation of the results. Hence, it is challenging to evaluate the strength of some of the main results. The authors could consider the suggestions below, which will hopefully clarify the point of view expressed above. I am ordering the comments according to figure numbers, highlighting what in my mind are the points that could most benefit from clarifications/additional modeling/additional analysis/simplification.

Fig 1:

1) The single cell tuning analysis in Fig 1c assumes that neurons have nice unimodal tuning curves. This might not be the case (as suggested by the off-diagonal peaks in 1g, and it has been shown that it's not the case with analysis of ephys recordings from Peyrache and collaborators for instance). The resulting correlation between data and prediction of this model (1f) could perhaps be improved by relaxing this assumption.

2) Major: I think that the single cell analysis above, the Bayesian decoding analysis (1i), and the feedforward neural network analysis (1h) assume that the ADN dynamics is low dimensional – at most 2d for the analysis in 1h. Why not show this explicitly? It could be that some of the intriguing dynamics described in the subsequent figures is correlated with changes along other dimensions in neural space. It could be helpful to show that it's not the case.

3) Fig 1h: Why is there no dispersion along the radial dimension here? As shown later angular velocity correlates with the gain/radius, one would expect that the angular velocity would play a role here.

Fig2:

1) Major: The analysis in Fig 1h is used as a motivation for the experiment in Fig2 (beginning of section “Network gain covaries with resetting dynamics during cue manipulation”). The intuition provided in the text relies on the knowledge of the energy in e.g. continuous attractor networks,

but this intuition is never unpacked explicitly and could be confusing without additional clarification. One option could be to elaborate more on this in the text (perhaps with accompanying schematic panels), but a simpler/more conservative option could be to remove the ff neural network-based analysis from the main text. After all, the radial dimension in this polar representation is only used transiently, and immediately related to the gain of the population activity in Fig 2d,e. The latter appears to be a more intuitive description of the additional dimension.

2) Major: The term drift used to describe the relationship between the internal HD signal and the external visual cue could be confusing. This quantity is for instance referred to as "offset" in the literature (see e.g. references to the drosophila HD system used in the ms). I think that drift typically refers to a component of the dynamics (equation of motion) of so-called drift-diffusion processes, where the drift corresponds to the deterministic component that drives the anchoring, and the diffusion to the noisy component (described as standard deviation of the drift in a subsequent figure). If the authors agree with these definitions, amending the current terminology in the ms would be of help.

3) Major: The drift (or offset) as defined in the ms is a rather abstract concept. I think that showing what happens to the HD representations during a single session (as shown for instance in the bottom panels of Fig 5a,b) earlier in the manuscript, to illustrate what the authors mean by drift, would go a long way toward making the ms more interpretable.

4) Major: the model beautifully shows how a simple non HD-specific external input can recapitulate the anchoring dynamics during light epochs. However, the model does not take into account the fact that (i) the animal is moving during the trial – hence the visual input is changing, and the angular velocity of the animal is being integrated by the model network. This becomes particularly relevant when the authors shows that the angular velocity just preceding the cue was a good predictor of fast/slow resets (EDF 7) and that the anchoring dynamics depended on the distance between pre-cue and final drift (EDF 8). Why choose to only model one aspect of these rich dynamics? I thought that the model was a very valuable addition, in that it helped with providing "mechanistic insight". Extending the model to capture the additional relevant variables of the problem (heading of the animal, location of the cue, angular velocity) would be illuminating. Moreover, the model is unfortunately abandoned for the rest of the paper- relying on it throughout could really help to contextualize all the interesting phenomena reported in the ms and provide insights into how these phenomena "work".

Fig 3:

1) Major: The focus now shifts on diffusion of the offset during the dark period. Fig 3d is one of the most striking results in the paper in my opinion, also emphasized by the author statement: "To our knowledge, this is the first evidence of such long-term experience-dependent preferential firing in the thalamic HD cells". It would be great if "internal HD" here would be defined as precisely as possible, perhaps describing this more explicitly or depicting it with a time-course of populations activity across conditions, where both the offset re-centering towards the cue and the respective amplitude tuning, can be observed. My understanding of this result is complicated by a description of what happens in the main text: "The HD neurons that fired when the animal was facing the cue during baseline have higher average firing rates during darkness, following a reset", as opposed to what it is said in Methods ("decoded HD" in the "gain heatmap analysis" section). Is it possible that with one of these definitions the internal HD is defined in a such a way that "neurons that fired when the animal was facing the cue during baseline" continue to fire when the animal is facing the new cue location due to the offset tethering (resets and drifts in dark), and they do so with higher firing rate than the other neurons? The results shown in Figure 3d would then mean that the amplitude tuning encodes the cue position estimate, and that tuning emerges and is tightened at the later periods (ED Figure 11). Does this tuning exist only during the dark periods? How do the gain profiles from the light periods

preceding the D2-D4 periods look like? If the tuning is only present during the dark periods, could you comment on why it appears later, is this somehow related to the drift? Are there similar gain profiles during the 'bright' and 'dark' periods of the optic flow experiments where there are drifts?

Fig 4:

1) Why is the protocol used in this experiments (cue at 0, -90, 90, -90, 90, 0) different what the one used before (cue at 0, 90, 180, -90, 0)?

2) Major: 4a, Why does the offset goes to 90 degrees in the presence of a cue at -90 degrees (as opposed to Fig 2)?

3) Major: 4d. This is a new analysis technique, showing that the drift dynamics should converge to fixed points during the dark period (why not only one at 0 degrees btw, is there a prediction that can be made there?). Isn't this result somewhat expected from the dynamics of the drift shown in Fig 4a – could there have been alternative pictures? In addition, this dynamical systems description ignores other relevant variables, such as the gain shown in the other panels. Should one thing about this flow field as being dependent on additional variables? If so, why ignore them? Also, as mentioned above, a network model of these phenomena could be more illuminating. Similar comments apply to Fig 5.

4) There is an asymmetry in Fig 4f, larger gain difference for negative AV at internal HD ~90 degrees. How should this be interpreted?

Author Rebuttals to Initial Comments:

Referee #1 (Remarks to the Author):

This paper reports an investigation of anterior thalamic head direction neurons, made using microendoscopy in mice exploring a dynamically cue-controlled arena. The main finding is that in addition to the orientation of network activity relative to the outside world (the head direction signal, which was known) there is a second experience-modulated variable, network gain, that is related to the first in various interesting ways: most surprising of which is its coupling to the location of an absent visual cue, during periods of darkness.

I think these findings are interesting, and the experiment is well conducted. I have some a few comments on how the findings are presented and interpreted, as follows.

Abstract: “brief presentations of a rotated landmark revealed an attraction of the network back to its initial orientation, suggesting a time-dependent mechanism underlying the formation of these network gain memory traces” – the meaning of this wasn't clear, without yet knowing the comparison between the long and sort cue exposures. Perhaps “network showed attraction back to baseline after brief but not longer exposures to a rotated cue” or something similar. Also, the conclusion from the abstract about “new mechanistic insights” is weak and uninformative – what mechanistic insights exactly? A similar comment pertains to “novel insights” on p3.

We would like to thank the reviewer for the positive assessment and the constructive feedback of our work. We agree with the reviewer's suggested changes to the abstract. We also consider that the time-dependent drift response is an important observation that suggests the involvement of plasticity in the HD network which warrants inclusion in the abstract.

Based on these suggestions, we have substantially edited the abstract. The new abstract reads the following:

“The head direction (HD) system is often intuitively thought of as the brain's internal compass^{1,2},

classically formalized as a 1-dimensional ring attractor network^{3,4}. Unlike a globally consistent magnetic compass, however, the HD system does not have a universal reference frame. Instead, it anchors to local cues, maintaining a stable offset when these cues rotate⁵⁻⁸ and drifting in the absence of reliable referents^{5,8-10}. While much is known about the cues that anchor the HD system under certain navigational conditions from small ensemble recordings, there remain several open questions about the mechanisms underlying anchoring and drift which can only be answered at the population level. For example, the extent to which the 1-dimensional description of population activity holds under conditions of drift, and whether this might interact with anchoring mechanisms, remains unknown. To fill this knowledge gap, we performed large population recordings of thalamic HD cells using calcium imaging in freely behaving mice during a variety of controlled rotations of a visual landmark in a familiar environment. Across experiments, we found that population activity also varies along a second dimension, which we refer to as 'network gain', especially under circumstances of cue conflict and ambiguity. Activity along this network gain dimension predicted realignment and drift dynamics, including the speed of network realignment. During darkness following removal of the visual landmark, the 360° azimuthal profile of network gain maintained a 'memory trace' of the previously displayed landmark, suggesting plasticity in the formation of network gain patterns. Further experiments demonstrated that the HD network returned to its baseline offset after brief but not longer exposures to a rotated cue. The experience-dependent nature of this phenomenon suggests that a memory of prior associations between the HD neurons and allocentric cues can be maintained and can still influence the internal HD representation, following brief conflicts in external sensory inputs. Building on these results, we show that continuous rotation of a visual landmark induced a similar rotation of the HD

representation which persisted following removal of the landmark, demonstrating that experience-dependent recalibration of the HD system can extend to dynamic reference frames. Finally, we reproduced many of our key findings with a computational model which formalizes how the neural compass can flexibly adapt to changing environmental cues to maintain a reliable representation of head direction. Together, these results challenge classical 1-dimensional interpretations of the HD system while providing new insights into the dynamic interactions between this system and the cues to which it anchors.”

With regard to ‘new mechanistic insights’, in its previous version the manuscript suggested that a general source of gain modulation (included in the model) and plastic processes between the core HD network and cortical and parahippocampal regions (PoS, RSC, etc...) could underlie the effects we observe. In the current version we formalize this suggestion in a more detailed model which includes these mechanisms (from Figs. 3, 4 and 5). We now show how a model based on Hebbian learning between cortical and subcortical layers could explain the ‘reversions ’in darkness, as well as how recurrent excitation could sustain a long-term recalibration of vestibular input integration. These results demonstrate that these mechanisms can provide a plausible qualitative explanation for many of the phenomena we report.

P3 It would be helpful to state the number of animals and sessions. How long were the recording sessions?

Thank you for this suggestion. We agree. We have added this information into the text:

“We performed calcium imaging of the anterodorsal thalamic nucleus (ADN) in three mice using a miniaturized head-mounted endoscope³²⁻³⁴. Our recordings (n=102 sessions in total; Each session lasted for 20 minutes with baseline period ranging between 3, 5 and 10 minutes depending on the experiment) allowed us to track the neural activity from up to 255 ADN cells simultaneously as mice freely explored a small elevated circular platform inside a larger enclosed chamber (Fig. 1a-g, Extended Data Fig. 1, Methods).”

I have issues with the term “firing rate” for the calcium signal; firing rate pertains to spike data and is measured in Hz, whereas what is being measured here is spiking probability. Network gain is thus the ensemble average of this probability. I note that gain might decrease if new neurons became weakly active, which might actually reflect an increase in activation (more neurons crossing firing threshold, but pulling the average down) so it would be good to have it confirmed that this was not the case.

First, we agree with the reviewer’s comment on the use of the term ‘firing rate’. In the revision, we have replaced it with the terms ‘firing activity ’or ‘neural activity ’throughout the text. Thank you for this suggestion.

In addition, the reviewer raises an interesting point regarding the effect of new neurons becoming increasingly active which might pull the average down. We think that one way of showing this is not the case, in our data, is by looking at the average bump of activity for different gain amplitudes. Extended Data Figure 6 (which also serves as an answer to the reviewer’s next comment) has been added to show that a decrease in gain corresponds to a global decrease in neural activity across all internal angles, including neurons outside the activity packet (i.e. baseline activity level). Moreover, the width of the bump of activity does not change significantly.

A question about gain: did this contra-vary with individual cells' tuning curve width? That is, did the attractor bump get smaller and broader vs. taller and narrower?

Thank you for this question. This is indeed an important question that was not covered in the initial submission. We have added a new figure (**Extended Data Figure 6**; also see below) to address this question. This figure below shows that gain modulation was not accompanied by significant changes in the width of the bump of activity.

Extended Data Figure 6. Reconstructed bump of activity (averaged over $N=42$ sessions of the first experiment) for varying network gain ranges. The gain modulation not only affects the activity packet but also baseline activity. The decreasing baseline amplitude at low network gain indicates that the modulation is not driven by increased activity outside the main activity packet (e.g. secondary bumps of activity). Notice that the width of the activity packet remains within a narrow range. 'fwhm': the full width at half maximum.

P5 The dimension reduction procedure needs some additional explanation, as does Figure 1h. At first read-through it seemed to me that the authors were getting out what they put in: they constrained the first dimension to be circular and then found a ring-like structure... this seemed trivial. Also, Figure 1h seemed to be using three graphical variables (angle, radial distance and color) to describe two data variables, with color being redundant. Eventually I realised that there are in fact three data variables: one is head direction and one is the internal directional representation of the network (the third is the network gain), and it's only in the baseline condition that head direction and internal orientation are the same. These become decoupled with cue shift, however, and this is why, in Figure 2c, color no longer maps exactly to angle. It might be useful to explain the situation in these more concrete terms for readers not used to thinking in terms of state space variables. Additionally, Figure 2c and 1h could perhaps be shown side by side so that the reason for this way of presenting the data (which is useful) is understood.

We agree that the description of the dimensionality reduction method needed to be improved. The state space analysis was meant to illustrate the need for a multidimensional investigation of the internal HD representation. Within the scope of our study, the multidimensional investigation is limited to the 2D case. Since, the internal HD representation is circular by nature (angular dimension), the simplest dimensionality augmentation would be to add a radial dimension, thus the proposed analysis. While our

algorithm is trained on circular data (the ‘imposed’ circularity is only due to training), there are no constraints on the radius, even for baseline data. The fact that we obtain a ring structure during baseline is because there was no significant variability in the latent variable R (i.e. radius). Just to clarify: Figures 2c and 1h do not represent the same recording session. In 2c, we show a direct comparison between baseline and entire session from the same example.

We have substantially revised the text to better convey the intuition behind this analysis. The new text reads:

“To visualize the potentially low-dimensional structure of the representational space of the head direction network, we developed a method to project the high dimensional input data from the neural space onto a 2-dimensional polar state-space (Methods, Dimensionality reduction using feedforward neural network). This method involves training a deep neural network on the measured head direction while allowing an untrained latent variable to capture variability in the neural data that cannot be explained by changes in the head direction alone. This inferred latent variable constitutes the radial component in the 2-dimensional polar state-space (i.e. secondary dimension). When applied to the baseline data, we obtain a ring-like structure (Fig. 1h), reminiscent of both attractor HD network models and prior analyses^{3,17,42}. This further confirms that, in stable conditions, the internal HD representation is indeed approximately unidimensional.”

“state transitions would be fastest at the lower end of the radial component because of the decreasing distance between states representing different angles” - this doesn’t make sense, because the smaller distance just arises from the arbitrary geometry of polar plots. If the variables were represented in, say, a Cartesian coordinate system then the distances would be the same. Indeed I’m not sure there is a direct physical or geometric analogy or explanation for the (interesting) observation that the network is easier to reorient when it is in a low-gain state: it likely arises from, as the authors say, the lower levels of the lateral inhibition that stabilizes the attractor bump. However, this is a post hoc explanation, not a prediction.

Thank you for this comment. We understand that the previous description could be interpreted in different ways. Our goal is to provide a physical intuition for possible fluctuations in the population activity. For that, we have toned down the first paragraph in the Section **“Network gain covaries with resetting dynamics during cue manipulation”**:

“To investigate how HD network dynamics enable reorientation, we recorded the HD network during a reorientation paradigm where the visual cue disappeared and reappeared in different locations. Specifically, following a baseline recording, the cue was removed for two minutes (darkness) after which it reappeared at a 90° shifted position for two minutes. We repeated this sequence four times per recording session (Fig. 2a).”

Later in this section we briefly describe this intuition about network transitions in 2D polar state space and point to a new Extended Data Figure 5 for more details:

“Intuitively, allowing the internal HD representation to live in a 2D polar state space makes the distance between any two given angles θ_1 and θ_2 a function of the radial component, which led us to hypothesize that changes in radius not only reflected changes in the overall population activity but would also correlate with changes in the speed of reset (Extended Data Fig. 5).”

Our new schematic figure (**Extended Data Figure 5**; see below) has been added to better explain the intuition behind our idea. Put simply, with a lower network activity (i.e. lower gain), shifting the activity bump along the circle by a certain degree would require a smaller change of neural activity compared to the change under a high-network-activity condition. Mathematically, this could be understood by a smaller distance traveled in the space of neural population activity. We feel that this provides a useful intuition for interpreting our results.

Extended Data Figure 5. Schematic showing the dynamics of changes in the internal HD representation from population-activity and state-space perspectives. **a, b.** The amount of the change of neural activity in the activity during bump movement depends on the gain of the network. Here, the x-axis represents the neuronal space (i.e., neurons sorted according to the preferred head directions, assuming a uniform distribution of HD cells according to their preferred firing directions). Mathematically, the distance between representations of the internal HD from start to end of a rotation, in the Euclidean sense, is smaller at lower network gain. **D:** Euclidean distance, $\mathbf{r}^{activity}$: $N \times 1_t$ vector of firing rates from N HD neurons composing the HD network, at time t for different activity levels (i.e. high or low). **c.** The concept of decreasing distance between internal HD representations, at lower network gain, is naturally captured in the 2D polar coordinate system if we assume that the radius reflects the level of neural network activity. The distance traveled in the hypothetical state-space of the HD network will be greater when the radius is larger as well as when the net gain is higher, which could be quantified by the total change of the firing rate across the network. Thus, we hypothesize that the radius is correlated with the overall population activity (i.e. network gain) and that decreasing distance facilitates rotations across the HD network. Assuming that the internal HD representation lives in a 2D polar state space where each state is defined by a phase and a radius, state transitions would be fastest at the lower end of the radial component because of the decreasing distance between states representing different angles, near the center of the baseline ring. Bar graphs are only indicative and not to scale.

Fig. 2 legend: “however state points are color coded ...” perhaps “shaded”, as they are all blue.

Thank you for this suggestion. We have revised the text as follows:

“Left: same as in (c) however state points are shaded according to their radius.”

“Reconstructed mean bump of activity in egocentric reference frame across radius ranges.” – it needs to be explained what “egocentric” means in this context, and indeed I don’t think it is the

right word since it isn't strictly speaking egocentric. Perhaps "Internal network reference frame"? (I assume that's what was meant).

We agree with the reviewer's suggested change. In our case, the "egocentric" reference frame refers to a polar coordinate system that has its origin at the center of the animal's head, 0° is the direction from the origin towards the nose, 90° is the direction from the origin towards the left ear, -90° is the direction from the origin towards the right ear and $\pm 180^\circ$ is the direction from the origin towards the back of the head. We assume that animal movement is contained within the azimuthal plane (no pitch or roll). The egocentric bump of activity is obtained through a circular shift of the angular axis (x axis in Fig. 2) by the amount of measured HD (i.e. allocentric reference frame), in the opposite direction of head rotation, which is why the bump is always centered around 0° .

We have revised the text to avoid the confusion:

"Reconstructed mean bump of activity in internal reference frame across radius ranges."

The fast/slow resetting – was this a bimodal phenomenon or did reset rates span the whole range? I gather the latter and wonder if a correlation analysis would be better than binarizing the data.

Thank you for this question. The separation of resets into fast and slow groups was meant to provide an easy-to-understand illustration of the covariation of gain and reset speed by comparing the mean curves for both variables. This separation was not meant to imply a bimodal phenomenon (see figure below of reset-speed distribution) but was rather a first step to infer the dynamical relationship between gain and reset response.

We plotted the averaged instantaneous drift speed during resets as a function of (binned) gain with no prior binarization (see main **Figure 2i**) and showed that the two quantities were indeed correlated. We believe the latter analysis directly addresses the reviewer's comment suggestion regarding the correlation analysis.

In addition, we provided a computational model that replicated the said dynamical relationship and which was a good predictor of reset response based on input gain (see Fig. 2k, Extended data Figs. 8, 9).

Reviewer Figure 1: Distribution of reset speeds.

P7 “Behavioral differences before and after cue onset were not significant” – what behaviors specifically? They only report AHV so it would be better to specify.

“Behavioral differences as measured by the head angular velocity before and after cue onset were not significant and could not explain the sharp decrease in gain amplitudes (Extended Data Fig. 10a). However, reduced head angular velocity immediately preceding cue events was predictive of fast resets, and vice versa (Extended Data Fig. 10b, c).”

P8 “we observed an increase in the variability of drift” – it was ambiguous whether this meant variability of drift *rate* or of total drift, and it is implied in the next sentence that the meaning was not in fact variability but amount – this could be clarified (see later comment about the figure).

The reviewer is correct that drift variability here refers indeed to the total drift as we show it in Extended Fig. 10a. We have clarified this as follows:

“we observed an increase in the average amount of drift relative to baseline...”

P9 “One interpretation of these results is that the HD network keeps a ‘memory trace ’of the visual input even after it is removed which might be used as an internal reference to guide behaviour” – this is an intriguing idea but the notion that it has functional significance in driving behavior seems a little implausible, since the natural world has many cues in every direction, which would thus activate the whole network uniformly. It seems more likely to arise from an asymmetry in synaptic connections in this specific, unusual environment having only one single focal cue. That is, perhaps those cells that were active when the animal faced the cue (when it was present) acquired stronger focal directional inputs from other cues as well, via Hebbian association, and thus continue to fire more strongly when the animal faces in their preferred firing direction even without the visual cue. One could call this a memory trace of sorts, but not one with a likely functional significance. Nevertheless it is an intriguing observation, revealing of selective plasticity in the visual/HD network, and merits further investigation to find out what does lie behind it.

We agree with the reviewer’s comment. Although the speculation that this ‘memory trace ’provides an internal reference to guide behavior may be tempting, we agree that we cannot make claims about the behavioral-relevance of this signal from our current data.

While this remains speculative, we believe that the text could benefit from linking this observation to our hypothesis regarding the HD system operating at different energy levels. In other words, we believe that, at a circuit level, the memory traces could stabilize the HD network by maintaining a high enough level of neural activity around the internal representation of salient cues which reduces the influence of existing background noise because of increasing lateral inhibition. In the case of the current experiment, there is only one salient cue, and network gain around the internal representation of this cue is kept at ~baseline level. On the other hand, the gain drops significantly below baseline level, farther away from the internal cue location, which produces a less accurate representation of HD due to a lower signal-to-noise ratio which, in turn, may induce random representational drifts caused by the existing background noise in neural activity and the reduced lateral inhibition.

Regarding the comment on multiple cues, it is true that, according to this logic, more salient cues in different directions would activate the network uniformly which we would expect to result in a more

stable representation of HD in darkness with less drift. However, we think that, in such a case, the overlapping effect of multiple cues on the HD network gain would make it difficult (if not impossible) to separate the individual contribution of each cue.

We have revised the text to clarify these points:

“One interpretation of these results is that the HD network keeps a ‘memory trace’ of the visual input even after it is removed which might help stabilize the system in darkness by maintaining a high level of neural activity (i.e. high signal-to-noise ratio) around the internal representation of salient cues and thus prevent spontaneous drifts of the network states due to background noise.”

P10 “each epoch exhibited a different stereotyped pattern of drifts” – it should be mentioned that there is a temporal confound here, as the rotations were always conducted in the same direction and always by 90 degrees. This might account for at least some of the difference between the 90 deg CW and 90 deg CCW drift rates.

Indeed, our interpretation of the differences between the 90°-CW and the 90°-CCW drift rates is that it is due to the persistent cue shift in one direction.

To address this point, we added the following text:

“Moreover, the persistent shifting of the visual cue in one direction appeared to further bias the drift in that direction (Extended Data Fig. 13b, c).”

P11/12 “These results indicate that the internal representation of the baseline allocentric reference frame is not entirely lost after a reset and can still influence the HD network, in darkness, depending on the duration of experience within the competing reset reference frame context, which implicates plastic processes at play at some stage in this network.” Why does this necessarily imply plasticity? (Other than in the trivial sense that the system originally had to learn about the cue orientations with respect to the network). Could it not just be cue combination, with the static background cues (olfactory etc) and the dynamic visual cue exerting competing effects on the network?

This is an important point that was not thoroughly addressed in the previous version of the manuscript. Thank you for pointing this out. The observed drift patterns in the 2-minute and 20-sec shifted-cue exposure experiments show that attraction to baseline is experience-dependent in some way. While one can imagine experience-dependent mechanisms that do not involve plasticity as the reviewer makes clear, plasticity within the HD system is a plausible mechanism that is attractive in terms of the scope of the phenomena it can explain in our data. Plasticity involvement has also been speculated in previous computational models of the HD system (Cope et al., 2017; Ocko et al., 2018; Page & Jeffery, 2018) and, experimental data in fruit flies (Fisher et al., 2019; Kim et al., 2019) provide convincing evidence for it. To illustrate this point explicitly, we have added the below diagram as an Extended Data figure of the revised manuscript:

Extended Data Figure 18. *The role of plasticity in stabilizing the internal HD representation during darkness. To explain drift dynamics that we observed in darkness, we propose a model that incorporates a ‘sensorimotor-by-HD’ layer that represents a cortical consensus about the directional sensory experience. Each neuron in this layer synapses onto all HD neurons via plastic synapses. Depending on the duration of exposure to the shifted cue context, the network either has enough time (i.e. 2-minute case) to form new associations between neurons of the HD and sensorimotor-by-HD layers which results in the emergence of a new steady state and no reversion to baseline, or not enough time (i.e. 20-second case) and so, baseline associations between the two layers are maintained which causes the internal HD representation to revert to baseline configuration.*

Furthermore, simulations of the above model replicate the observations as can be seen in the following figure which we also included as Extended Data figure 19 (details of the model have been added to the supplementary material):

Extended Data Figure 19. Model simulation of reversion. **a.** Synaptic weight matrix linking the HD layer to the Sensorimotor-by-HD layer, during baseline. **b.** Simulations of representational drifts in 20-second (bottom row), 2-minute (middle row) and 4-minute (top row) exposures to the reset context. Behavior for individual examples (i.e. head angular velocity) is shared across scenarios and is taken from actual recordings. **c.** Synaptic weight matrices in darkness for the three scenarios showing the strengthening of new associations between HD and Sensorimotor -by-HD layers while baseline connections become weaker with increased duration of exposure to the reset context. **d.** Mean drifts (solid lines) in darkness across scenarios shaded areas indicate SEM.

We have added the following text to describe these new analyses:

“These results indicate that the internal representation of the baseline allocentric reference frame is not entirely lost after a reset and can still influence the HD network in darkness, depending on the duration of experience within the competing reset reference frame context. This could suggest plastic processes in the HD network. Indeed, we showed that by adding Hebbian learning to our model, HD neurons could form new associations with the unchanged allocentric cues, depending on the duration of exposure to the reset context. Given enough time, the synaptic strength of these new associations increased while old associations were depleted, resulting in a new steady state for the HD system. In this scenario, our simulations of the internal HD representation showed limited baseline attraction. On the other hand, following short exposures to the reset context, the synaptic weights did not change significantly, and baseline associations remained dominant which resulted in strong reversions, in our simulation data (Extended Data Figs. 18, 19).”

Further motivation and explanation for the plasticity involvement is added in the section “4-Plasticity” in the supplementary material which reads in its first two paragraphs:

“Our experimental data demonstrated that the duration of an animal’s exposure to a shifted-cue context predicted the drift behavior of the internal HD representation, in a subsequent darkness. The time-dependent nature of these observations led us to hypothesize that plasticity could explain the variability in representational drift patterns. Previous theoretical works have already speculated about plasticity involvement in the HD network^{27-29,31}. Indeed, computational models show that, through Hebbian learning, long-term associations between HD neurons and ‘landmark cells’ – which convey information about sensory cues – could form, thus leading to a successful integration of combined self- motion cues and sensory perception of landmarks, and a stable internal representation of the directional experience within the environment. Though there is no direct experimental evidence for plasticity in the mammalian HD system, research in fruit flies^{30,31} has provided key insights into the neural mechanisms that underlie the formation of stable internal HD representations and convincing evidence for the involvement of plastic processes in the HD network. Specifically, a type of cells called ‘ring neurons’^{30,31,45} may fit the role of ‘landmark cells’. Each of these cells synapses onto all compass

neurons (equivalent to HD cells in mammals). More so, these synapses are suggested to be plastic which allows the formation of long-term associations between compass neurons and visual scenes. In our model, we propose an additional layer that we refer to as the ‘sensorimotor-by-HD’ layer and composed of neurons that are akin to ring neurons. Each sensorimotor-by-HD cell synapses onto all neurons of the HD layer via plastic synapses (Model Fig. 5). Through Hebbian learning, new associations can be formed between the two layers depending on the animal’s experience within the environment. The sensorimotor-by-HD layer reflects a consensus about the current sensory experience, and we hypothesize that it can only sustain movements of a unique bump of activity. This means that the emergence of activity on this layer is likely mediated by a (cortical) attractor network that operates in concert with the main attractor network. However, this hypothesis is out of the scope of this work.”

Discussion: there were some areas that could be clarified, as follows.

The summary paragraph of the main findings omitted the interesting drop in network gain with reorientation, although this is discussed later.

Thank you for this suggestion. We have revised the text accordingly:

“We showed that controlled manipulations of a visual cue induced global fluctuations in network activity that could be captured by a measure we termed network gain. We demonstrated that the network gain represents a functionally critical dimension in the internal HD representation, in addition to its classically-appreciated angular dimension. Tracking this measure across repeated instances of transitions from a darkness period to a shifted-cue display event revealed a persistent pattern of a transient decline before stabilization around the baseline amplitude level. The magnitude of these drops was correlated with the speed of the HD network’s realignment with the rotated visual reference frame (i.e. reset). By extending a standard model of the HD system⁴⁴ to incorporate a variable network gain, we were able to predict the pattern of the reset response.”

“informatively linked to future network dynamics” – I wasn’t clear what this meant

The phrase referred to the predictive power of tracking network gain with regards to future reset response. To avoid the confusion, we have removed this sentence and replaced it with the following (note that this text also addressed the prior comment):

“The magnitude of these drops was correlated with the speed of the HD network’s realignment with the rotated visual reference frame (i.e. reset). By extending a standard model of the HD system⁴⁴ to incorporate a variable network gain, we were able to predict the pattern of the reset response.”

I’m not sure I agree with “network gain landscape can maintain a memory trace of a stable reference frame” as I don’t feel that this was demonstrated. Insofar as there were traces in the gain landscape, these were of the missing visual cue, but this is not necessarily a reference frame. The stable reference frame defining the baseline state of the system is surely preserved by the persisting static environmental cues, not memory - because where would the memory be held, and how, given the movements of the animal? Are the authors proposing the superposition of two simultaneous reference frames in the network itself, both continuously updated by path integration? This seems unlikely to me. If that’s what they *are* proposing it needs better explanation and some defense. Surely the simplest explanation is that the network has inputs from two sets of cues, static background cues that remain fixed in the room reference frame and a dynamic visual cue – when these are dissociated the attractor is excited at two places. The increased gain when the rat faces the removed cue is likely due, as mentioned earlier, to cue- potentiated enhancement of the other co-existent inputs at that orientation.

We agree that the use of the phrase “memory of a stable reference frame” can be confusing. We replaced the phrase in the main text with the following:

“We also showed that network gain was not only a simple product of ongoing sensory experience, but rather dynamically reflected the past experience of the system: a polarizing visual landmark could induce persistent distortions in the network gain profile, forming a ‘memory trace’ of said landmark even after the landmark was removed. Furthermore, these changes in network gain patterns were dependent on the duration of prior shifted-cue exposure, suggestive of plastic processes in the HD network.”

We believe that the reviewer’s suggested interpretation and the possibility of the attractor being excited at two places is very similar to how our model was designed, which produces simulations very close to the experimental observations (Extended Data Figure 19).

“This suggests that the visual flow can be used by the HD system to recalibrate the integration of angular velocity information (i.e. vestibular input) in order to anchor the internal HD representation to a dynamic visual reference frame”. This was an interesting observation, consistent with other recent reports in the spatial literature of dynamic neuron recalibration, but I slightly take issue with “in order to anchor...” as this is making interpretations that go beyond the data. Indeed, I think it’s very unlikely that this would be the adaptive function of this process because when, in nature, would an animal need to anchor its internal compass to a rotating world? It seems more likely that the adaptive reason for this process is the reverse: it is to use stable landmarks to calibrate the vestibular system to compensate for fluctuations in its gain, perhaps due to factors like changes in endolymph density. In any case, whatever the speculation, I suggest it be reserved for the discussion.

We agree with the reviewer’s interpretation. Indeed, it seems that what we are observing is rather an indication the HD system is using the most salient cue (i.e., visual) to calibrate the vestibular input integration, which cannot be generalized beyond the context of the experiment as it might have been inadvertently implied in the original statement. This phrase has been removed in the revised text. We thank the reviewer for the insight.

“realign” should be “realigning”

We thank the reviewer for suggesting this improved wording. This is now fixed.

“strong influence of the internal representation of the baseline reference frame” – as above, I don’t think we can conclude that it’s internal; surely it must be external. That is, due to olfactory and other static environment cues, linked to the network, that help drive it when the visual cue starts to get near their range of influence and thus pull the network back towards that baseline. As mentioned above, if the authors want to argue that it is not due to persistent environmental cues they would need to come up with some explanation for how two reference frames could be maintained and constantly updated in the one network, and why they would sometimes interact (when the frame orientated by the visual cue gets near the frame defined by the baseline state.)

We understand the reviewer’s concern about our interpretation of these results. We consider that baseline attraction following cue rotation is a further evidence of the experience-dependent aspect of internal representation’s drift patterns for which plasticity provides a plausible and fitting explanation. To better illustrate our points, we have performed additional neural network simulations and included these new analyses in the main text as an Extended Data figure and in supplementary material. These simulations replicate the observations based on the idea of Hebbian learning between the HD layer and the sensorimotor-by-HD layer, shown below:

Extended Data Figure 21. *Model simulation of vestibular input recalibration by visual experience. a.* Synaptic weight matrix linking the HD layer to the Sensorimotor-by-HD layer, during baseline. **b.** Simulations of offset traces during cue rotation (7 minutes) and in subsequent darkness (4 minutes) for the fast (3°/s) and slow (1.5°/s; 1.28°/s) cases. Behavior for individual examples (i.e. head angular velocity) is shared across scenarios and is taken from actual recordings. Sessions without vestibular input recalibration (i.e. vestibular angular velocity neurons do not receive input from the bias cells – see model details in Supplementary Material) for both 3°/s and 1.5°/s cases were used as test examples. The 1.28°/s cue-rotation sessions were used to show the effect of cue rotation speed on drift biases regardless of offset proximity to baseline condition. **c.** Synaptic weight matrices at the beginning of the darkness phase for the fast (3°/s) and slow (1.5°/s) scenarios showing that baseline associations remain dominant even after 7 minutes of cue rotation which explains the stabilization around the 0°-offset line. **d.** Mean drifts (solid lines) in darkness across scenarios shaded areas indicate SEM.

Furthermore, we have made the following change to the main text:

“We also observed an attraction to the baseline internal representation, similar to what we observed in prior experiments. Here, the system starts to stabilize once the internal HD representation comes close to realigning with baseline reference frame (Fig. 5e, Extended Data Fig. 20). This phenomenon could also be reproduced in our model where, after 7 minutes of cue rotation, no new associations between HD neurons and allocentric cues could emerge to form a new steady state. Instead, baseline associations remained dominant albeit with a significant weight decay in the synaptic matrix (Extended Data Fig. 21c). Together, these results indicate that experience with dynamic reference frames can also bias the HD network, and implicate asymmetric integration of vestibular information within the HD network as a potential source of this bias.”

P16 typo “–grid ells”

This is now corrected.

P25 Where the variables are explained, N should be listed here as well. Also, perhaps combine HD network simulation and Attractor network model as there doesn't seem a good reason to split these.

We thank the reviewer for the suggestion. We have incorporated these suggested changes.

P33 I am slightly uneasy this arbitrary division of the data into fast and slow, and the comparison of a factors (AHV) that a priori wouldn't have been predicted to co-vary like this. It feels a little like data mining, although I appreciate that many findings in this field were not predicted a priori. One thing that could help is an internal replication: i.e., to show that the effect is present separately in all three mice. Incidentally, here and elsewhere, the y axes should start at zero so as not to visually inflate effects.

The data is divided in the same way as in main Fig. 2 using k-means clustering to separate resets between fast and slow. This separation is blind to AHV. The goal is to show that the observed gain drops cannot be explained by AHV modulation and that gain decline is likely caused by an extra- network global inhibition.

To address the reviewer's point further, in Extended Data Figure 7, we show the separation of the data between fast and slow for each mouse.

P34 The drift speed analysis doesn't seem right to me. The authors have picked an arbitrary time point, 5s, at which to compare the deg/s drift rate but it is evident from the first graph that the rate is an exponential function of time, so the best comparison would have been the exponential time constant. What was the take-home message from this analysis? In part c I didn't quite understand “Each dot represents a correct reset” since shouldn't the resets have been at 90 deg (or multiples)?

We understand the reviewer's concern and we would like to provide further explanation for the way the data was presented. A correct reset does not have to be a multiple of 90°. Because of the cumulated drift, in darkness right before the shifted-cue display, the internal HD representation can be at any offset from the shifted visual reference frame. The 5s period was chosen to make sure that the comparison of gain between mid- and long-range resets was pre-stabilization (i.e. transient regime). This figure puts emphasis on range and its relationship to network gain. The speed comparison was only to show that speed cannot explain the range-dependent variability in network gain fluctuations.

The take home message is that there appears to be a variability in gain decline, following shifted-cue display, linked to the estimated amount of phase error between the internal representation and the visual reference frame which appears to be independent of the variability in gain decline related to drift speed modulation.

P36 Figure part a – Drift variability: I'm not sure why attention is on the variability (size of error bars) rather than the amount of drift. The error scales with the amount so this seems slightly

trivial. The more important observations here are (i) that the drift increases across epochs (maybe due to decreased cue control with mismatch learning?) and (ii) that the direction is towards baseline in D2 and D4.

Thank you for this comment. This comparison is meant to show that, unless there is a prior cue manipulation (i.e D2, D3 and D4), drift remains in close proximity to baseline (D1). Though the representation becomes less stable and drift variability increases initially, following cue removal, drift fluctuations are maintained within a ~constant STD for the remainder of D1, which is not the case for the other darkness periods.

P38 I suggest to rotate the heat plots 90 deg so that the x axes align with the line plots underneath.

We agree and have incorporated this suggestion.

P39 “2m-cue-exposure” should be “2min-cue-exposure” and “(same data as in e)” should (I think) be “(same data as in b)”

We agree and have made the suggested changes.

P44 The description of the inhibition-based attractor should mention and reference Song and Wang (<https://pubmed.ncbi.nlm.nih.gov/15673682/>) who were the first (I believe) to show that a ring attractor could work this way.

We agree and now cite this reference. Thank you for this suggestion.

Signed: Kate Jeffery

Referee #2 (Remarks to the Author):

This paper explores the dynamics of the head-direction (HD) representation in neurons imaged from the anterodorsal thalamus of freely behaving mice, following various manipulations of a prominent visual cue in the environment. The first manipulation consists of alternating epochs of light and darkness, where in different light periods the cue is presented at different locations on a screen that surrounds the animal. It is well known that the HD representation is “anchored” to the cue, i.e. if the cue is moved from one side of the room to another, the HD representation rotates accordingly – in other words, the mouse HD signal is referenced to the location of the cue. The first question asked by the authors is how the dynamics of this anchoring process unfold. They identify the population activity gain – the strength of the HD representation essentially – as an important factor contributing to these dynamics. When the gain is low (high), the HD signals anchors quickly (slowly). This phenomenon can be accounted for by a semi-standard neural network model of the HD signal, where the gain is realized by an external input. In my understanding, when the gain input is low, the external visual inputs to the network dominate the recurrent inputs, resulting in faster anchoring. The second question is what happens during the

dark epochs. Analysis of the neural dynamics reveals that the population gain is higher in correspondence of the previous location of the cue. This striking observation could potentially underly the very slow anchoring dynamics observed during dark epochs. In other words, despite the absence of visual cues, the HD representation could maintain a memory of a past cue location in the form a location-dependent gain modulation. In a second set of cue manipulations, the authors zoom into the memory phenomenon revealed by the first experiment, where the HD representation appears to slowly anchor to a visual cue that is no longer present.

To better characterize this phenomenon, the authors perform a new experiment where the light periods (except the first) are of shorter duration. In this case, the anchoring to new cue locations is only transient, and in the absence of cues the representation anchors back to the location of the cue during the first light period. A dynamical systems-based analysis of the population activity is consistent with this relaxation phenomenon. Lastly, the authors wonder whether a slowly rotating cue during the light period would not only continuously anchor the HD representation, but also similarly affect its dynamics during the subsequent dark period. Strikingly, this was indeed the case.

The HD system is of fundamental importance for spatial navigation, and the analysis of large-scale imaging of thalamic activity in this contribution reveals novel, intriguing neural correlates that might underlie, for instance, memory guided goal directed behavior. In my opinion, the topic is of potential interest to a broad audience.

However, this reviewer found the ms difficult to understand. This is likely due, in no particular order, to the (i) large amount of different analysis techniques and modeling employed in the paper, which might not always lead towards a crisp main message (ii) the enumeration of several phenomena, as somewhat implied by the generic Title, without the guidance that could provide the “mechanistic insights” promised by the authors in the Abstract (iii) potentially lacking or confusing description/use of core concepts, methods, terminology, and interpretation of the results. Hence, it is challenging to evaluate the strength of some of the main results.

The authors could consider the suggestions below, which will hopefully clarify the point of view expressed above. I am ordering the comments according to figure numbers, highlighting what in my mind are the points that could most benefit from clarifications/additional modeling/additional analysis/simplification.

We would like to thank the reviewer for their positive assessment and the constructive feedback on our work. We have revised the manuscript substantially according to the suggestions. Detailed responses to individual critiques are provided below.

Fig 1:

- 1) The single cell tuning analysis in Fig 1c assumes that neurons have nice unimodal tuning curves. This might not be the case (as suggested by the off-diagonal peaks in 1g, and it has been shown that it's not the case with analysis of ephys recordings from Peyrache and collaborators for instance). The resulting correlation between data and prediction of this model (1f) could perhaps be improved by relaxing this assumption.

We agree that, in the previous version of the manuscript, Fig 1c could inadvertently lead readers to think that all recorded HD neurons had stereotypical tuning curves. The examples in Fig. 1c were, in fact, selected on purpose to show the HD cells with the strongest HD tuning. The goal was to demonstrate that the tuning curves of deconvolved spiking activity have the characteristic shape that

we would expect to see in electrophysiological recording of HD neurons. We do indeed see a variability in directional tuning as shown in 1g and 1f (where directional tuning strength is measured by the correlation coefficient). To address this point, we changed the figure legend which now mentions: “Example tuning curves of ADN cells with high directional tuning in polar coordinates. Red lines and numbers show the Mean Resultant Vectors (MRV) and preferred firing direction (PFD), respectively.”

We also added an Extended Data Figure 2 (shown below) with all ADN cells recorded from each mouse, in an example baseline recording of 10 minutes:

Extended Data Figure 2. Polar tuning curves of ADN neurons from a 10-minute baseline recording for each mouse (total number of neurons = 502). The directional tuning of each ADN neuron is shown by the correlation coefficients above each tuning curve.

To address the reviewer’s second point: the correlation analysis does not assume a unimodal tuning curve but rather measures the tuning strength of ADN neurons to the direction of peak firing rate in the horizontal plane which resulted in an overwhelming majority of neurons (>90% in all mice) with

significant directional tuning. Because of that and to eliminate selection biases, in all subsequent analyses (including HD decoding, dimensionality reduction, gain measurement, etc...) we used the firing activity from all ADN neurons.

To avoid this potential confusion, we added the following text under ‘HD decoding from neural data’ in the methods section:

“We note that, because of the predominance of HD tuned neurons among detected cell segments and to avoid selection biases, the neural activity from all ADN cells was used as an input to the decoding algorithm.”

2) Major: I think that the single cell analysis above, the Bayesian decoding analysis (1i), and the feedforward neural network analysis (1h) assume that the ADN dynamics is low dimensional – at most 2d for the analysis in 1h. Why not show this explicitly? It could be that some of the intriguing dynamics described in the subsequent figures is correlated with changes along other dimensions in neural space. It could be helpful to show that it’s not the case.

The reviewer raises an important and interesting question about the dimensionality of the HD representation. We understand the reviewer’s point of view and would like to provide the motivation for the way the data was presented.

One of the objectives of the current study is to show the multidimensional aspect of the internal HD representation in the rodent brain. As we discuss in the methods section, the dimensionality of this internal construct could be higher than 2, however, we decided to limit the analysis to the 2D case for the sake of simplicity. Our analyses show that the secondary dimension is highly correlated with network gain which indicates that fluctuations of this quantity dominate the variance that cannot be explained by the angular dimension alone.

The goal of our study is to show that the dimensionality is higher than 1 (in contrast with how the HD representation has long been regarded and analyzed). For that, we provided evidence that there is at least one more dimension that is functionally critical. Additional dimensions could exist, yet we feel that a systematic investigation of this important question is beyond the current work.

3) Fig 1h: Why is there no dispersion along the radial dimension here? As shown later angular velocity correlates with the gain/radius, one would expect the that the angular velocity would play a role here.

The reviewer raises a valid point regarding the effect of angular velocity on the variability along the radial dimension. The angular velocity correlates indeed with some of the variability along the radial component. However, this variability is still limited within a close range of baseline amplitude. This is in contrast with the substantial variability observed following resets, Fig. 2. This result further confirms that the internal representation can be fairly approximated by a unidimensional ring manifold, in stable conditions, and that the variability is predominantly captured by the angular dimension.

Nevertheless, the reviewer is right to point out the lack of dispersion along the radial dimension. Our analyses (see figure below) show that although the radius is modulated by angular velocity, the variability in radius amplitude appears to scale down as we move from the inner to outer circles of the attractor manifold. We believe this is an artifact of the nonlinearities introduced by the neural network. Note that this effect does not impact the pattern of AV modulation of neural activity which is similar to what we see for network gain.

Reviewer Figure 2: Modulation of network activity by head angular velocity. Each trace is the mean across mice and sessions. aAHV: absolute angular head velocity. Note that there is dispersion along the radial/gain dimensions, and that the reviewer is correct that these correlate with angular velocity. However, while radius and gain follow the same patterns, the range of AV modulation of the radius is much smaller than that of network gain, especially during baseline. Note that the feedforward neural network used for dimensionality reduction uses ‘tanh’ and ‘relu’ activation functions which could explain the decreased range of variability due to the nonlinearities in the computation.

We notice that, during baseline, the radius starts decreasing at higher AHVs, while it saturates, during darkness. We confirm that these patterns are not due to the neural network by showing that they also exist in network gain (Reviewer Figure 2). Previous studies have also shown similar patterns of firing in angular velocity cells – which provide input to HD neurons – for high AHVs (Bassett & Taube, 2001; Keshavarzi et al., 2022; Turner-Evans et al., 2017).

Fig2:

Major: The analysis in Fig 1h is used as a motivation for the experiment in Fig2 (beginning of section “Network gain covaries with resetting dynamics during cue manipulation”). The intuition provided in the text relies on the knowledge of the energy in e.g. continuous attractor networks, but this intuition is never unpacked explicitly and could be confusing without additional clarification. One option could be to elaborate more on this in the text (perhaps with accompanying schematic panels), but a simpler/more conservative option could be to remove the ff neural network-based analysis from the main text. After all, the radial dimension in this polar representation is only used transiently, and immediately related to the gain of the population activity in Fig 2d,e. The latter appears to be a more intuitive description of the additional dimension.

We understand the reviewer’s concern about the incorporation of the state space analyses in the main text. We would like to provide the following explanation for the rationale behind it.

The key concept in the state-space analysis is the existence of multiple energy levels (total amount of neural network activity) in the internal HD representation. Our hypothesis is that movements between

angle-states become faster at low energy levels because of the decreasing distance between the said angle-states (see schematic below).

The idea of a multidimensional internal HD representation is central to the current study and we think the dimensionality reduction of the high-dimensional input data using a feed-forward neural network provides a visual illustration that may help the reader understand the intuition behind expanding the investigation of the HD system along a secondary (radial) dimension (e.g. decreasing distances between angular states closer to the center, shortcuts between distant states, etc...). We think the state-space analysis also provides a compelling pretext for the calculation and use of network gain as a variable of interest in the study of the HD system during reset and drift because of the high correlation between state-radius and gain. For this reason, we think that it could be better to talk about the dimensionality reduction in the main text. However, we agree that better wording is needed and that a schematic could provide further illustration of the intuition behind the analysis.

To make the point explicitly, we have added the following paragraph to the main text:

“To visualize the potentially low-dimensional structure of the representational space of the head direction network, we developed a method to project the high dimensional input data from the neural space onto a 2-dimensional polar state-space (Methods, Dimensionality reduction using feedforward neural network). This method involves training a deep neural network on the measured head direction while allowing an untrained latent variable to capture variability in the neural data that cannot be explained by changes in the head direction alone. This inferred latent variable constitutes the radial component in the 2-dimensional polar state-space (i.e. secondary dimension). When applied to the baseline data, we obtain a ring-like structure (Fig. 1h), reminiscent of both attractor HD network models and prior analyses^{3,17,42}. This further confirms that, in stable conditions, the internal HD representation is indeed approximately unidimensional.”

In addition, we have added a new schematic figure (**Extended Data Figure 5**; see below) to better explain the intuition behind our idea.

Extended Data Figure 5. Schematic showing the dynamics of changes in the internal HD representation from population-activity and state-space perspectives. **a, b.** The amount of the change of neural activity in the activity during bump movement depends on the gain of the network. Here, the x-axis represents the neuronal space (i.e., neurons sorted according to the preferred head directions, assuming a uniform distribution of HD cells according to their preferred firing directions). Mathematically, the distance between representations of the internal HD from start to end of a

rotation, in the Euclidean sense, is smaller at lower network gain. D: Euclidean distance, $\mathbf{r}^{\text{activity}}$: $N \times 1$ vector of firing rates from N HD neurons composing the HD network, at time t for different activity levels (i.e. high or low). c. The concept of decreasing distance between internal HD representations, at lower network gain, is naturally captured in the 2D polar coordinate system if we assume that the radius reflects the level of neural network activity. The distance traveled in the hypothetical state-space of the HD network will be greater when the radius is larger as well as when the net gain is higher, which could be quantified by the total change of the firing rate across the network. Thus, we hypothesize that the radius is correlated with the overall population activity (i.e. network gain) and that decreasing distance facilitates rotations across the HD network. Assuming that the internal HD representation lives in a 2D polar state space where each state is defined by a phase and a radius, state transitions would be fastest at the lower end of the radial component because of the decreasing distance between states representing different angles, near the center of the baseline ring. Bar graphs are only indicative and not to scale.

2) Major: The term drift used to describe the relationship between the internal HD signal and the external visual cue could be confusing. This quantity is for instance referred to as “offset” in the literature (see e.g. references to the drosophila HD system used in the ms). I think that drift typically refers to a component of the dynamics (equation of motion) of so-called drift-diffusion processes, where the drift corresponds to the deterministic component that drives the anchoring, and the diffusion to the noisy component (described as standard deviation of the drift in a subsequent figure). If the authors agree with these definitions, amending the current terminology in the ms would be of help.

We thank the reviewer for this suggestion. We agree and have made the suggested changes. We now use the word ‘offset’ to refer to the mismatch (i.e. angular difference) between the measured and decoded HD while we kept the use of the word ‘drift’ to describe the instability in the internal HD representation, akin to that of the drift-diffusion process.

3) Major: The drift (or offset) as defined in the ms is a rather abstract concept. I think that showing what happens to the HD representations during a single session (as shown for instance in the bottom panels of Fig 5a,b) earlier in the manuscript, to illustrate what the authors mean by drift, would go a long way toward making the ms more interpretable.

We thank the reviewer for his suggestion. We agree and have now added the panel below to main figure 2a.

Main Figure 2a: Network gain covaries with resetting dynamics. *a. Top row: Experimental protocol. Middle row: Example session showing raw (blue dots) and smoothed (red line) offset traces as obtained by subtracting the decoded HD from the measured HD. Darkness periods are shaded in dark gray. Bottom row: Traces of measured HD (blue) from the behavioral camera and decoded HD (green) from neural activity.*

4) **Major:** the model beautifully shows how a simple non HD-specific external input can recapitulate the anchoring dynamics during light epochs. However, the model does not take into account the fact that (i) the animal is moving during the trial – hence the visual input is changing, and the angular velocity of the animal is being integrated by the model network. This becomes particularly relevant when the authors shows that the angular velocity just preceding the cue was a good predictor of fast/slow resets (EDF 7) and that the anchoring dynamics depended on the distance between pre-cue and final drift (EDF 8). Why choose to only model one aspect of these rich dynamics? I thought that the model was a very valuable addition, in that it helped with providing “mechanistic insight”. Extending the model to capture the additional relevant variables of the problem (heading of the animal, location of the cue, angular velocity) would be illuminating. Moreover, the model is unfortunately abandoned for the rest of the paper- relying on it throughout could really help to contextualize all the interesting phenomena reported in the ms and provide insights into how these phenomena “work”.

We thank the reviewer for this comment. We agree that a model could provide a better interpretation of the potential mechanisms underlying the phenomena reported in this study and set the theoretical framework to test most of the hypotheses put forward, in the text. For that, we have extended the network model, substantially, to cover additional phenomena that we observed throughout the different experiments, including drift behavior in darkness, in both the short and long shifted-cue exposures, as well as in the continuous cue rotation scenarios. For details, please see the answers to the comments below.

To address the reviewer’s comment on the integration of further variables in model. We agree that the phenomena would always be better accounted for with a more detailed model that incorporates the visual changes that accompany movement through the space. However, such a model would also require us to make many additional assumptions about how angular velocity and non-uniform visual inputs

contribute individually to the measured network gain which we can only broadly speculate about and might obscure the explanatory power of the current model, specifically for the reset-response prediction. This is because the measured network gain is likely to reflect the influence of multiple inputs, including but not limited to angular head velocity and visual input. Indeed, as we show in Fig. 3e,c (main text) and in Reviewer Fig. 2 (above), gain is modulated by both angular head velocity and light condition. However, the effects of such modulations remain limited in comparison with the substantial decline in gain that we observe during resets. As such, we have to account for a possibly unobservable source of modulation which adds to the speculations and diverts this part of the model (i.e. reset-response prediction) from its main objective, which is to show that network gain, taken as a whole and as an uniform external input to all neurons of the HD layer, allows accurate prediction of the reset response when applied to a standard network model of the HD system.

Therefore, while we agree in principle with the reviewer's comment, we believe that the current reduced model for reset-response prediction is better suited for illustrating the network gain modulation and its impact on the reset response. That being said, since network gain is not of primordial importance in subsequent sections of the model (i.e. for drift prediction in darkness), the measured angular head velocity was integrated as an input to the HD network, which adds to the variability in the simulated drifts.

Fig 3:

Major: The focus now shifts on diffusion of the offset during the dark period. Fig 3d is one of the most striking results in the paper in my opinion, also emphasized by the author statement: "To our knowledge, this is the first evidence of such long-term experience-dependent preferential firing in the thalamic HD cells". It would be great if "internal HD" here would be defined as precisely as possible, perhaps describing this more explicitly or depicting it with a time-course of populations activity across conditions, where both the offset re-centering towards the cue and the respective amplitude tuning, can be observed. My understanding of this result is complicated by a description of what happens in the main text: "The HD neurons that fired when the animal was facing the cue during baseline have higher average firing rates during darkness, following a reset", as opposed to what it is said in Methods ("decoded HD" in the "gain heatmap analysis" section). Is it possible that with one of these definitions the internal HD is defined in a such a way that "neurons that fired when the animal was facing the cue during baseline" continue to fire when the animal is facing the new cue location due to the offset tethering (resets and drifts in dark), and they do so with higher firing rate than the other neurons? The results shown in Figure 3d would then mean that the amplitude tuning encodes the cue position estimate, and that tuning emerges and is tightened at the later periods (ED Figure 11). Does this tuning exist only during the dark periods? How do the gain profiles from the light periods preceding the D2-D4 periods look like? If the tuning is only present during the dark periods, could you comment on why it appears later, is this somehow related to the drift? Are there similar gain profiles during the 'bright 'and 'dark 'periods of the optic flow experiments where there are drifts?

We thank the reviewer for this comment. We understand the confusion that the previous wording might have caused. We would like to confirm that we used the same definition of internal HD throughout the manuscript, unless specifically mentioned (such as in Fig2 d-right; EDF 8a-bottom row; EDF 6; where HD is shown in the internal (egocentric) reference frame through circular shifts of the bump of activity by the instantaneous amount of head rotation).

Internal HD is simply the output of the HD decoder as shown in Fig. 1i. On the other hand, measured HD is defined within an allocentric reference frame, with 0° corresponds to the direction of

the initial cue location. During baseline, both measured and decoded HDs overlap, which means that cells with PFDs $\sim 0^\circ$ are those that become active when the animal is facing the cue. By definition, a reset corresponds to the rotation of the HD network with the amount of visual-cue shift, meaning that after a 90° -cue rotation, if the animal faces the new cue location (i.e. $\sim 90^\circ$), HD neurons of PFDs $\sim 0^\circ$ are now active. In other words, each time the animal is facing the cue (regardless of its position in the allocentric reference frame), it is the same group of neurons (i.e. PFD $\sim 0^\circ$) that is active. The activity packet formed by these 0° -PFD neurons is what we refer to as the internal cue location. These are the same neurons that fired during baseline whenever the animal faced the initial cue location (i.e. 0°). Therefore, we confirm the reviewer’s statement that “the results shown in Figure 3d would then mean that the amplitude tuning encodes the cue position estimate”, however, we want to make emphasize that this internal cue position (i.e. 0° in internal HD) shifts w.r.t to the baseline allocentric reference frame. To avoid unnecessary confusion in the main text, the phrase “*The HD neurons that fired when the animal was facing the cue during baseline have higher average firing rates during darkness, following a reset*” was removed.

The reviewer raises an important question about the gain tuning in light conditions preceding darkness. We have added the Extended Data figure (see below) to the manuscript to show that the tuning is a characteristic of darkness periods.

Extended Data Figure 14. Event-dependent changes in the network gain tuning curve. **a.** Average gain tuning curves across light conditions. **b.** Average gain tuning curves across darkness conditions showing gradual decrease of network gain away from the internal cue location (dashed yellow line) from D1 to D4. Tuning curves are shown as mean (solid line) and SEM (shaded area) and bar graphs indicate mean \pm SEM.

We believe that the gain tuning becomes narrower in later darkness periods because directional cues away from the visual landmark become less and less reliable following multiple resets. The decrease in gain could reflect a decrease in certainty in the HD encoding of cue-less directions.

The reviewer also asks a question about gain patterns in darkness periods following cue rotation (Fig. 5). We show below the gain landscapes for both cue rotation speeds (1.5°/s (slow) and 3°/s (fast)) during cue rotation (Cue-rot) and during darkness (D2).

Reviewer Figure 3. Gain landscapes for the cue rotation experiment. Upper row: Slow cue rotation experiment ($1.5^\circ/\text{s}$). Bottom row: Fast cue rotation experiment ($3^\circ/\text{s}$). Gain landscapes are shown during cue rotation events (Cue-rot) and during the following darkness (D2).

We are yet to find an insightful explanation other than that the gain landscapes become more distorted following faster cue rotation. Note that our analyses yielded no predictive power of these gain patterns with regards to drift behaviour.

Fig 4:

Why is the protocol used in this experiments (cue at 0, -90, 90, -90, 90, 0) different what the one used before (cue at 0, 90, 180, -90, 0)?

This is an important question raised by the reviewer regarding the rationale behind the experimental protocol. In the previous section, we observed evidence for both an attraction of the internal HD representation towards baseline as well as a drift bias in the direction of cue rotation. In this section, we would like to test the attraction hypothesis. We alternated the cue shifts between $\pm 90^\circ$ to avoid any possible bias that could be induced by the $0^\circ, 90^\circ, 180^\circ, -90^\circ$ pattern, and because there is an inherent ambiguity in the direction of baseline attraction in the 180° cue case.

2) Major: 4a, Why does the offset goes to 90 degrees in the presence of a cue at -90 degrees (as opposed to Fig 2)?

There was a typo in the figure. Thank you for pointing this out. Indeed, the experimental protocol starts with a $+90^\circ$ cue shift rather than a -90° cue shift. This is now corrected.

3) Major: 4d. This is a new analysis technique, showing that the drift dynamics should converge to fixed points during the dark period (why not only one at 0 degrees btw, is there a prediction that can be made there?). Isn't this result somewhat expected from the dynamics of the drift shown in Fig 4a – could there have been alternative pictures? In addition, this dynamical systems description ignores other relevant variables, such as the gain shown in the other panels. Should one thing about this flow field as being dependent on additional variables? If so, why ignore them?

Also, as mentioned above, a network model of these phenomena could be more illuminating. Similar comments apply to Fig 5.

These questions regarding the dynamical landscape of drifts are indeed important ones. Regarding the bifurcation of fixed points, this is due to the combination of CW and CCW cue-shift events. The resulting dynamical plot is indeed expected from Fig. 4a-c, however, we would like to point out that there is a subtle difference: While Fig. 4a-c show the average time-dependent drift pattern, Fig. 4d shows, at each point of the grid, the mean acceleration given the drift angle and the drift speed. As such, the time-varying aspect of drift is omitted here. This figure was meant to illustrate the predictability of drift patterns and the attractive effect of the baseline configuration. This is to be contrasted with Fig. 5e where we observe different dynamics. The intention behind this presentation is to highlight the perspective that the HD system can be thought of as a dynamical system where the dynamics change depending on prior experience. This could motivate further model developments that take into account the dynamical aspect to disentangle different influences on the stabilization of the internal HD representation.

The reviewer also asks a question about the dependence of the drift patterns on other variables such as gain. Our analyses did not show direct correlations between the two variables (drift and gain bumps). We show this in Extended Data figures 15b, 16a, 17b. The relationship between them remains unclear to us.

The reviewer suggests further development to the model to explain other observed phenomena. We thank them for this comment. We totally agree that this could be illuminating in many ways and would benefit readers who may be interested in testing some of the proposed hypotheses, in future experiments.

Below are figures that summarize the simulation results (included as Extended Data figures) as well as a detailed description of the additions to the original model (included as supplementary information). Further descriptions and references to the model are added to the main text.

First, for the 20sec cue-shift experiment:

Extended Data Figure 18. *The role of plasticity in stabilizing the internal HD representation during darkness. To explain drift dynamics that we observed in darkness, we propose a model that incorporates a ‘sensorimotor-by-HD’ layer that represents a cortical consensus about the directional sensory experience. Each neuron in this layer synapses onto all HD neurons via plastic synapses. Depending on the duration of exposure to the shifted cue context, the network either has enough time (i.e. 2-minute case) to form new associations between neurons of the HD and sensorimotor-by-HD layers which results in the emergence of a new steady state and no reversion to baseline, or not enough time (i.e. 20-second case) and so, baseline associations between the two layers are maintained which causes the internal HD representation to revert to baseline configuration.*

Extended Data Figure 19. *Model simulation of reversion. a. Synaptic weight matrix linking the HD layer to the Sensorimotor-by-HD layer, during baseline. b. Simulations of representational drifts in*

20-second (bottom row), 2-minute (middle row) and 4-minute (top row) exposures to the reset context. Behavior for individual examples (i.e. head angular velocity) is shared across scenarios and is taken from actual recordings. **c.** Synaptic weight matrices in darkness for the three scenarios showing the strengthening of new associations between HD and Sensorimotor -by-HD layers while baseline connections become weaker with increased duration of exposure to the reset context. **d.** Mean drifts (solid lines) in darkness across scenarios shaded areas indicate SEM.

Below is the model description as added in the supplementary information:

“4- Plasticity

Our experimental data demonstrated that the duration of an animal’s exposure to a shifted-cue context predicted the drift behavior of the internal HD representation, in a subsequent darkness. The time-dependent nature of these observations led us to hypothesize that plasticity could explain the variability in representational drift patterns. Previous theoretical works have already speculated about plasticity involvement in the HD network^{27-29,31}. Indeed, computational models show that, through Hebbian learning, long-term associations between HD neurons and ‘landmark cells’ – which convey information about sensory cues – could form, thus leading to a successful integration of combined self- motion cues and sensory perception of landmarks, and a stable internal representation of the directional experience within the environment. Though there is no direct experimental evidence for plasticity in the mammalian HD system, research in fruit flies^{30,31} has provided key insights into the neural mechanisms that underlie the formation of stable internal HD representations and convincing evidence for the involvement of plastic processes in the HD network. Specifically, a type of cells called ‘ring neurons’^{30,31,45} may fit the role of ‘landmark cells’. Each of these cells synapses onto all compass neurons (equivalent to HD cells in mammals). More so, these synapses are suggested to be plastic which allows the formation of long-term associations between compass neurons and visual scenes.

In our model, we propose an additional layer that we refer to as the ‘sensorimotor-by-HD’ layer and composed of neurons that are akin to ring neurons. Each sensorimotor-by-HD cell synapses onto all neurons of the HD layer via plastic synapses (Model Fig. 5). Through Hebbian learning, new associations can be formed between the two layers depending on the animal’s experience within the environment. The sensorimotor-by-HD layer reflects a consensus about the current sensory experience, and we hypothesize that it can only sustain movements of a unique bump of activity. This means that the emergence of activity on this layer is likely mediated by a (cortical) attractor network that operates in concert with the main attractor network. However, this hypothesis is out of the scope of this work.

Model Figure 5. Connections between the Sensorimotor-by-HD and HD layers. Projections from the most active sensorimotor-by-HD neuron only are shown. Arrows represent plastic synapses that are updated following a Hebbian rule. Width of arrows indicates synaptic strength. All shown connections are excitatory. Color gradients indicate the level of activity for each neuron (i.e., opacity increases with firing activity).

To implement plasticity in the model, we applied a Hebbian update rule – following Skaggs et al. (1994) – on synaptic weights linking each sensorimotor-by-HD cell to all HD neurons, such that:

$$\Delta w_{i \rightarrow j}(t) = \alpha (w_{\max} f_H(r^j(t)) - w_{i \rightarrow j}(t)) r^i(t) \quad (23)$$

Where, $\Delta w_{i \rightarrow j}$ is the change in synaptic weight from presynaptic sensorimotor-by-HD neuron i to postsynaptic HD neuron j , α is the learning rate, w_{\max} is the upper bound of synaptic weights, $w_{i \rightarrow j}$ is the synaptic weight from presynaptic sensorimotor-by-HD neuron i to postsynaptic HD neuron j , r^i

and r_i are the firing rates of presynaptic sensorimotor-by-HD neuron i and postsynaptic HD neuron j , respectively. f_H is a sigmoidal activation function of the form:

$$f_H(x) = \frac{c_H}{1 + e^{b_H - a_H x}} \quad (24)$$

Where, a_H , b_H and c_H are optimization parameters. After each iteration, the synaptic weights are updated as follows:

$$w_{i \rightarrow j}(t) \leftarrow w_{i \rightarrow j}(t) + \Delta w_{i \rightarrow j}(t) \quad (25)$$

Which we integrate in the total voltage of each neuron of the HD layer such that:

$$V^{\text{HD-Hebbian}}(t) = V^{\text{HD}}(t) + g_H W_{\text{sm-by-HD} \rightarrow \text{HD}} V^{\text{sm-by-HD}}(t) \quad (26)$$

Where, $V^{\text{HD-Hebbian}}$ is a $N \times 1$ vector of HD neuron voltages, V^{HD} is a $N \times 1$ vector of HD neuron voltages excluding the input from the sensorimotor-by-HD layer (see equation 12), g_H is a scaling factor, $W_{\text{sm-by-HD} \rightarrow \text{HD}}$ is a $N \times N$ synaptic weight matrix linking the sensorimotor-by-HD and HD layers, and $V^{\text{sm-by-HD}}$ is a $N \times 1$ vector of sensorimotor-by-HD neuron voltages. The activity on the sensorimotor-by-HD layer is assumed to have the same shape and amplitude as the visual layer and is updated through circular shifts by assuming a perfect integration of the vestibular input (i.e. with same rate of change of θ_{VE}).

Upon optimization, the parameters are assigned the values $\alpha = 40$, $a_H = 20$, $b_H = 15$, $c_H = 2$, and $g_H = 0.8$.

Second, for the cue rotation experiment:

Extended Data Figure 21. Model simulation of vestibular input recalibration by visual experience. *a.* Synaptic weight matrix linking the HD layer to the Sensorimotor-by-HD layer, during baseline. *b.* Simulations of offset traces (7 minutes) and in subsequent darkness (4 minutes) for the fast (3°/s) and slow (1.5°/s; 1.28°/s) cases. Behavior for individual examples (i.e. head angular velocity) is shared across scenarios and is taken from actual recordings. Sessions without vestibular input recalibration (i.e. vestibular angular velocity neurons do not receive input from the bias cells – see model details in Supplementary Material) for both 3°/s and 1.5°/s cases were used as test examples. The 1.28°/s cue-rotation sessions were used to show the effect of cue rotation speed on drift biases regardless of offset proximity to baseline condition. *c.* Synaptic weight matrices at the beginning of the darkness phase for the fast (3°/s) and slow (1.5°/s) scenarios showing that baseline associations remain dominant even after 7 minutes of cue rotation which explains the stabilization around the 0°-offset line. *d.* Mean drifts (solid lines) in darkness across scenarios shaded areas indicate SEM.

Below is the model description as added in the Supplementary Information:

“5- Vestibular input recalibration through visual feedback

Our third experiment showed that the visual experience (i.e. continuous cue rotation) can cause a persistent bias in the internal HD representation. We built upon the model in the previous section to gain further insights into the mechanisms that could lead to such behavior in the network.

To account for a flexible integration of the vestibular input, we added a recalibration circuit that adjusts the firing activity of angular velocity (AV) cells depending on external sensory inputs. Our experimental data showed that the internal HD is phase-locked to a rotating visual cue, even when the animal is motionless. The same experiment showed that the representational drift maintains an angular speed similar to the cue’s, in a subsequent darkness period. We interpret this observation as an asymmetric bias added unilaterally to the vestibular AV cells (CW or CCW). The resulting imbalance causes movements along the attractor network in the direction of increased bias, even in periods of immobility. We hypothesize that the bias signal reflects the difference between the ‘perceived’ AV and the ‘vestibular’ AV. We believe the perceived AV is a cortical construct that measures the rate of angular displacement of the bump of activity along the HD layer. Note that the bump’s movement is not only dependent on vestibular and sensorimotor input but also can be affected by plasticity as we saw, in the previous section.

To achieve such calibration, we propose a simple circuit composed of a bias cell that takes as inputs the perceived AV (excitatory) and the vestibular AV (inhibitory) and, through a recurrent excitation (i.e. it synapses onto itself with an excitatory projection), the cell can maintain firing activity that is proportional to the difference in AV inputs (only when the perceived AV is larger than the vestibular AV). The bias cell also synapses onto the vestibular AV neuron through excitatory projections which allows compensation of the vestibular input to match the perceived AV. This circuit

Model Figure 6. Proposed neural circuit for vestibular input recalibration through bias. Triangular arrowheads indicate excitatory projections. Flat arrowheads indicate inhibitory projection.

has a directional specificity (CW or CCW) and is duplicated to allow readjustment of CW and CCW vestibular inputs separately (Model Fig. 6).

To integrate the recalibration circuit into our model, we first update the output signal of the Bias neurons every timestep dt as follows:

$$\text{perceived } Bias^{CW}(t + dt) = g_{AV} (AV^{CW}_{\text{perceived}}(t) - (AV^{CW}_{\text{Vestibular}}(t) + \beta Bias^{CW}(t))) + W_{B_{CW} \rightarrow B_{CW}} Bias^{CW}(t) \quad (27)$$

$$\text{perceived } Bias^{CCW}(t + dt) = g_{AV} (AV^{CCW}_{\text{perceived}}(t) - (AV^{CCW}_{\text{Vestibular}}(t) + \beta Bias^{CCW}(t))) + W_{B_{CCW} \rightarrow B_{CCW}} Bias^{CCW}(t) \quad (28)$$

where g_{AV} is the synaptic weight of the perceived and vestibular AV neurons’ projections onto the bias cell, β represents the synaptic weight of the excitatory projections from the bias neuron onto the vestibular AV neuron and, $W_{B_{CW} \rightarrow B_{CW}}$ and $W_{B_{CCW} \rightarrow B_{CCW}}$ are the synaptic weights of the recurrent projections of CW and CCW bias neurons, respectively. We ensure that the output of these bias cells is always positive by imposing:

$$Bias^{CW}(t) \leftarrow \max(0, Bias^{CW}(t)) \quad (29)$$

$$Bias^{CCW}(t) \leftarrow \max(0, Bias^{CCW}(t)) \quad (30)$$

While we do not have access to the perceived-AV cells, we may still derive their firing rates by assuming that they are proportional to the speed of rotation of the HD layer's bump of activity θ^{HD} . To allow the bias cells to compare between equivalent AV inputs, the vestibular AV cells' firing rates are also derived from the speed of rotation of the HD layer's bump of activity in a network that assumes perfect integration of the vestibular input θ^{HD} (see section '2- Simulation of drift and output gain'):

$$\begin{aligned} AV^{CW}_{perceived}(t) &= k_{AV} H(\theta^{HD}_{sim}(t)) \\ AV^{CCW}_{perceived}(t) &= k_{AV} H(-\theta^{HD}_{sim}(t)) \\ AV^{CW}_{vestibular}(t) &= k_{AV} H(\theta^{HD}_{ref}(t)) \\ AV^{CCW}_{vestibular}(t) &= k_{AV} H(-\theta^{HD}_{ref}(t)) \end{aligned} \quad (31)$$

Where, k_{AV} is a scaling factor and H is a function defined as:

$$H(x) = \begin{cases} x, & x > 0, \\ 0, & x \leq 0, \end{cases} \quad (32)$$

Linking the perceived and vestibular AV cells' firing rates to the bump of activity's speed of rotation means that their activity is highly dependent on the angular resolution of the network (i.e. number and distribution of HD states (angles) that can be represented) which could limit the network's capability of seamless recalibration across different cue rotation speeds if the resolution is low.

Finally, we modify the AV input to the AV-by-HD cells as follows:

$$AV^{CW}(t) = AV^{CW}(t) + \beta Bias^{CW}_{input}(t) \quad (33)$$

$$AV^{CCW}(t) = AV^{CCW}(t) + \beta Bias^{CCW}_{input}(t) \quad (34)$$

where AV^{CW} and AV^{CCW} are the same as in equation (9).

Upon optimization, the optimal parameter values are: $\beta = 0.434$, $g_{AV} = 1$, $W_{B_{CW} \rightarrow B_{CW}} = 0.1$ and $k_{AV} = 1$.

4) There is an asymmetry in Fig 4f, larger gain difference for negative AV at internal HD ~90 degrees. How should this be interpreted?

We think this may be due to a difference in sampling, which is why we added the significance matrix. Nevertheless, this could be due to undetected irregularities in sensory inputs (e.g. auditory, olfactory, etc...) even though the experimental environment was designed to maximize rotational symmetry of all non-visual sensory cues.

References:

- Bassett, J. P., & Taube, J. S. (2001). Neural correlates for angular head velocity in the rat dorsal tegmental nucleus. *J Neurosci*, 21(15), 5740-5751. <https://www.ncbi.nlm.nih.gov/pubmed/11466446>
- Fisher, Y. E., Lu, J., D'Alessandro, I., & Wilson, R. I. (2019). Sensorimotor experience remaps visual input to a heading-direction network. *Nature*, 576(7785), 121-125. <https://doi.org/10.1038/s41586-019-1772-4>
- Keshavarzi, S., Bracey, E. F., Faville, R. A., Campagner, D., Tyson, A. L., Lenzi, S. C., Branco, T., & Margrie, T. W. (2022). Multisensory coding of angular head velocity in the retrosplenial cortex. *Neuron*, 110(3), 532-543 e539. <https://doi.org/10.1016/j.neuron.2021.10.031>
- Kim, S. S., Hermundstad, A. M., Romani, S., Abbott, L. F., & Jayaraman, V. (2019). Generation of stable heading representations in diverse visual scenes. *Nature*, 576(7785), 126-131. <https://doi.org/10.1038/s41586-019-1767-1>

Turner-Evans, D., Wegener, S., Rouault, H., Franconville, R., Wolff, T., Seelig, J. D., Druckmann, S., & Jayaraman, V. (2017). Angular velocity integration in a fly heading circuit. *Elife*, 6.

<https://doi.org/10.7554/eLife.23496>

Cope, A. J., Sabo, C., Vasilaki, E., Barron, A. B., & Marshall, J. A. (2017). A computational model of the integration of landmarks and motion in the insect central complex. *PLoS One*, 12(2), e0172325.

<https://doi.org/10.1371/journal.pone.0172325>

Ocko, S. A., Hardcastle, K., Giocomo, L. M., & Ganguli, S. (2018). Emergent elasticity in the neural code for space. *Proc Natl Acad Sci U S A*, 115(50), E11798-E11806. <https://doi.org/10.1073/pnas.1805959115>

Page, H. J. I., & Jeffery, K. J. (2018). Landmark-Based Updating of the Head Direction System by Retrosplenial Cortex: A Computational Model. *Front Cell Neurosci*, 12, 191.

<https://doi.org/10.3389/fncel.2018.00191>

Reviewer Reports on the First Revision:

Referees' comments:

Referee #1 (Remarks to the Author):

The authors have done a good and thorough job of responding to my comments, and the additional modelling is impressive.

I have one small remaining issue which I don't feel strongly about and it is not an impediment to publication, but it's this: I still don't find the polar representation an intuitive explanation for the relationship between reorientation and gain. The polar representation is a description, not an explanation. Both the polar plot and the data share the property that movement along one dimension (radially) affects the amount of change in the other (angularly), and the polar plot is a good way to *visualise* this, but does not *explain* it. Models are not mechanisms (I see this conflation not infrequently in the literature).

That said, I'm not so bothered about this small point that I think it needs addressing. The rest of the paper is a useful contribution to the literature.

Kate Jeffery

Referee #2 (Remarks to the Author):

The revised manuscript essentially addresses my previous concerns. I think that the novel experimental observations, in particular the finding of long-term experience-dependent firing of HD cells, will be of interest to spatial navigation researchers in the systems, computational, and theoretical neuroscience communities.

I only have a few comments:

1) I find the additional modeling work very valuable, as it succinctly summarizes some of the key findings, and it provides possible hypotheses that could be tested in future experiments. As such, I think that it would be useful for the modeling work to be more prominently displayed, rather than somewhat buried in Extended Data Figures. For instance, some parts of EDF19 could be perhaps moved to Fig 4 (with an emphasis on the model's relationship with the experimental data in Fig 3 and 4). Similarly, part of EDF21 could be moved to Fig 5.

2) In reference to the answer to my question about the dimensionality of the data in the previous set of comments: I agree with the authors that the issue of additional dimensions present in the data would be outside the scope of the current work. However, I think it'd be valuable for other

researchers to get a sense of how much of the variance in the data can be explained by changes in gain. It is quite common to apply simple dimensionality reduction to data. Would restricting the data to the first few principal components, for instance, preclude the observation of gain changes that the authors have convincingly shown to be important?

3) In reference to a previous point about possible model's extensions that would capture additional relevant variables: I appreciate the points that the authors make, i.e. that additional assumptions would be required, for instance gain likely reflecting the influence of multiple inputs. The insightful points exposed by the authors in that section could be moved to Discussion.

4) Minor: Could the authors double check the order of EDFs? I think that the references to EDFs might jump around a bit, e.g. refs to EDF18,19 are used before referencing EDF 17.

Author Rebuttals to First Revision:

Referees' comments:

First, we would like to thank both reviewers for their comments/remarks. We would like to acknowledge the fact that their previous and present suggestions helped substantially in improving the manuscript in many ways.

Referee #1 (Remarks to the Author):

The authors have a done good and thorough job of responding to my comments, and the additional modelling is impressive.

I have one small remaining issue which I don't feel strongly about and it is not an impediment to publication, but it's this: I sill don't find the polar representation an untuitive explanation for the relationship between reorientation and gain. The polar representation is a description, not an explanation. Both the polar ploat and the data share the property that movement along one dimension (radially) affects the amount of change in the other (angularly), and the polar plot is a good way to *visualise* this, but does not *explain* it. Models are not mechanisms (I see this conflation not infrequently in the literature).

That said, I'm not so bothered about this small point that I think it needs addressing. The rest of the paper is a useful contribution to the literature.

Kate Jeffery

We thank Prof. Jefferey for her positive assessment and important remarks. We understand the point made about the utility of using the polar representation/model as a description and not an explanation of actual mechanisms. In the main text, we specified that this is for visualization purposes:

“To visualize the low-dimensional structure of the HD representation, we developed a method to project large ensemble recordings onto a 2-dimensional polar state-space (Methods).”

Also, in the main text, we specified that the intuition gained from the polar description is only used to motivate our hypothesis about the relationship between radius and speed of reorientation:

“Intuitively, allowing the internal HD representation to occupy a 2D polar state space makes the distance between any two given angles θ_1 and θ_2 a function of the radial component, which led us to hypothesize that changes in radius not only reflected changes in the overall population activity but would also correlate with changes in the speed of reset (Extended Data Fig. 5).”

To fully address Prof. Jefferey's comment, we changed the title of Extended Data Figure 5 which now reads:

“Schematic showing a visual description of the hypothesized relationship between the population-activity and movements in the polar state-space and, how they may relate to the dynamics of the internal HD representation.”

Referee #2 (Remarks to the Author):

The revised manuscript essentially addresses my previous concerns. I think that the novel experimental observations, in particular the finding of long-term experience-dependent firing of HD cells, will be of interest to spatial navigation researchers in the systems, computational, and theoretical neuroscience communities.

We thank the reviewer for their positive assessment of our work and their suggestions for further improvements.

I only have a few comments:

1) I find the additional modeling work very valuable, as it succinctly summarizes some of the key findings, and it provides possible hypotheses that could be tested in future experiments. As such, I think that it would be useful for the modeling work to be more prominently displayed, rather than somewhat buried in Extended Data Figures. For instance, some parts of EDF19 could be perhaps moved to Fig 4 (with an emphasis on the model's relationship with the experimental data in Fig 3 and 4). Similarly, part of EDF21 could be moved to Fig 5.

We thank the reviewer for their comment on the modeling work. We agree that the model could be more prominently displayed in the main figures. However, as much as we tried to reduce the length of the manuscript, we think that we only are at the limit of what is allowed for publication. As such, adding more panels and figure legends will make the manuscript considerably exceed the said limit. This is why we kept the model figures in the Extended Data Figure section while referring to it in the main text.

2) In reference to the answer to my question about the dimensionality of the data in the previous set of comments: I agree with the authors that the issue of additional dimensions present in the data would be outside the scope of the current work. However, I think it'd be valuable for other researchers to get a sense of how much of the variance in the data can be explained by changes in gain. It is quite common to apply simple dimensionality reduction to data. Would restricting the data to the first few principal components, for instance, preclude the observation of gain changes that the authors have convincingly shown to be important?

We thank the reviewer for this comment and the suggested analysis to determine the importance of gain as a secondary dimension in terms variance explained. We propose a slightly different – but equivalent – approach to address this point. Below is a figure (and its legend) showing a schematic describing the approach as well as the results of the analysis:

Extended Data Figure 22. Variance explained by gain. *a.* Method used to determine the variance explained by gain. Using the internal HD and neural activity from all recorded neurons ($S_{\text{neuron } i}$; 5 examples shown here for illustration purposes) per session as inputs, we can extract the tuning curve of each neuron (average firing rate as a function of internal HD throughout the session, $f(\theta_t)$) as well as the gain signal (g_t), while assuming that pairwise coherence between HD cells is preserved. Two reconstructions of the neural activity are then produced from tuning curves and internal HD: In the first case (Dark blue traces) neural activity is multiplied by gain signal ($R_{\text{neuron } i}^g$) while in the second case (Light blue traces) the gain signal is not taken into account ($R_{\text{neuron } i}$). The sum of variance across neurons is calculated for each group of neural activity (including the ground truth). *b.* Comparison of variance explained in percentage between the neural activity reconstruction with and without gain (sum of variance in each group is divided by the sum of variance in the ground truth group) ($n=42$ sessions; Two-sided Wilcoxon rank-sum test: $p=0.0245$, $Z=2.2499$). Error bars show $\text{mean} \pm \text{SEM}$. *c.* Increase in variance explained when gain is applied to reconstructed neural activity relative to the case where gain is not applied (i.e. ratio between % variance explained with and without gain, minus 1) ($n=42$ sessions; Mean=13.71%, STD=5.14%). Error bars show $\text{mean} \pm \text{STD}$. Dots represent individual data points.

3) In reference to a previous point about possible model's extensions that would capture additional relevant variables: I appreciate the points that the authors make, i.e. that additional assumptions would be required, for instance gain likely reflecting the influence of multiple inputs. The insightful points exposed by the authors in that section could be moved to Discussion.

We thank the reviewer for this comment. We agree with his suggestion about adding the points we presented in our previous response to the discussion section. However, here also, we found it not possible to add further text to the manuscript without exceeding the limits allowed for publication, even after all our efforts to reduce the length as much as possible.

4) Minor: Could the authors double check the order of EDFs? I think that the references to EDFs might jump around a bit, e.g. refs to EDF18,19 are used before referencing EDF 17.

We thank the reviewer for noting the mistake in the order of EDFs. We corrected this and reordered the EDFs to comply with the editorial guidelines.